# Natural Posterior Network: Deep Bayesian Uncertainty for Exponential Family Distributions

**Bertrand Charpentier,** **Oliver Borchert,** **Daniel Zügner, Simon Geisler, Stephan Günnemann**
Department of Informatics & Munich Data Science Institute
Technical University of Munich, Germany
`{charpent, borchero, zuegnerd, geisler, guennemann}@in.tum.de`

## Abstract

Uncertainty awareness is crucial to develop reliable machine learning models. In this work, we propose the Natural Posterior Network (NatPN) for fast and high-quality uncertainty estimation for any task where the target distribution belongs to the exponential family. Thus, NatPN finds application for both classification and general regression settings. Unlike many previous approaches, NatPN does not require out-of-distribution (OOD) data at training time. Instead, it leverages Normalizing Flows to fit a single density on a learned low-dimensional and task-dependent latent space. For any input sample, NatPN uses the predicted likelihood to perform a Bayesian update over the target distribution. Theoretically, NatPN assigns high uncertainty far away from training data. Empirically, our extensive experiments on calibration and OOD detection show that NatPN delivers highly competitive performance for classification, regression and count prediction tasks.

## 1 Introduction

Accurate and rigorous uncertainty estimation is key for reliable machine learning models in safety-critical domains. It quantifies the confidence of machine learning models, thus allowing them to validate knowledgeable predictions corresponding to correct/wrong predictions, flag predictions on unknown input domains corresponding to anomaly or Out-of-Distribution detection, or detect natural shifts of the data facilitating real-time model maintenance (Filos et al., 2019; Malinin et al., 2021; Ovadia et al., 2019). Specifically, a reliable model can handle all these failure modes with high-quality estimates of *aleatoric* and *epistemic* uncertainty (Gal, 2016). These two levels of uncertainty allow a model to account for both irreducible data uncertainty (e.g. a fair dice's chance of $1/6$ for each face) and uncertainty due to the lack of knowledge about unseen data (e.g. input features differing significantly from training data or a covariate shift) respectively. Aleatoric and epistemic uncertainty levels can eventually be combined into an overall *predictive uncertainty* (Gal, 2016)

Traditional neural networks are not readily applicable in safety-critical domains as they show overconfident prediction, in particular on data that is different from training data (Guo et al., 2017; Lakshminarayanan

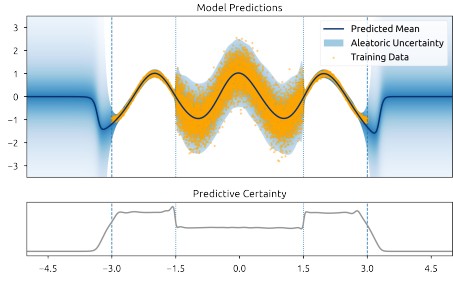

Toy Regression Task

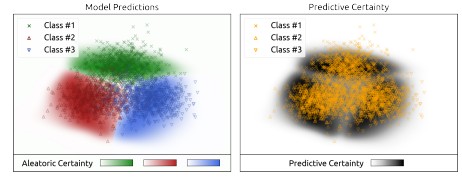

Toy Classification Task

Figure 1: Visualization of the aleatoric and predictive uncertainty estimates of NatPN on two toy regressions and classifications tasks. NatPN correctly assigns higher uncertainty to regions far from the training data.

---

*Equal contribution

et al., 2017). To mitigate this problem, an important model family for uncertainty estimation directly predicts the parameters of a *conjugate prior distribution* on the predicted *target probability distribution*, thus accounting for the different levels of uncertainty. These models are efficient as they only require *a single forward pass* for target and uncertainty prediction. Most of those models focus on classification and thus predict parameters of a Dirichlet distribution (Biloš et al., 2019; Charpentier et al., 2020; Gast & Roth, 2018; Malinin & Gales, 2018; 2019; Nandy et al., 2020; Sensoy et al., 2020; Shi et al., 2020; Stadler et al., 2021; Zhao et al., 2020b). However, only two works (Amini et al., 2020; Malinin et al., 2020a) have focused on regression by learning parameters of a Normal Inverse-Gamma (NIG) distribution as conjugate prior. Hence, all these models are limited to a *single* task (e.g. either classification or regression). Some approaches even require out-of-distribution (OOD) data at training time (Malinin & Gales, 2018; 2019) which is an unrealistic assumption in many real-world applications where anomalies are a priori diverse, rare or unknown.

**Our contribution.** We propose Natural Posterior Network (NatPN) as a new approach parametrizing conjugate prior distributions for versatile uncertainty estimation. NatPN is motivated from both the theoretical and practical perspective. **(1)** NatPN can estimate predictive uncertainty for *any* task described by the general group of exponential family distributions contrary to existing approaches from this family of models. Notably, this encompasses very common tasks such as classification, regression and count prediction which can be described with Categorical, Normal and Poisson distributions, respectively. **(2)** In theory, NatPN is based on a *new unified exponential family framework* which performs an input-dependent Bayesian update. For every input, it predicts the parameters of the posterior over the target exponential family distribution. We show that this Bayesian update is *guaranteed* to predict high uncertainty far from training data. **(3)** In practice, NatPN requires *no OOD data for training*, only adds *a single normalizing flow* density to the last predictor layer and provides fast uncertainty estimation in a *single forward pass*. Our extensive experiments showcase the high performances of NatPN for various criteria (accuracy, calibration, OOD and shift detection) and tasks (classification, regression and count prediction). We illustrate the accurate aleatoric and predictive uncertainty predictions of NatPN on two toy examples for classification and regression in Fig. 1. *None of the conjugate prior related works* have similar theoretical and practical properties.

## 2 RELATED WORK

In this section, we describe other work related to uncertainty estimation for supervised learning. We refer to Gawlikowski et al. (2021) for a detailed survey on uncertainty estimation in deep learning.

**Sampling-based methods.** A first family of models estimates uncertainty by aggregating statistics (e.g. mean and variance) from different samples of an implicit predictive distribution. Examples are ensemble (Kim & Ghahramani, 2012; Lakshminarayanan et al., 2017; Simpson et al., 2012; Wen et al., 2020; Wenzel et al., 2020) and dropout (Gal & Ghahramani, 2016) models which provide high-quality uncertainty estimates (Ovadia et al., 2019) at the cost of an expensive sampling phase at inference time. Moreover, ensembles usually require training multiple models. Further, Bayesian neural networks (BNN) (Blundell et al., 2015; Maddox et al., 2019; Ritter et al., 2018) model the uncertainty on the weights and also require multiple samples to estimate the uncertainty on the final prediction. While recent BNNs have shown reasonably good performance (Dusenberry et al., 2020; Farquhar et al., 2020; Osawa et al., 2019), modelling the distribution on the weights suffers from pathological behavior thus limiting these approaches in practice (Foong et al., 2020; Graves, 2011; Izmailov et al., 2021). In particular, Izmailov et al. (2021) uses an enormous computation budget by parallelizing the computation over 512 TPUv3 devices and running tens of thousands of training epochs to achieve a more exact Bayesian inference which is not suitable for practical applications. In contrast, NatPN predicts uncertainty in *a single forward pass* with a *closed-form posterior distribution* over the target variable. NatPN *does not* model uncertainty on the weights.

**Sampling-free methods.** A second family of models is capable of estimating uncertainty in a single forward pass. The family of models parametrizing conjugate prior distributions is the main focus of this paper (Gast & Roth, 2018; Kopetzki et al., 2021; Nandy et al., 2020; Sensoy et al., 2020; Shi et al., 2020; Stadler et al., 2021; Ulmer, 2021). Beyond this family of models, we differentiate between four other families of sampling-free models for uncertainty estimation. A first family aims at learning deep Gaussian processes with random features projections or learned inducing points (Biloš et al., 2019; Lakshminarayanan et al., 2020; van Amersfoort et al., 2020; 2021). A second family aims at learning

deep energy-based models (Elflein et al., 2021; Grathwohl et al., 2020). Another family of models aims at propagating uncertainty across layers (Gast & Roth, 2018; Hernandez-Lobato & Adams, 2015; Postels et al., 2019; Shekhovtsov & Flach, 2019; Wang et al., 2016). They model uncertainty at the weight and/or activation levels and are generally constrained to specific transformations. In contrast, NatPN only models the uncertainty on the predicted target variable and does not enforce any constraint on the encoder architecture. Further, some of the models propagating uncertainty already used the exponential family framework (Ranganath et al., 2015; Wang et al., 2016). However, while they parametrize exponential family distributions, NatPN parametrizes the *conjugate prior of the target exponential family distributions* which accounts for the epistemic uncertainty. Finally, while the family of calibration models aims at calibrating predictions (Kuleshov et al., 2018; Moon et al., 2020; Rahimi et al., 2020; Song et al., 2019; Zhao et al., 2020a), NatPN aims at accurately modelling both aleatoric and epistemic uncertainty on in- and out-of-distribution data.

# 3 NATURAL POSTERIOR NETWORK

At the very core of NatPN stands the Bayesian update rule: $\mathbb{Q}(\boldsymbol{\theta} \mid \mathcal{D}) \propto \mathbb{P}(\mathcal{D} \mid \boldsymbol{\theta}) \times \mathbb{Q}(\boldsymbol{\theta})$ where $\mathbb{P}(\mathcal{D} \mid \boldsymbol{\theta})$ is the target distribution of the target data $\mathcal{D}$ given its parameter $\boldsymbol{\theta}$, and $\mathbb{Q}(\boldsymbol{\theta})$ and $\mathbb{Q}(\boldsymbol{\theta} \mid \mathcal{D})$ are the prior and posterior distributions, respectively, over the target distribution parameters. The target distribution $\mathbb{P}(\mathcal{D} \mid \boldsymbol{\theta})$ could be any likelihood describing the observed target labels. The Bayesian update has three main advantages: **(1)** it introduces a prior belief which represents the safe default prediction if no data is observed, **(2)** it updates the prior prediction based on observed target labels, and **(3)** it assigns a confidence for the new target prediction given the aggregated evidence count of observed target labels. While NatPN is capable to perform a Bayesian update for every possible input given the observed training data, we first recall the Bayesian background for a single exponential family distribution.

| Likelihood $\mathbb{P}$ | Conjugate Prior $\mathbb{Q}$ | Parametrization Mapping $m$ | Bayesian Loss (Eq. 5) |
|---|---|---|---|
| $y \sim \mathrm{Cat}(\boldsymbol{p})$ | $\boldsymbol{p} \sim \mathrm{Dir}(\boldsymbol{\alpha})$ | $\boldsymbol{\chi} = \boldsymbol{\alpha}/n$ $n = \sum_c \alpha_c$ | **(i)** $= \psi(\alpha_{y*}^{(i)}) - \psi(\alpha_0^{(i)})$ **(ii)** $= \log B(\boldsymbol{\alpha}^{(i)}) + (\alpha_0^{(i)} - C)\psi(\alpha_0^{(i)}) - \sum_c (\alpha_c^{(i)} - 1)\psi(\alpha_c^{(i)})$ |
| $y \sim \mathcal{N}(\mu, \sigma)$ | $\mu, \sigma \sim \mathcal{N}\Gamma^{-1}(\mu_0, \lambda, \alpha, \beta)$ | $\boldsymbol{\chi} = \begin{pmatrix} \mu_0 \\ \mu_0^2 + \frac{2\beta}{n} \end{pmatrix}$ $n = \lambda = 2\alpha$ | **(i)** $= \frac{1}{2}\left(-\frac{\alpha}{\beta}(y - \mu_0)^2 - \frac{1}{\lambda} + \psi(\alpha) - \log\beta - \log 2\pi\right)$ **(ii)** $= \frac{1}{2} + \log\left((2\pi)^{\frac{1}{2}}\beta^{\frac{3}{2}}\Gamma(\alpha)\right) - \frac{1}{2}\log\lambda + \alpha - (\alpha + \frac{3}{2})\psi(\alpha)$ |
| $y \sim \mathrm{Poi}(\lambda)$ | $\lambda \sim \Gamma(\alpha, \beta)$ | $\chi = \alpha/n$ $n = \beta$ | **(i)** $= (\psi(\alpha) - \log\beta)y - \frac{\alpha}{\beta} - \sum_{k=1}^{y} \log k$ **(ii)** $= \alpha + \log\Gamma(\alpha) - \log\beta + (1 - \alpha)\psi(\alpha)$ |

Table 1: Examples of Exponential Family Distributions where $\psi(x)$ and $B(x)$ denote Digamma and Beta function, respectively.

## 3.1 EXPONENTIAL FAMILY DISTRIBUTION

Distributions from the exponential family are very widely used and have favorable analytical properties. Indeed, **(1)** they cover a wide range of target variables like discrete, continuous, counts or spherical coordinates, and **(2)** they benefit from intuitive and generic formulae for their parameters, density functions and statistics which can often be evaluated in closed-form. Important examples of exponential family distributions are Normal, Categorical and Poisson distributions (see Tab. 1). Formally, an exponential family distribution on a target variable $y \in \mathbb{R}$ with *natural parameters* $\boldsymbol{\theta} \in \mathbb{R}^L$ can be denoted as

$$\mathbb{P}(y \mid \boldsymbol{\theta}) = h(y)\exp\left(\boldsymbol{\theta}^T \boldsymbol{u}(y) - A(\boldsymbol{\theta})\right) \tag{1}$$

where $h : \mathbb{R} \to \mathbb{R}$ is the *carrier or base measure*, $A : \mathbb{R}^L \to \mathbb{R}$ the *log-normalizer* and $\boldsymbol{u} : \mathbb{R} \to \mathbb{R}^L$ the *sufficient statistics* (Bishop, 2006; Nielsen & Nock, 2010). The entropy of an exponential family distribution can always be written as $\mathbb{H}[\mathbb{P}] = A(\boldsymbol{\theta}) - \boldsymbol{\theta}^T \nabla_{\boldsymbol{\theta}} A(\boldsymbol{\theta}) - \mathbb{E}[\log h(y)]$ (Nielsen & Nock, 2010). An exponential family distribution always admits a conjugate prior, which often also is a member of the exponential family:

$$\mathbb{Q}(\boldsymbol{\theta} \mid \boldsymbol{\chi}, n) = \eta(\boldsymbol{\chi}, n)\exp\left(n\,\boldsymbol{\theta}^T \boldsymbol{\chi} - nA(\boldsymbol{\theta})\right) \tag{2}$$

where $\eta(\boldsymbol{\chi}, n)$ is a normalization coefficient, $\boldsymbol{\chi} \in \mathbb{R}^L$ are *prior parameters* and $n \in \mathbb{R}^+$ is the *evidence*. Given a set of $N$ target observations $\{y^{(i)}\}_i^N$, it is easy to compute a closed-form Bayesian update $\mathbb{Q}(\boldsymbol{\theta} \mid \boldsymbol{\chi}^{\mathrm{post}}, n^{\mathrm{post}}) \propto \mathbb{P}(\{y^{(i)}\}_i^N \mid \boldsymbol{\theta}) \times \mathbb{Q}(\boldsymbol{\theta} \mid \boldsymbol{\chi}^{\mathrm{prior}}, n^{\mathrm{prior}})$:

$$\mathbb{Q}(\boldsymbol{\theta} \mid \boldsymbol{\chi}^{\mathrm{post}}, n^{\mathrm{post}}) \propto \exp\left(n^{\mathrm{post}}\boldsymbol{\theta}^T \boldsymbol{\chi}^{\mathrm{post}} - n^{\mathrm{post}}A(\boldsymbol{\theta})\right) \tag{3}$$

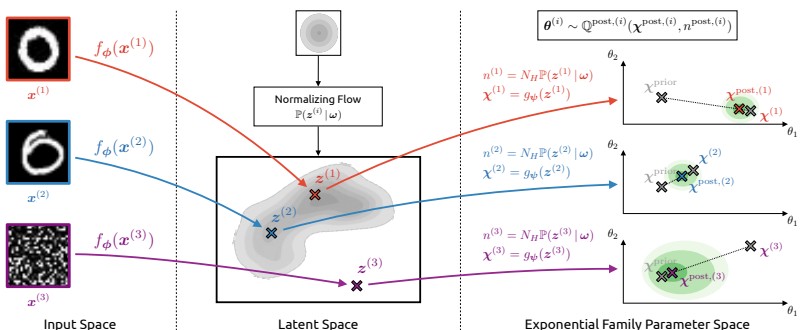

Figure 2: Overview of Natural Posterior Network. Inputs $\boldsymbol{x}^{(i)}$ are first mapped to a low-dimensional latent representation $\boldsymbol{z}^{(i)}$ by the encoder $f_\phi$. From $\boldsymbol{z}^{(i)}$, the decoder $g_\psi$ derives the parameter update $\boldsymbol{\chi}^{(i)}$ while a normalizing flow $\mathbb{P}_{\boldsymbol{\omega}}$ yields the evidence update $n^{(i)}$. Posterior parameters are obtained from a weighted combination of prior and update parameters according to $n^{\text{post},(i)}$.

where $\boldsymbol{\chi}^{\text{post}} = \frac{n^{\text{prior}}\boldsymbol{\chi}^{\text{prior}} + \sum_j^N \boldsymbol{u}(y^{(j)})}{n^{\text{prior}} + N}$ and $n^{\text{post}} = n^{\text{prior}} + N$. We see that $\boldsymbol{\chi}^{\text{prior}}$ (resp. $\boldsymbol{\chi}^{\text{post}}$) can be viewed as the average sufficient statistics of $n^{\text{prior}}$ (resp. $n^{\text{post}}$) fictitious samples (Bishop, 2006). Further, the average sufficient statistic of fictitious samples is equal to the expected sufficient statistic of the conjugate distribution, i.e. $\boldsymbol{\chi} = \mathbb{E}_{\mathbb{Q}(\boldsymbol{\chi},n)}[\boldsymbol{\theta}]$ (Brown, 1986; Diaconis & Ylvisaker, 1979). Thus, the parameter $\boldsymbol{\chi}^{\text{post}}$ carries the inherent aleatoric uncertainty on the target distribution with natural parameters $\boldsymbol{\theta}$, while the evidence $n^{\text{post}}$ aligns well with the epistemic uncertainty (i.e. a low evidence means few prior target observations). We stress that the natural conjugate prior parametrization $\boldsymbol{\chi}, n$ is often different from the "well-known" parametrization $\boldsymbol{\kappa}$ used by standard coding libraries. By definition, a bijective mapping $m(\boldsymbol{\kappa}) = (\boldsymbol{\chi}, n)$ from the natural parametrization to the commonly used parametrization always exists (see examples in Tab. 1). Finally, exponential family distributions always admit a closed-form posterior predictive distribution (Gelman et al., 2004).

## 3.2 INPUT-DEPENDENT BAYESIAN UPDATE FOR EXPONENTIAL FAMILY DISTRIBUTIONS

We propose to leverage the power of exponential family distributions for the more complex task when the prediction $y^{(i)}$ depends on the input $\boldsymbol{x}^{(i)}$. Hence, NatPN extends the Bayesian treatment of a single exponential family distribution prediction by predicting an individual posterior update per input. We distinguish between the chosen prior parameters $\boldsymbol{\chi}^{\text{prior}}, n^{\text{prior}}$ shared among samples, and the additional predicted parameters $\boldsymbol{\chi}^{(i)}, n^{(i)}$ dependent on the input $\boldsymbol{x}^{(i)}$ leading to the updated posterior parameters:

$$\boldsymbol{\chi}^{\text{post},(i)} = \frac{n^{\text{prior}}\boldsymbol{\chi}^{\text{prior}} + n^{(i)}\boldsymbol{\chi}^{(i)}}{n^{\text{prior}} + n^{(i)}}, \qquad n^{\text{post},(i)} = n^{\text{prior}} + n^{(i)} \qquad (4)$$

Equivalently, NatPN may be interpreted as predicting a set of $n^{(i)}$ pseudo observations $\{y^{(j)}\}_j^{(i)}$ such that their aggregated sufficient statistics satisfy $\sum_j^{n^{(i)}} y^{(j)} = n^{(i)}\boldsymbol{\chi}^{(i)}$, and perform the respective Bayesian update. This Bayesian update works for *any* choice of exponential family distributions as long as parameters are mapped to their standard form (see Tab. 1). According to the *principle of maximum entropy* (Rosenkrantz, 1989), a practical choice for the prior is to enforce high entropy for the prior distribution which is usually considered less informative. It is typically achieved when the prior pseudo-count $n^{\text{prior}}$ is small and the prior parameter $\boldsymbol{\chi}^{\text{prior}}$ shows a high aleatoric uncertainty.

Hence, NatPN proposes a generic way to perform the input-dependent Bayesian update $\boldsymbol{\chi}^{(i)}, n^{(i)}$ for *any* exponential family distribution in three steps (see Fig. 2): **(1)** An encoder $f_\phi$ maps the input $\boldsymbol{x}^{(i)}$ onto a low-dimensional latent vector $\boldsymbol{z}^{(i)} = f_\phi(\boldsymbol{x}^{(i)}) \in \mathbb{R}^H$ representing useful features for the prediction task (see left Fig.2). Note that the architecture of the encoder can be arbitrarily complex. Then, **(2)** the latent representation $\boldsymbol{z}^{(i)}$ is used in two different ways to predict the parameter update $\boldsymbol{\chi}^{(i)}$ and the evidence update $n^{(i)}$ (see center Fig.2). On the one hand, a linear decoder $g_\psi$ is trained to output the parameter update $\boldsymbol{\chi}^{(i)} = g_\psi(\boldsymbol{z}^{(i)}) \in \mathbb{R}^L$ accounting for the aleatoric uncertainty. On the other hand, a single normalized density is trained to output the evidence update $n^{(i)} = N_H\,\mathbb{P}(\boldsymbol{z}^{(i)}\,|\,\boldsymbol{\omega})$ accounting for the epistemic uncertainty. The intuition is that increasing the evidence on training data during training forces the evidence everywhere else (incl. far from

training data) to decrease thanks to the density normalization constraint. The constant $N_H$ is a certainty budget distributed by the normalized density $\mathbb{P}(z^{(i)} \mid \omega)$ over the latent representations $z^{(i)}$ i.e. $N_H = \int N_H \mathbb{P}(z^{(i)} \mid \omega) dz^{(i)} = \int n^{(i)} dz^{(i)}$. In practice, we observed that scaling the certainty budget w.r.t. the latent dimension $H$ helped the density to cover larger volumes in higher dimension (see app.). Finally, **(3)** NatPN computes the posterior parameters $\chi^{\text{post},(i)}$ and $n^{\text{post},(i)}$ which can be viewed respectively as the mean and concentration of the posterior distribution (see right Fig.2). Note that the posterior parameter $\chi^{\text{post},(i)}$ is a simple weighted average of the prior parameter $\chi^{\text{prior}}$ and the update parameter $\chi^{(i)}$ as shown by Eq. 4.

NatPN extends PostNet (Charpentier et al., 2020) which also performs an input-dependent Bayesian update with density estimation. Yet, it has three crucial differences which lead to major practical improvements. First, the new exponential family framework is significantly more flexible and is not restricted to classification. Second, the Dirichlet $\alpha$ parameter computation is different: NatPN computes the $\chi$ parameters – which can be viewed as standard softmax output – and the $n$ evidence separately (i.e. $\alpha = n\chi$) while PostNet computes one evidence pseudo-count per class. Third, NatPN is computationally more efficient. It requires a single density while PostNet requires $C$ densities.

### 3.3 ID and OOD Uncertainty Estimates

NatPN intuitively leads to reasonable uncertainty estimation for the two limit cases of strong in-distribution (ID) and out-of-distribution (OOD) inputs (see red and purple samples in Fig. 2). For very likely *in-distribution* data (i.e. $\mathbb{P}(z^{(i)} \mid \omega) \to \infty$), the posterior parameter overrules the prior (i.e. $\chi^{\text{post},(i)} \to \chi^{(i)}$). Conversely, for very unlikely *out-of-distribution* data (i.e. $\mathbb{P}(z^{(i)} \mid \omega) \to 0$), the prior parameter takes over in the posterior update (i.e. $\chi^{\text{post},(i)} \to \chi^{\text{prior}}$). Hence, the choice of the prior parameter should reflect the default prediction when the model lacks knowledge. We formally show under mild assumptions on the encoder that NatPN predicts very low additional evidence ($n^{(i)} \approx 0$) for (almost) any input $x^{(i)}$ far away from the training data (i.e. $||x^{(i)}|| \to +\infty$), thus recovering prior predictions (i.e. $\chi^{\text{post},(i)} \approx \chi^{\text{prior}}$) (see proof in app.).

**Theorem 1.** *Let a NatPN model be parametrized with a (deep) encoder $f_\phi$ with ReLU activations, a decoder $g_\psi$ and the density $\mathbb{P}(z \mid \omega)$. Let $f_\phi(x) = V^{(l)} x + a^{(l)}$ be the piecewise affine representation of the ReLU network $f_\phi$ on the finite number of affine regions $Q^{(l)}$ (Arora et al., 2018). Suppose that $V^{(l)}$ have independent rows and the density function $\mathbb{P}(z \mid \omega)$ has bounded derivatives, then for almost any $x$ we have $\mathbb{P}(f_\phi(\delta \cdot x) \mid \omega) \underset{\delta \to \infty}{\to} 0$. i.e the evidence becomes small far from training data.*

This theorem only requires that the density avoids very unlikely pathological behavior with unbounded derivatives (Dix, 2013). A slightly weaker conclusion holds using the notion of limit in density if the density function does not have bounded derivatives (Niculescu & Popovici, 2011). Finally, the independent rows condition is realistic for trained networks with no constant output (Hein et al., 2019). It advantageously leads NatPN to consistent uncertainty estimation contrary to standard ReLU networks which are overconfident far from training data (Hein et al., 2019).

### 3.4 Bayesian NatPN Ensemble

Interestingly, it is natural to extend the Bayesian treatment of a single NatPN to an ensemble of NatPN models (NatPE). An ensemble of $m$ NatPN models is intuitively equivalent to performing $m$ successive Bayesian updates using each NatPN member separately. More formally, given an input $x^{(i)}$ and an ensemble of $m$ jointly trained NatPN models, the Bayesian update for the posterior distribution becomes $\chi^{\text{post},(i)} = \frac{n^{\text{prior}} \chi^{\text{prior}} + \sum_k^m n_k^{(i)} \chi_k^{(i)}}{n^{\text{prior}} + \sum_k^m n_k^{(i)}}$ and $n^{\text{post},(i)} = n^{\text{prior}} + \sum_k^m n_k^{(i)}$. Note that the standard Bayesian averaging which is used in many ensembling methods (Chipman et al., 2007; Lakshminarayanan et al., 2017; Wen et al., 2020; Wenzel et al., 2020) is different from this Bayesian combination. While Bayesian averaging assume that only one model is correct, the Bayesian combination of NatPN allows *more* or *none* of the models to be "expert" for some input (Monteith et al., 2011). For example, an input $x^{(i)}$ unfamiliar to every model $m$ (i.e. $n_m^{(i)} \approx 0$) would recover the prior default prediction $\chi^{\text{prior}}, n^{\text{prior}}$. Existing models already had similar properties for Bayesian combination of classifiers (Kim & Ghahramani, 2012; Simpson et al., 2012).

## 3.5 Optimization

The choice of the optimization procedure is of primary importance in order to obtain both high-quality target predictions and uncertainty estimates regardless of the task.

**Bayesian Loss.** We follow Charpentier et al. (2020) and aim at minimizing the Bayesian formulation:

$$\mathcal{L}^{(i)} = -\underbrace{\mathbb{E}_{\boldsymbol{\theta}^{(i)} \sim \mathbb{Q}^{\text{post},(i)}}[\log \mathbb{P}(y^{(i)} \mid \boldsymbol{\theta}^{(i)})]}_{\text{(i)}} - \underbrace{\mathbb{H}[\mathbb{Q}^{\text{post},(i)}]}_{\text{(ii)}} \tag{5}$$

where $\mathbb{H}[\mathbb{Q}^{\text{post},(i)}]$ denotes the entropy of the predicted posterior distribution $\mathbb{Q}^{\text{post},(i)}$. Similarly to the ELBO loss, this loss is guaranteed to be optimal when the predicted posterior distribution is close to the true posterior distribution $\mathbb{Q}^*(\boldsymbol{\theta} \mid \boldsymbol{x}^{(i)})$ i.e. $\mathbb{Q}^{\text{post},(i)} \approx \mathbb{Q}^*(\boldsymbol{\theta} \mid \boldsymbol{x}^{(i)})$ (Bissiri et al., 2016; Shawe-Taylor & Williamson, 1997; Zellner, 1988). However, this loss is generally *not* equal to the ELBO loss especially for real valued targets i.e. $y \in \mathbb{R}$ (see app.). The term **(i)** is the expected likelihood under the predicted posterior distribution. It can be viewed as the Uncertain Cross Entropy (UCE) loss (Biloš et al., 2019) which is known to reduce uncertainty on observed data. The term **(ii)** is an entropy regularizer acting as a prior which favors uninformative distributions $\mathbb{Q}^{\text{post},(i)}$ with high entropy. In our case, we assume the likelihood $\mathbb{P}(y^{(i)} \mid \boldsymbol{\theta}^{(i)})$ and the posterior $\mathbb{Q}^{\text{post},(i)}$ to be members of the exponential family. We take advantage of the convenient computations for such distributions and derive a more explicit formula for the Bayesian formulation (5) (see derivation in the appendix):

$$\mathcal{L}^{(i)}_{\lambda} \propto \mathbb{E}[\boldsymbol{\theta}]^T \boldsymbol{u}(y^{(i)}) - \mathbb{E}[A(\boldsymbol{\theta})] - \lambda \, \mathbb{H}[\mathbb{Q}^{\text{post},(i)}] \tag{6}$$

where $\lambda$ is an additional regularization weight tuned with a grid search. Note that the term $\mathbb{E}[\boldsymbol{\theta}]^T \boldsymbol{u}(y^{(i)})$ favors a good alignment of the expected sufficient statistic $\mathbb{E}[\boldsymbol{\theta}] = \boldsymbol{\chi}$ with the observed sufficient statistic $\boldsymbol{u}(y^{(i)})$. In practice, all terms can be computed efficiently in closed form for most exponential family distributions (see examples in Tab. 1). In particular, simplifications are possible when the conjugate prior distribution is also in an exponential family which is often the case. Ultimately Eq. (6) applies to *any* exponential family distribution unlike Charpentier et al. (2020).

**Optimization Scheme.** NatPN is fully differentiable using the closed-form Bayesian loss. Thus, we train the encoder $f_{\boldsymbol{\phi}}$, the parameter decoder $g_{\boldsymbol{\psi}}$ and the normalizing flow $\mathbb{P}(\boldsymbol{z}^{(i)} \mid \boldsymbol{\omega})$ w.r.t. parameters $\boldsymbol{\phi}, \boldsymbol{\psi}, \boldsymbol{\omega}$ jointly. Further, we observed that "warm-up training" (Ash & Adams, 2020) and "fine-tuning" (Käding et al., 2016) of the density helped to improve uncertainty estimation for more complex flows and datasets. Thus, we train the normalizing flow density to maximize the likelihood of the latent representations before and after the joint optimization while keeping all other parameters fixed.

## 3.6 Model Limitations

**Task-Specific OOD.** Previous works show that density estimation is unsuitable for acting on the raw image input (Choi et al., 2019; Nalisnick et al., 2019; 2020) or on a non-carefully transformed space (Lan & Dinh, 2020). To circumvent this issue, NatPN does not perform OOD detection directly on the input but rather fits a normalizing flow on a learned space. In particular, the latent space is **(1)** low-dimensional, **(2)** task-specific and **(3)** encodes meaningful semantic features. Similarly, Charpentier et al. (2020); Kirichenko et al. (2020); Morningstar et al. (2020); Winkens et al. (2020) already improved OOD detection of density-based methods by leveraging a task-induced bias or low-dimensional statistics. In the case of NatPN, the low-dimensional latent space has to contain relevant features to linearly predict the sufficient statistics required for the task. For example, NatPN aims at a linearly separable latent space for classification. The downside is that NatPN is capable of detecting OOD samples only with respect to the considered task and requires labeled examples during training. As an example, NatPN likely fails to detect a change of image color if the task aims at classifying object shapes and the latent space has no notion of color. Hence, we underline that NatPN comes with a task-dependent OOD definition, which is a reasonable choice in practice.

**Model-Task Mismatch.** Second, we emphasize that the uncertainty estimation quality of NatPN for (close to) ID data depends on the convergence of the model, the encoder architecture (e.g. MLP, Conv., DenseDepth (Eigen et al., 2014)) and the target distribution (e.g. Poisson, Normal distributions) choice which should match the task needs. However, we show *empirically* that NatPN provides high quality uncertainty estimates in practice on a wide range of tasks. Further, we show *theoretically* that

NatPN leads to uncertain prediction far away from training data for *any* exponential family target distributions. In comparison, Meinke & Hein (2020) showed akin guarantees for classification only.

## 4 EXPERIMENTS

In this section, we compare NatPN to existing methods on extensive experiments including three different tasks: classification, regression and count prediction. For each task type, we evaluate the prediction quality based on target error and uncertainty metrics. These various set-ups aim to highlight the versatility of NatPN. In particular, NatPN is the only model that adapts to all tasks and achieves high performances for all metrics without requiring multiple forward passes.

Table 2: Classification results on Sensorless Drive with Categorical target distribution. Best scores among all single-pass models are in bold. Best scores among all models are starred.

|  | Accuracy | Brier | 9/10 Alea. | 9/10 Epist. | OODom Alea. | OODom Epist. |
|---|---|---|---|---|---|---|
| Dropout | $98.62 \pm 0.11$ | $3.79 \pm 0.29$ | $30.20 \pm 0.85$ | $32.57 \pm 1.45$ | $27.03 \pm 0.51$ | $95.30 \pm 1.66$ |
| Ensemble | $98.83 \pm 0.17$ | $3.00 \pm 0.54$ | $30.79 \pm 0.74$ | $32.61 \pm 1.06$ | $27.16 \pm 0.59$ | $99.97 \pm 0.01$ |
| NatPE | $*99.66 \pm 0.03$ | $*0.68 \pm 0.05$ | $77.05 \pm 1.93$ | $83.73 \pm 1.89$ | $99.99 \pm 0.00$ | $*100.00 \pm 0.00$ |
| R-PriorNet | $98.85 \pm 0.25$ | $2.01 \pm 0.47$ | $40.13 \pm 2.99$ | $30.07 \pm 0.81$ | $*100.00 \pm 0.00$ | $23.59 \pm 0.00$ |
| EnD$^2$ | $93.95 \pm 2.35$ | $28.09 \pm 6.40$ | $26.35 \pm 0.60$ | $24.85 \pm 0.43$ | $84.43 \pm 15.21$ | $23.58 \pm 0.00$ |
| PostNet | $\mathbf{99.64 \pm 0.02}$ | $\mathbf{0.75 \pm 0.08}$ | $80.60 \pm 1.68$ | $*\mathbf{92.57 \pm 1.41}$ | $*100.00 \pm 0.00$ | $*100.00 \pm 0.00$ |
| NatPN | $99.61 \pm 0.05$ | $1.04 \pm 0.29$ | $*\mathbf{81.43 \pm 1.89}$ | $79.54 \pm 2.62$ | $99.98 \pm 0.00$ | $*100.00 \pm 0.00$ |

Table 3: Classification results on CIFAR-10 with Categorical target distribution. Best scores among all single-pass models are in bold. Best scores among all models are starred. Gray numbers indicate that R-PriorNet has seen samples from the SVHN dataset during training.

|  | Accuracy | Brier | SVHN Alea. | SVHN Epist. | CelebA Alea. | CelebA Epist. | OODom Alea. | OODom Epist. |
|---|---|---|---|---|---|---|---|---|
| Dropout | $88.15 \pm 0.20$ | $19.59 \pm 0.41$ | $80.63 \pm 1.59$ | $73.09 \pm 1.51$ | $71.84 \pm 4.28$ | $71.04 \pm 3.92$ | $18.42 \pm 1.11$ | $49.69 \pm 9.10$ |
| Ensemble | $*89.95 \pm 0.11$ | $17.33 \pm 0.17$ | $85.26 \pm 0.84$ | $82.51 \pm 0.63$ | $76.20 \pm 0.87$ | $74.23 \pm 0.78$ | $25.30 \pm 4.02$ | $89.21 \pm 7.55$ |
| NatPE | $89.21 \pm 0.09$ | $17.41 \pm 0.12$ | $85.66 \pm 0.34$ | $*83.16 \pm 0.67$ | $*78.95 \pm 1.15$ | $*82.06 \pm 1.30$ | $87.27 \pm 1.79$ | $*98.88 \pm 0.26$ |
| R-PriorNet | $\mathbf{88.94 \pm 0.23}$ | $*\mathbf{15.99 \pm 0.32}$ | $99.87 \pm 0.02$ | $99.94 \pm 0.01$ | $67.74 \pm 4.86$ | $59.55 \pm 7.90$ | $42.21 \pm 8.77$ | $38.25 \pm 9.82$ |
| EnD$^2$ | $84.03 \pm 0.25$ | $40.84 \pm 0.36$ | $*\mathbf{86.47 \pm 0.66}$ | $\mathbf{81.84 \pm 0.92}$ | $75.54 \pm 1.79$ | $75.94 \pm 1.82$ | $42.19 \pm 8.77$ | $15.79 \pm 0.27$ |
| PostNet | $87.95 \pm 0.20$ | $20.19 \pm 0.40$ | $82.35 \pm 0.68$ | $79.24 \pm 1.49$ | $72.96 \pm 2.33$ | $75.84 \pm 1.61$ | $85.89 \pm 4.10$ | $92.30 \pm 2.18$ |
| NatPN | $87.90 \pm 0.16$ | $19.99 \pm 0.46$ | $82.29 \pm 1.11$ | $77.83 \pm 1.22$ | $\mathbf{76.01 \pm 1.18}$ | $\mathbf{76.87 \pm 3.38}$ | $*\mathbf{93.67 \pm 3.03}$ | $\mathbf{94.90 \pm 3.09}$ |

Table 4: Results on the Bike Sharing Dataset with Normal $\mathcal{N}$ and Poison Poi target distributions. Best scores among all single-pass models are in bold. Best scores among all models are starred.

|  | RMSE | Calibration | Winter Epist. | Spring Epist. | Autumn Epist. | OODom Epist. |
|---|---|---|---|---|---|---|
| Dropout-$\mathcal{N}$ | $70.20 \pm 1.30$ | $6.05 \pm 0.77$ | $15.26 \pm 0.51$ | $13.66 \pm 0.16$ | $15.11 \pm 0.46$ | $99.99 \pm 0.01$ |
| Ensemble-$\mathcal{N}$ | $*48.02 \pm 2.78$ | $5.88 \pm 1.00$ | $42.46 \pm 2.29$ | $21.28 \pm 0.38$ | $21.97 \pm 0.58$ | $*100.00 \pm 0.00$ |
| EvReg-$\mathcal{N}$ | $\mathbf{49.58 \pm 1.51}$ | $3.77 \pm 0.81$ | $17.19 \pm 0.76$ | $15.54 \pm 0.65$ | $14.75 \pm 0.29$ | $34.99 \pm 17.02$ |
| NatPN-$\mathcal{N}$ | $49.85 \pm 1.38$ | $*\mathbf{1.95 \pm 0.34}$ | $*\mathbf{55.04 \pm 6.81}$ | $*\mathbf{23.25 \pm 1.20}$ | $*\mathbf{27.78 \pm 2.47}$ | $*\mathbf{100.00 \pm 0.00}$ |
| Dropout-Poi | $66.57 \pm 4.61$ | $55.00 \pm 0.22$ | $16.02 \pm 0.48$ | $13.48 \pm 0.38$ | $18.09 \pm 0.82$ | $*100.00 \pm 0.00$ |
| Ensemble-Poi | $*48.22 \pm 2.06$ | $55.31 \pm 0.21$ | $83.88 \pm 1.22$ | $34.21 \pm 1.81$ | $41.29 \pm 3.23$ | $*100.00 \pm 0.00$ |
| NatPN-Poi | $51.79 \pm 0.78$ | $*\mathbf{31.04 \pm 1.81}$ | $*\mathbf{85.15 \pm 3.61}$ | $*\mathbf{37.03 \pm 2.35}$ | $*\mathbf{42.73 \pm 4.38}$ | $*\mathbf{100.00 \pm 0.00}$ |

### 4.1 SETUP

In our experiments, we change the encoder architecture of NatPN to match the dataset needs. We perform a grid search over normalizing flows types (i.e. radial flows (Rezende & Mohamed, 2015) and MAF (Germain et al., 2015; Papamakarios et al., 2017)) and latent dimensions. We show further experiments on architecture, latent dimension, normalizing flow choices and certainty budget choice in the appendix. Furthermore, we use approximations of the log-Gamma $\log \Gamma(x)$ and the Digamma $\psi(x)$ functions for large input values to avoid unstable floating computations (see app.). As prior parameters, we set $\chi^{\mathrm{prior}} = \mathbf{1}_C / C, n^{\mathrm{prior}} = C$ for classification, $\chi^{\mathrm{prior}} = (0, 100)^T, n^{\mathrm{prior}} = 1$ for regression and $\chi^{\mathrm{prior}} = 1, n^{\mathrm{prior}} = 1$ for count prediction enforcing high entropy for prior distributions.

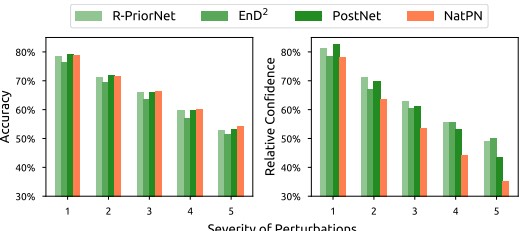

Figure 3: Averaged accuracy and confidence under 15 dataset shifts on CIFAR-10 (Hendrycks & Dietterich, 2019). On more severe perturbations (i.e. data further away from data distribution), NatPN maintains a competitive accuracy while assigning higher epistemic uncertainty as desired. Baselines provide a slower relative confidence decay.

Table 5: Regression results on models trained on different UCI datasets with Normal target distribution. The upper half displays models trained on Kin8nm, the lower half shows models trained on Concrete Compressive Strength.

|  | RMSE | Calibration | Energy Alea. | Energy Epist. | Concrete Alea. | Concrete Epist. | Kin8nm Alea. | Kin8nm Epist. |
|---|---|---|---|---|---|---|---|---|
| **Dropout** | $0.09 \pm 0.00$ | $3.13 \pm 0.43$ | $90.18 \pm 6.00$ | $99.94 \pm 0.06$ | $*100.00 \pm 0.00$ | $*100.00 \pm 0.00$ | | |
| **Ensemble** | $*0.07 \pm 0.00$ | $2.69 \pm 0.49$ | $*100.00 \pm 0.00$ | $*100.00 \pm 0.00$ | $*100.00 \pm 0.00$ | $*100.00 \pm 0.00$ | \multicolumn{2}{c}{**in-distribution**} | |
| **NatPE** | $0.08 \pm 0.00$ | $5.49 \pm 0.30$ | $*100.00 \pm 0.00$ | $*100.00 \pm 0.00$ | $*100.00 \pm 0.00$ | $*100.00 \pm 0.00$ | | |
| **EvReg** | $0.09 \pm 0.00$ | $3.74 \pm 0.53$ | $88.06 \pm 11.94$ | $88.06 \pm 11.94$ | $\mathbf{*100.00 \pm 0.00}$ | $86.84 \pm 13.16$ | \multicolumn{2}{c}{**in-distribution**} | |
| **NatPN** | $\mathbf{0.08 \pm 0.00}$ | $\mathbf{*2.04 \pm 0.45}$ | $\mathbf{*100.00 \pm 0.00}$ | $\mathbf{*100.00 \pm 0.00}$ | $\mathbf{*100.00 \pm 0.00}$ | $\mathbf{*100.00 \pm 0.00}$ | | |
| **Dropout** | $5.67 \pm 0.07$ | $*3.03 \pm 0.40$ | $9.33 \pm 0.36$ | $93.53 \pm 2.41$ | | | $1.09 \pm 0.13$ | $64.30 \pm 7.14$ |
| **Ensemble** | $5.69 \pm 0.20$ | $3.81 \pm 0.67$ | $54.19 \pm 18.93$ | $*100.00 \pm 0.00$ | \multicolumn{2}{c}{**in-distribution**} | | $72.57 \pm 19.32$ | $*100.00 \pm 0.00$ |
| **NatPE** | $*4.78 \pm 0.20$ | $5.58 \pm 1.27$ | $*100.00 \pm 0.00$ | $*100.00 \pm 0.00$ | | | $*100.00 \pm 0.00$ | $*100.00 \pm 0.00$ |
| **EvReg** | $6.04 \pm 0.18$ | $7.36 \pm 1.04$ | $8.93 \pm 0.02$ | $51.39 \pm 18.56$ | \multicolumn{2}{c}{**in-distribution**} | | $0.93 \pm 0.00$ | $34.44 \pm 20.95$ |
| **NatPN** | $\mathbf{5.83 \pm 0.23}$ | $\mathbf{5.41 \pm 1.33}$ | $\mathbf{*100.00 \pm 0.00}$ | $\mathbf{*100.00 \pm 0.00}$ | | | $\mathbf{*100.00 \pm 0.00}$ | $\mathbf{*100.00 \pm 0.00}$ |

Table 6: Regression results on NYU Depth v2 with Normal target distribution. RMSE is in cm. OOD scores on LSUN are reported on the held-out classes 'classrooms' (left) and 'churches' (right).

|  | RMSE | Calibration | LSUN Alea. | LSUN Epist. | KITTI Alea. | KITTI Epist. | OODom Alea. | OODom Epist. |
|---|---|---|---|---|---|---|---|---|
| **Dropout** | $46.95$ | $4.03$ | $*95.29 / 97.74$ | $83.89 / 83.22$ | $98.07$ | $84.90$ | $74.40$ | $*100.00$ |
| **EvReg** | $*28.88$ | $*1.05$ | $58.70 / 56.71$ | $70.19 / 64.02$ | $56.60$ | $62.67$ | $75.43$ | $56.39$ |
| **NatPN** | $29.72$ | $1.14$ | $\mathbf{94.13 / *98.67}$ | $\mathbf{*89.08 / *90.56}$ | $\mathbf{*98.93}$ | $\mathbf{*93.15}$ | $\mathbf{*100.00}$ | $\mathbf{*100.00}$ |

**Baselines.** We focus on recent models parametrizing prior distributions over the target distribution. For classification, we compare NatPN to Reverse KL divergence Prior Networks (R-PriorNet) (Malinin & Gales, 2019), Ensemble Distribution Distillation (EnD$^2$) (Malinin et al., 2020b) and Posterior Networks (PostNet) (Charpentier et al., 2020). Note that Prior Networks require OOD training data — we use an auxiliary dataset when available and Gaussian noise otherwise. For regression, we compare to Evidential Regression (EvReg) (Amini et al., 2020). Beyond baselines parametrizing conjugate prior distributions, we also compare to dropout (Dropout) (Gal & Ghahramani, 2016) and ensemble (Ensemble) (Lakshminarayanan et al., 2017) models for all tasks. These sampling baselines require multiple forward passes for uncertainty estimation. Further details are given in the appendix.

**Datasets.** We split all datasets into train, validation and test sets. For classification, we use one tabular dataset (Sensorless Drive (Dua & Graff, 2017)) and three image datasets (MNIST (LeCun et al., 2010), FMNIST (Xiao et al., 2017) and CIFAR-10 (Krizhevsky et al., 2009)). For count prediction, we use the Bike Sharing dataset (Fanaee-T & Gama, 2014) to predict the number of bike rentals within an hour. For regression, we also use the Bike Sharing dataset where the target is viewed as continuous, real-world UCI datasets used in Amini et al. (2020); Hernandez-Lobato & Adams (2015) and the image NYU Depth v2 dataset (Nathan Silberman & Fergus, 2012) where the goal is to predict the image depth per pixel. All inputs are rescaled with zero mean and unit variance. We also scale the output target for regression. Further details are given in the appendix.

**Metrics.** Beyond the target prediction error, we evaluate model uncertainty estimation using calibration and OOD detection scores. Furthermore, we report the inference speed. Further results including histograms with uncertainty estimates or latent space visualization are presented in appendix. *Target error:* We use the accuracy (Accuracy) for classification and the Root Mean Squared Error (RMSE) for regression and count prediction. *Calibration:* For classification, we use the Brier score (Brier) (Gneiting & Raftery, 2007). For regression and count prediction, we use the quantile calibration score (Calibration) (Kuleshov et al., 2018). *OOD detection:* We evaluate how the uncertainty scores enable to detect OOD data using the area under the precision-recall curve (AUC-PR) the area under the receiver operating characteristic curve (AUC-ROC) in the appendix. We use two different uncertainty measures: the negative entropy of the predicted target distribution accounting for the aleatoric uncertainty (Alea. OOD) and the predicted evidence or variance of the predicted mean (Epist. OOD). Similarly to (Charpentier et al., 2020; Ovadia et al., 2019), we use four different types of clear OOD samples: Unseen datasets (KMNIST Clanuwat et al. (2018), Fashion-MNIST (Xiao et al., 2017), SVHN (Netzer et al., 2011), LSUN (Yu et al., 2015), CelebA (Liu et al., 2015), KITTI (Geiger et al., 2013)), left-out data (classes 9/10 for Sensorless Drive, winter/spring/autumn seasons for Bike Sharing), out-of-domain data not normalized in $[0, 1]$ (OODom) and dataset shifts (corrupted CIFAR-10 (Hendrycks & Dietterich, 2019)). Further details are given in the appendix.

## 4.2 RESULTS

**Classification.** We show results for the tabular dataset Sensorless Drive with unbounded input domain in Tab. 2, and the image datasets MNIST, FMNIST and CIFAR-10 with bounded input domain in

Tab. 3 and appendix. Overall, for classification NatPN performs on par with best single-pass baselines (i.e. 12/30 top-1 scores, 25/30 top-2 scores) and NatPE performs the best among multiple-pass models (i.e. 28/30 top-1 scores). A single NatPN achieves accuracy and calibration performance on par with the most calibrated baselines, namely PostNet and R-PriorNet which requires one normalizing flow per class or training OOD data. Further, NatPE consistently improves accuracy and calibration performance of a single NatPN which underlines the benefit of aggregating multiple models predictions for accuracy and calibration (Lakshminarayanan et al., 2017). Without requiring OOD data during training, both NatPN and NatPE achieve excellent OOD detection scores w.r.t. all OOD types. This strongly suggests that NatPN does *not* suffer from the flaws in Choi et al. (2019); Nalisnick et al. (2019; 2020). In particular, NatPN and NatPE achieve almost perfect OODom scores contrary to all other baselines except PostNet. This observation aligns well with the theoretical guarantee of NatPN far from training data (see Thm. 1) which also applies to each NatPE member. The similar performance of NatPN and PostNet for classification is intuitively explained by their akin design: both models perform density estimation on a low-dimensional latent space. Similarly to Charpentier et al. (2020), we compute the average confidence avg-conf $= \frac{1}{N} \sum_i^N n^{(i)} = \frac{1}{N} \sum_i^N \alpha_0^{(i)}$ and then compare the average confidence change. The average confidence change is computed by taking the ratio of the average confidence of $N$ corrupted data at severity $s$ and the average confidence of $N$ clean data (i.e. the corrupted data at severity 0) i.e. $\frac{\text{avg-conf}_s}{\text{avg-conf}_0}$ for $s \in [\![1, 5]\!]$. NatPN maintains a competitive accuracy (Fig. 3, left) while assigning higher epistemic uncertainty as desired (Fig. 3, right). Baselines provide a slower relative confidence decay.

**Regression & Count Prediction.** We show the results for the Bike Sharing, the tabular UCI datasets and the image NYU Depth v2 datasets in Tab. 4, 5, 6. For the large NYU dataset, we compare against all baselines which require only a single model to be trained. Overall, NatPN outperforms other single-pass models for 23/26 scores for regression, thus significantly improving calibration and OOD detection scores. Further, NatPN shows a strong improvement for calibration and OOD detection for count prediction with Poisson distributions among all models. Interestingly, all the models are less calibrated on the Bike Sharing dataset using a target Poisson distribution rather than a target Normal distribution. This suggests a mismatch of the Poisson distribution for this particular task. The almost perfect OODom scores of NatPN validate again Thm. 1 which also holds for regression.

**Inference Speed.** We show the average inference time per batch for all models on CIFAR-10 for classification and the NYU Depth v2 dataset for regression in Tab. 7. NatPN shows a significant improvement over Dropout and Ensemble which are both approximately five times slower since they require five forward passes for prediction. Notably, the NatPN speedup does not deteriorate target error and uncertainty scores. NatPN is slightly slower than R-PriorNet, EnD$^2$ and EvReg as they do not evaluate an additional normalizing flow. However, NatPN – which uses a single normalizing flow – is faster than

Table 7: Batched Inference Time (in ms), NVIDIA GTX 1080 Ti

|  | CIFAR-10 (batch size 4,096) | NYU Depth v2 (batch size 4) |
|---|---|---|
| **Dropout** | $407.91 \pm 5.65$ | $650.96 \pm 0.22$ |
| **Ensemble** | $361.61 \pm 5.41$ | $649.78 \pm 0.18$ |
| **R-PriorNet** | $61.83 \pm 2.57$ | − |
| **EnD$^2$** | $61.83 \pm 2.57$ | − |
| **PostNet** | $88.56 \pm 0.06$ | − |
| **EvReg** | − | $129.88 \pm 0.75$ |
| **NatPN** | $75.64 \pm 0.04$ | $137.13 \pm 0.18$ |
| **NatPE** | $370.17 \pm 0.09$ | $676.74 \pm 0.38$ |

PostNet – which scales linearly w.r.t. the number of classes since it evaluates one normalizing flow per class. Lastly, NatPN is the only single-pass model that can be used for *both* tasks.

## 5 CONCLUSION

We introduce Natural Posterior Network which belongs to the family of models parametrizing conjugate prior distributions. NatPN is capable of efficient uncertainty estimation for *any* task where the target distribution is in the exponential family (incl. classification, regression and count prediction). NatPN relies on the Bayes formula and the general form of exponential family distributions to perform a closed-form input-dependent posterior update over the target distribution. Further, an ensemble of NatPNs has a principled Bayesian combination interpretation. Theoretically, NatPN guarantees high uncertainty far from training data. Experimentally, NatPN achieves fast, versatile and high-quality uncertainty predictions with strong performance in calibration and OOD detection.

## 6 ETHICS STATEMENT

Accurate uncertainty estimation aims at improving trust in safety-critical domains subject to automation, and in a maintenance context where the underlying data distribution might slowly shift over time. In this regard, NatPN significantly improves the applicability of uncertainty estimation across a wide range of prediction domains (e.g. classification, regression, count prediction, etc) while maintaining a fast inference time. This could be particularly beneficial in industrial applications with time pressure and potential critical consequences (e.g. finance, medicine, policy decision making, etc).

Nonetheless, while NatPN achieves high-quality uncertainty estimation, there is always a risk that NatPN does not fully capture the real-world complexity e.g. for OOD data close to ID data. Furthermore, we raise awareness about two other risks of excessive trust related to the *Dunning-Kruger effect* (Kruger & Dunning, 2000): human excessive trust in Machine Learning model capacity, and human excessive trust in its own interpretation capacity. Therefore, we encourage practitioners to proactively confront the model design and its uncertainty estimates to desired behaviors in real-world use cases.

## 7 REPRODUCIBILITY STATEMENT

We provide all datasets and the model code at the following project page [1]. In App. F, we give a detailed description for each dataset used in this paper. This description includes the task description, the dataset size, the input/output dimensions, the input/output preprossessing and the train/validation/test splits used in the experiments. In App. G, we give a detailed description for the architecture and grid search performed for each model used in this paper. Specifically, we provide an hyper-parameter study for NatPN in App. I.6, In App. H, we give a detailed description of the metrics used in the experiments. Finally, we provide a complete proof of Thm. 1, a detailed description of the Bayesian loss and relevant formulae for exponential family distributions in App. A, App. B and App. C.

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

## A  THEOREM 1

We prove theorem 1 based on Lemmas 2 and 3. Lemma 2 states that the input space can be divided in a finite number of linear regions (Arora et al., 2018). Lemma 3 states that a probability density with bounded derivatives has to converge to 0 at infinity (Dix, 2013). We additionally recall Lemma 4 which provide a similar convergence guarantee without the bounded derivative constraint (Niculescu & Popovici, 2011). Finally, Lemma 5 particularly shows that the guarantee of theorem 1 can be obtained with Gaussian Mixtures which are commonly used for density estimation or radial flows which are used in the experiments.

**Lemma 2.** *(Arora et al., 2018) Let $\{Q_l\}_l^R$ be the set of linear regions associated to the piecewise ReLU network $f_\phi(\boldsymbol{x})$. For any $\boldsymbol{x} \in \mathbb{R}^D$, there exists $\delta^* \in \mathbb{R}^+$ and $l^* \in 1, ..., R$ such that $\delta\boldsymbol{x} \in Q_{l^*}$ for all $\delta > \delta^*$.*

**Lemma 3.** *(Dix, 2013) Let $p \in L^1(0, \infty)$ with bounded first derivative $p'$, then $p(\delta) \underset{\delta \to \infty}{\to} 0$. This convergence is stronger than in Lem. 4 as the limit is not in density but with standard limit notation.*

**Lemma 4.** *(Niculescu & Popovici, 2011) Let $p \in L^1(0, \infty)$, then $p(\delta) \underset{\delta \to \infty}{\to} 0$ in density. This means that the sets where $p(t)$ is far from its 0 limit (i.e. $\{t \geq 0 : |p(t)| \geq \epsilon\}$ with $\epsilon > 0$) has zero density.*

**Lemma 5.** *Let $\mathbb{P}(\boldsymbol{z} \mid \boldsymbol{\omega})$ be parametrized with a Gaussian Mixture Model (GMM) or a radial flow, then $\mathbb{P}(\boldsymbol{z} \mid \boldsymbol{\omega}) \underset{||\boldsymbol{z}|| \to \infty}{\to} 0$.*

*Proof.* We prove now lem. 5 for GMM and radial flow. The proof is straightforward for the GMM parametrization since every Gaussian component of the mixture has 0 limit when $||\boldsymbol{z}|| \to \infty$.

Let denote now $p_1(z) = \mathbb{P}(\boldsymbol{z} \mid \boldsymbol{\omega})$ be parametrized with a radial flow transformation $g(\boldsymbol{z})$ and a base unit Gaussian distribution $p_0$ i.e.:

$$p_1(\boldsymbol{z}) = p_0(g(\boldsymbol{z})) \times |\det\frac{\partial g(\boldsymbol{z})}{\partial \boldsymbol{z}}|$$

Further, we can express the transformation $g(\boldsymbol{z})$ and its determinant $\det\frac{\partial g(\boldsymbol{z})}{\partial \boldsymbol{z}}$ as follows:

$$g(\boldsymbol{z}) = \boldsymbol{z} + \beta h(\alpha, r)(\boldsymbol{z} - \boldsymbol{z}_0)$$

$$\det\frac{\partial g(\boldsymbol{z})}{\partial \boldsymbol{z}} = \frac{1 + \beta h(\alpha, r) + \beta h'(\alpha, r)r}{(1 + \beta h(\alpha, r))^{H-1}}$$

where $h(\alpha, r) = \frac{1}{\alpha+r}$ and $r = ||\boldsymbol{z} - \boldsymbol{z}_0||$. On one hand, we have $||g(\boldsymbol{z})|| \to +\infty$ when $||\boldsymbol{z}|| \to \infty$ since $||\beta h(\alpha, r)(\boldsymbol{z} - \boldsymbol{z}_0)|| < \beta$. Thus, the base Gaussian density $p_0(g(\boldsymbol{z})) \to 0$ when $||\boldsymbol{z}|| \to \infty$. On the other hand, we have $|\det\frac{\partial g(\boldsymbol{z})}{\partial \boldsymbol{z}}| \to 1$ since $\beta h(\alpha, r) \to 0$ and $\beta h'(\alpha, r)r \to 0$ when $||\boldsymbol{z}|| \to \infty$. Therefore, the transformed density $p_0(g(\boldsymbol{z})) \times |\det\frac{\partial g(\boldsymbol{z})}{\partial \boldsymbol{z}}| \to 0$ when $||\boldsymbol{z}|| \to \infty$ which ends the proof. Note that this proof can be extended to stacked radial flows by induction. $\square$

**Theorem.** *Let a NatPN model parametrized with a (deep) encoder $f_\phi$ with piecewise ReLU activations, a decoder $g_\psi$ and the density $\mathbb{P}(\boldsymbol{z} \mid \boldsymbol{\omega})$. Let $f_\phi(\boldsymbol{x}) = V^{(l)}\boldsymbol{x} + a^{(l)}$ be the piecewise affine representation of the ReLU network $f_\phi$ on the finite number of affine regions $Q^{(l)}$ (Arora et al., 2018). Suppose that $V^{(l)}$ have independent rows and the density function $\mathbb{P}(\boldsymbol{z} \mid \boldsymbol{\omega})$ has bounded derivatives, then for almost any $\boldsymbol{x}$ we have $\mathbb{P}(f_\phi(\delta \cdot \boldsymbol{x}) \mid \boldsymbol{\omega}) \underset{\delta \to \infty}{\to} 0$. i.e the evidence becomes small far from training data.*

*Proof.* We prove now thm. 1. Let $\boldsymbol{x} \in \mathbb{R}^D$ be a non-zero input and $f_\phi$ be a ReLU network. Lem. 2 implies that there exists $\delta^* \in \mathbb{R}^+$ and $l \in \{1, ..., R\}$ such that $\delta \cdot \boldsymbol{x} \in Q^{(l)}$ for all $\delta > \delta^*$. Thus, $\boldsymbol{z}_\delta = f_\phi(\delta \cdot \boldsymbol{x}) = \delta \cdot (V^{(l)}\boldsymbol{x}) + a^{(l)}$ for all $\delta > \delta^*$. Note that for $\delta \in [\delta^*, +\infty]$, $\boldsymbol{z}_\delta$ follows an affine half line $S_{\boldsymbol{x}} = \{\boldsymbol{z} \mid \boldsymbol{z} = \delta \cdot (V^{(l)}\boldsymbol{x}) + a^{(l)}, \delta > \delta^*\}$ in the latent space. Further, note that $V^{(l)}\boldsymbol{x} \neq 0$ and $||\boldsymbol{z}_\delta|| \underset{\delta \to \infty}{\to} +\infty$ since $\boldsymbol{x} \neq 0$ and $V^{(l)}$ has independent rows.

We now define the function $p(\delta) = \mathbb{P}(\boldsymbol{z}_\delta \mid \boldsymbol{\omega})$ which is the density function $\mathbb{P}(\boldsymbol{z} \mid \boldsymbol{\omega})$ restricted on the affine half line $S_{\boldsymbol{x}}$. Since $\mathbb{P}(\boldsymbol{z} \mid \boldsymbol{\omega})$ is a normalized probability density, then the function $\delta \mapsto p(\delta - \delta^*)$

is integrable on $[0, +\infty]$. Indeed we have:

$$\int_0^{+\infty} p(\delta - \delta^*)d\delta = \int_{\delta^*}^{+\infty} p(\delta)d\delta$$

$$= \int_{\delta^*}^{+\infty} \mathbb{P}(\delta \cdot (V^{(l)}\boldsymbol{x}) + a^{(l)} \,|\, \boldsymbol{\omega})d\delta$$

$$= \int^{S_{\boldsymbol{x}}} \mathbb{P}(\boldsymbol{z} \,|\, \boldsymbol{\omega})d\boldsymbol{z} < +\infty$$

Further since the function $\mathbb{P}(\boldsymbol{z} \,|\, \boldsymbol{\omega})$ has bounded derivatives, we can apply Lem. 3 to the function $\delta \mapsto p(\delta - \delta^*)$ to get the expected result i.e.

$$\mathbb{P}(f_\phi(\delta \cdot \boldsymbol{x}) \,|\, \boldsymbol{\omega}) = p(\delta) = p((\delta + \delta^*) - \delta^*) \underset{\delta \to \infty}{\to} 0$$

which ends the proof.

Alternatively a slightly weaker conclusion also holds if the density function does not have bounded derivatives using lem. 4 (instead of lem. 3) with the notion of limit in density. The stronger conclusion is valid if we parametrize $\mathbb{P}(\boldsymbol{z} \,|\, \boldsymbol{\omega})$ with a Gaussian Mixture Model or a radial flow density according to lem. 5 since $||\boldsymbol{z}_\delta|| \underset{\delta \to \infty}{\to} +\infty$. □

Further, we provide additional comments on the assumption that a trained network converges to linear transformation with exactly two or more dependent rows in Thm. 1. Under this realistic condition Hein et al. (2019), the null space is reduced to 0 according to the rank-nullity theorem meaning that there should be no dead input feature/pixel. If this condition does not hold, this would mean that this specific input feature/pixel is not informative for the prediction task. Thus it could be desired in practice that it does not affect the uncertainty on the prediction. This latter aspect is discussed in the "Task-Specific OOD" paragraph in Sec. 3.6.

## B  BAYESIAN LOSS

NatPN minimizes the following Bayesian formulation:

$$\mathcal{L}^{(i)} = -\underbrace{\mathbb{E}_{\boldsymbol{\theta}^{(i)} \sim \mathbb{Q}^{\mathrm{post},(i)}}[\log \mathbb{P}(y^{(i)} \,|\, \boldsymbol{\theta}^{(i)})]}_{(i)} - \underbrace{\mathbb{H}[\mathbb{Q}^{\mathrm{post},(i)}]}_{(ii)} \tag{7}$$

where $\mathbb{H}[\mathbb{Q}^{\mathrm{post},(i)}]$ denotes the entropy of the predicted posterior distribution $\mathbb{Q}^{\mathrm{post},(i)}$. This loss is generally not equal to the ELBO loss. While the term **(i)** can be viewed as an ELBO loss without KL regularization, the term **(ii)** is not necessarily equal to the prior KL regularization term in the ELBO loss since a proper uniform prior might not exist (e.g. the target $y$ is a real number). Indeed, if the target $y$ is a real number, there exists no uniform prior on $\boldsymbol{\theta}$ and the Bayesian loss and ELBO loss are different i.e. $KL(\mathbb{Q} \,||\, \mathbb{Q}^{\mathrm{prior}}) = \int \mathbb{Q}(\theta) \log(\frac{\mathbb{Q}(\theta)}{\mathbb{Q}^{\mathrm{prior}}(\theta)})d\theta \neq \int \mathbb{Q}(\theta) \log \mathbb{Q}(\theta)d\theta = \mathbb{H}(\mathbb{Q})$. Nonetheless, when a uniform prior $\mathbb{Q}^{\mathrm{unif}}$ exists (e.g. the target $y$ is a class), the loss optimization can be seen as an amortized variational optimization of an ELBO loss (Zhang et al., 2018) i.e. $\mathcal{L}^{(i)} = -\mathbb{E}_{\mathbb{Q}^{\mathrm{post},(i)}}[\log \mathbb{P}(y^{(i)} \,|\, \boldsymbol{\theta}^{(i)})] + \mathrm{KL}[\mathbb{Q}^{\mathrm{post},(i)} \,||\, \mathbb{Q}^{\mathrm{unif}}]$ where the predicted distribution $\mathbb{Q}^{\mathrm{post},(i)}$ is the variational distribution — which approximates the true posterior distribution. Indeed, the KL regularization term is equal to the entropy regularization term i.e. $KL(\mathbb{Q} \,||\, \mathbb{Q}^{\mathrm{unif}}) = \int \mathbb{Q}(\theta) \log(\frac{\mathbb{Q}(\theta)}{\mathbb{Q}^{\mathrm{unif}}(\theta)}d\theta \propto \int \mathbb{Q}(\theta) \log Q(\theta)d\theta = \mathbb{H}(\mathbb{Q})$. Hence, the loss name "Bayesian loss" (Charpentier et al., 2020) is motivated by its difference with the ELBO loss and its Bayesian property at optimum.

## C  FORMULAE FOR EXPONENTIAL FAMILY DISTRIBUTIONS

### C.1  GENERAL CASE

**Target Distribution.** An exponential family distribution on a target variable $y \in \mathbb{R}$ with natural parameters $\boldsymbol{\theta} \in \mathbb{R}^L$ can be denoted as

$$\mathbb{P}(y \,|\, \boldsymbol{\theta}) = h(y) \exp\left(\boldsymbol{\theta}^T \boldsymbol{u}(y) - A(\boldsymbol{\theta})\right) \tag{8}$$

where $h : \mathbb{R} \to \mathbb{R}$ is the carrier measure, $A : \mathbb{R}^L \to \mathbb{R}$ the log-normalizer and $\boldsymbol{u} : \mathbb{R} \to \mathbb{R}^L$ the sufficient statistics.

**Conjugate Prior Distribution.** An exponential family distribution $\mathbb{P}$ always admits a conjugate prior:

$$\mathbb{Q}(\boldsymbol{\theta} \,|\, \boldsymbol{\chi}, n) = \eta(\boldsymbol{\chi}, n) \exp\left(n \,\boldsymbol{\theta}^T \boldsymbol{\chi} - n A(\boldsymbol{\theta})\right) \tag{9}$$

where $\eta : \mathbb{R}^L \times \mathbb{R} \to \mathbb{R}$ is a normalization factor and $A$ the log-normalizer of the distribution $\mathbb{P}$ as in Eq. 8).

**Posterior Predictive Distribution.** The posterior predictive distribution is given as $\int \mathbb{P}(y^{(i)}|\boldsymbol{\theta}) \,\mathbb{Q}(\boldsymbol{\theta}|\boldsymbol{\chi}^{\text{post},(i)}, n^{\text{post},(i)}) d\boldsymbol{\theta}$ where the parameter $\boldsymbol{\theta}$ is marginalized out (Gal, 2016). This distribution can always be computed in closed form for exponential family distributions:

$$\mathbb{P}(y \,|\, \boldsymbol{\chi}, n) = h(y) \frac{\eta(\boldsymbol{\chi}, n)}{\eta\left(\frac{n\boldsymbol{\chi} + \boldsymbol{u}(y)}{n+1}, n+1\right)} \tag{10}$$

where $h$ is the carrier measure defined in Eq. 8 and $\eta$ is the normalization factor defined in Eq. 9. In particular, the posterior predictive distributions for Categorical, Normal and Poisson target distributions are Categorical, Student and Negative Binomial distributions, respectively.

**Likelihood.** The log-likelihood of an exponential family distribution can be written as follows:

$$\log \mathbb{P}(y^{(i)} \,|\, \boldsymbol{\theta}) = \log h(y^{(i)}) + \boldsymbol{\theta}^T \boldsymbol{u}(y^{(i)}) - A(\boldsymbol{\theta}) \tag{11}$$

**Expected Log-Likelihood.** Given the log-likelihood of an exponential family distribution, its expectation under the conjugate prior distribution $\mathbb{Q}(\boldsymbol{\theta}|\boldsymbol{\chi}, n)$ can be written as

$$\mathbb{E}_{\boldsymbol{\theta} \sim \mathbb{Q}(\boldsymbol{\chi}, n)}[\log \mathbb{P}(y^{(i)} \,|\, \boldsymbol{\theta})] = \log h(y^{(i)}) + \mathbb{E}_{\boldsymbol{\theta} \sim \mathbb{Q}(\boldsymbol{\chi}, n)}[\boldsymbol{\theta}]^T \boldsymbol{u}(y^{(i)}) - \mathbb{E}_{\boldsymbol{\theta} \sim \mathbb{Q}(\boldsymbol{\chi}, n)}[A(\boldsymbol{\theta})] \tag{12}$$

where $\mathbb{E}_{\mathbb{Q}(\boldsymbol{\theta}|\boldsymbol{\chi}, n)}[\boldsymbol{\theta}] = \boldsymbol{\chi}$ (Brown, 1986; Diaconis & Ylvisaker, 1979).

**Entropy.** The entropy of a random variable $y \sim \mathbb{P}(y|\boldsymbol{\theta})$ for an exponential family distribution $\mathbb{P}$ can be written as follows (Nielsen & Nock, 2010):

$$\mathbb{H}[\mathbb{P}(y|\boldsymbol{\theta})] = A(\boldsymbol{\theta}) - \boldsymbol{\theta}^T \nabla_{\boldsymbol{\theta}} A(\boldsymbol{\theta}) - \mathbb{E}_{y \sim \mathbb{P}(\boldsymbol{\theta})}[\log h(y)] \tag{13}$$

## C.2 CATEGORICAL & DIRICHLET DISTRIBUTIONS

The Dirichlet distribution $\boldsymbol{p} \sim \text{Dir}(\boldsymbol{\alpha})$ is the conjugate prior of the categorical distributions $\boldsymbol{y} \sim \text{Cat}(\boldsymbol{p})$.

**Target Distribution.** The density and the entropy of the categorical distribution are:

$$\text{Cat}(y \,|\, \boldsymbol{p}) = \sum_{i=1}^{K} \mathbb{I}[y_i = 1] \, p_i \tag{14}$$

$$\mathbb{H}[\text{Cat}(\boldsymbol{p})] = \sum_{c=1}^{C} \log p_c \tag{15}$$

**Conjugate Prior Distribution.** The density and the entropy of the Dirichlet distribution are:

$$\text{Dir}(\boldsymbol{p} \,|\, \boldsymbol{\alpha}) = \frac{\Gamma\left(\sum_{c=1}^{C} \alpha_c\right)}{\prod_{c=1}^{K} \Gamma(\alpha_c)} \prod_{c=1}^{C} p_c^{\alpha_c - 1} \tag{16}$$

$$\mathbb{H}[\text{Dir}(\boldsymbol{\alpha})] = \log B(\boldsymbol{\alpha}) + (\alpha_0 - C)\psi(\alpha_0) - \sum_c (\alpha_c - 1)\psi(\alpha_c) \tag{17}$$

where $\psi(\alpha)$ and $B(\boldsymbol{\alpha})$ denote Digamma and Beta functions, respectively, and $\alpha_0 = \sum_c \alpha_c$.

**Expected Log-Likelihood.** The expected likelihood of the categorical distribution $\text{Cat}(\boldsymbol{p})$ under the Dirichlet distribution $\text{Dir}(\boldsymbol{\alpha})$ is

$$\mathbb{E}_{\boldsymbol{p} \sim \text{Dir}(\boldsymbol{\alpha})}[\log \text{Cat}(y \,|\, \boldsymbol{p})] = \psi(\alpha_y) - \psi(\alpha_0) \tag{18}$$

where $\psi(\alpha)$ denotes Digamma function.

## C.3 Normal & Normal-Inverse-Gamma Distributions

The Normal-Inverse-Gamma (NIG) distribution $\mu, \sigma \sim \mathcal{N}\Gamma^{-1}(\mu_0, \lambda, \alpha, \beta)$ is the conjugate prior of the normal distribution $y \sim \mathcal{N}(\mu, \sigma)$. Note that as both parameters $\lambda$ and $\alpha$ can be viewed as pseudo-counts. However, the natural prior parametrization enforces a single pseudo-count $n$ corresponding to $\lambda = 2\alpha$.

**Target Distribution.** The density and the entropy of the Normal distribution are:

$$\mathcal{N}(y \mid \mu, \sigma) = \frac{1}{\sigma\sqrt{2\pi}} \exp\left(-\frac{(x-\mu)^2}{2\sigma^2}\right) \tag{19}$$

$$\mathbb{H}[\mathcal{N}(\mu, \sigma)] = \frac{1}{2}\log(2\pi\sigma^2) \tag{20}$$

**Conjugate Prior Distribution.** The density and the entropy of the NIG distribution are:

$$\mathcal{N}\Gamma^{-1}(\mu, \sigma \mid \mu_0, \lambda, \alpha, \beta) = \frac{\beta^\alpha\sqrt{\lambda}}{\Gamma(\alpha)\sqrt{2\pi\sigma^2}}\left(\frac{1}{\sigma^2}\right)^{\alpha+1}\exp\left(-\frac{2\beta + \lambda(\mu-\mu_0)^2}{2\sigma^2}\right) \tag{21}$$

$$\mathbb{H}[\mathcal{N}\Gamma^{-1}(\mu_0, \lambda, \alpha, \beta)] = \frac{1}{2} + \log\left((2\pi)^{\frac{1}{2}}\beta^{\frac{3}{2}}\Gamma(\alpha)\right) - \frac{1}{2}\log\lambda + \alpha - \left(\alpha + \frac{3}{2}\right)\psi(\alpha) \tag{22}$$

where $\Gamma(\alpha)$ denotes the Gamma function.

**Expected Log-Likelihood.** The expected likelihood of the Normal distribution $\mathcal{N}(\mu, \sigma)$ under the NIG distribution $\mathcal{N}\Gamma^{-1}(\mu_0, \lambda, \alpha, \beta)$ is:

$$\mathbb{E}_{(\mu,\sigma)\sim\mathcal{N}\Gamma^{-1}(\mu,\lambda,\alpha,\beta)}[\log\mathcal{N}(y\mid\mu,\sigma)] \tag{23}$$

$$= \mathbb{E}\left[-\frac{(y-\mu)^2}{2\sigma^2} - \log(\sigma\sqrt{2\pi})\right] \tag{24}$$

$$= \frac{1}{2}\left(-\mathbb{E}\left[\frac{(y-\mu_0)^2}{2\sigma^2}\right] - \mathbb{E}\left[\log\sigma^2\right] - \log 2\pi\right) \tag{25}$$

$$= \frac{1}{2}\left(-y^2\,\mathbb{E}\left[\frac{1}{\sigma^2}\right] + 2y\,\mathbb{E}\left[\frac{\mu}{\sigma^2}\right] - \mathbb{E}\left[\frac{\mu^2}{\sigma^2}\right] + \mathbb{E}\left[\log\frac{1}{\sigma^2}\right] - \log 2\pi\right) \tag{26}$$

$$= \frac{1}{2}\left(-\frac{\alpha}{\beta}(y-\mu_0)^2 - \frac{1}{\lambda} + \psi(\alpha) - \log\beta - \log 2\pi\right) \tag{27}$$

where $\psi(\alpha)$ denotes the Digamma function. We used here the moments of the NIG distribution $\mathbb{E}\left[\frac{\mu}{\sigma^2}\right] = \frac{\alpha\mu_0}{\beta}$, $\mathbb{E}\left[\frac{1}{\sigma^2}\right] = \frac{\alpha}{\beta}$, $\mathbb{E}\left[\frac{\mu^2}{\sigma^2}\right] = \frac{\alpha\mu_0^2}{\beta} + \frac{1}{\lambda}$, and the moment of the inverse Gamma distribution $\mathbb{E}\left[\log\frac{1}{\sigma^2}\right] = \psi(\alpha) - \log\beta$.

## C.4 Poisson & Gamma Distributions

The Gamma distribution $\lambda \sim \Gamma(\alpha, \beta)$ is the conjugate prior of the Poisson distributions $y \sim \mathrm{Poi}(\lambda)$.

**Target Distribution.** The density and the entropy of the Poisson distribution are:

$$\mathrm{Poi}(y \mid \lambda) = \frac{\lambda^y \exp(-\lambda)}{y!} \tag{28}$$

$$\mathbb{H}[\mathrm{Poi}(\lambda)] = \lambda(1 - \log(\lambda))) + \exp(-\lambda)\sum_{k=0}^{\infty}\frac{\lambda^k \log(k!)}{k!} \tag{29}$$

**Conjugate Prior Distribution.** The density and the entropy of the Gamma distribution are:

$$\Gamma(\lambda \mid \alpha, \beta) = \frac{\beta^\alpha}{\Gamma(\alpha)}\lambda^{\alpha-1}\exp(-\beta\lambda) \tag{30}$$

$$\mathbb{H}[\Gamma(\alpha, \beta)] = \alpha + \log\Gamma(\alpha) - \log\beta + (1-\alpha)\psi(\alpha) \tag{31}$$

where $\Gamma(\alpha)$ denotes the Gamma function.

**Expected Log-Likelihood.** The expected likelihood of the Poisson distribution $\text{Poi}(\lambda)$ under the Gamma distribution $\Gamma(\alpha, \beta)$ is

$$\mathbb{E}_{\lambda \sim \Gamma(\alpha,\beta)}[\log \text{Poi}(y \mid \lambda)] = \mathbb{E}[\log \lambda]y - \mathbb{E}[\lambda] - \sum_{k=1}^{y} \log k \tag{32}$$

$$= (\psi(\alpha) - \log \beta)y - \frac{\alpha}{\beta} - \sum_{k=1}^{y} \log k \tag{33}$$

where $\psi(\alpha)$ denotes Digamma function. We used here the moments the Gamma distributions $\mathbb{E}[\log \lambda] = \psi(\alpha) - \log \beta$ and $\mathbb{E}[\lambda] = \frac{\alpha}{\beta}$. Note that $\sum_{k=1}^{y} \log k$ is constant w.r.t. parameters $\alpha, \beta$.

## D  APPROXIMATION OF ENTROPIES

The computation of a distribution's entropy often requires subtracting huge numbers from each other. While these numbers tend to be very close together, this introduces numerical challenges. For large parameter values, we therefore approximate the entropy by substituting numerically unstable terms and simplifying the resulting formula. For this procedure, we make use of the following equivalences (taken from Rocktäschel (1922) and Whittaker & Watson (1927), respectively):

$$\log \Gamma(x) \approx \frac{1}{2} \log 2\pi - x + \left(x - \frac{1}{2}\right) \log x \tag{34}$$

$$\psi(x) = \log x - \frac{1}{2x} + \mathcal{O}\left(\frac{1}{x^2}\right) \tag{35}$$

We note that Eq. 35 especially implies $\psi(x) \approx \log x$ and $x\,\psi(x) \approx x \log x - \frac{1}{2}$ for large $x$.

### D.1  DIRICHLET DISTRIBUTION

We consider a Dirichlet distribution $\text{Dir}(\boldsymbol{\alpha})$ of order $K$ with $\alpha_0 = \sum_{i=1}^{K} \alpha_i$. For $\alpha_0 \geq 10^4$, we use the following approximation:

$$\mathbb{H}\left[\text{Dir}(\boldsymbol{\alpha})\right] \approx \frac{K-1}{2}(1 + \log 2\pi) + \frac{1}{2} \sum_{i=1}^{K} \log \alpha_i - \left(K - \frac{1}{2}\right) \log \sum_{i=1}^{K} \alpha_i \tag{36}$$

### D.2  NORMAL-INVERSE-GAMMA DISTRIBUTION

We consider a Normal-Inverse-Gamma distribution $\mathcal{N}\Gamma^{-1}(\mu, \lambda, \alpha, \beta)$. For $\alpha \geq 10^4$, we use the following approximation:

$$\mathbb{H}\left[\mathcal{N}\Gamma^{-1}(\mu, \lambda, \alpha, \beta)\right] \approx 1 + \log 2\pi - 2\log \alpha + \frac{3}{2}\log \beta - \frac{1}{2}\log \lambda \tag{37}$$

### D.3  GAMMA DISTRIBUTION

We consider a Gamma distribution $\Gamma(\alpha, \beta)$. For $\alpha \geq 10^4$, we use the following approximation:

$$\mathbb{H}\left[\Gamma(\alpha, \beta)\right] \approx \frac{1}{2} + \frac{1}{2}\log 2\pi + \frac{1}{2}\log \alpha - \log \beta \tag{38}$$

## E  FORMULAE FOR UNCERTAINTY ESTIMATES

**Aleatoric Uncertainty.** The entropy of the target distribution $\mathbb{P}(y|\boldsymbol{\theta})$ was used to estimate the aleatoric uncertainty i.e. $\mathbb{H}[\mathbb{P}(y|\boldsymbol{\theta})]$.

**Epistemic Uncertainty.** The evidence parameter $n^{\text{post},(i)}$ was used to estimate the epistemic uncertainty. Due to its interpretation as a pseudo-count of observed labels, the posterior evidence parameter is indeed a natural indicator for the epistemic uncertainty.

**Predictive Uncertainty.** The entropy of the posterior distribution $\mathbb{Q}(\boldsymbol{\theta}|\boldsymbol{\chi}^{\text{post},(i)}, n^{\text{post},(i)})$ was used to estimate the predictive uncertainty.

# F  DATASET DETAILS

We use a train/validation/test split in all experiments. For datasets with a dedicated test split, we split the rest of the data into training and validation sets of size 80%/20%. For all other datasets, we used 70%/15%/15% for the train/validation/test sets. All inputs are rescaled with zero mean and unit variance. Similarly, we also scale the output target for regression. We provide the datasets at the project page [2].

**Sensorless Drive (Dua & Graff, 2017)** This is a tabular dataset where the goal is to classify extracted motor current measurements into 11 different classes. We remove the last two classes (9 and 10) from training and use them as the OOD dataset for OOD detection experiments. Each input is composed of 48 attributes describing motor behavior. The dataset contains $58,509$ samples in total.

**MNIST (LeCun et al., 2010) & Fashion-MNIST (Xiao et al., 2017)** These are image dataset where the goal is to classify pictures of hand-drawn digits into 10 classes (from digit 0 to digit 9) or classify pictures of clothers. Each input is composed of a $1 \times 28 \times 28$ tensor. The dataset contains $70,000$ samples. For OOD detection experiments againt MNIST data, we use KMNIST (Clanuwat et al., 2018) and Fashion-MNIST (Xiao et al., 2017) containing images of Japanese characters and images of clothes, respectively. For OOD detection experiments againt Fasgion-MNIST data, we use KMNIST (Clanuwat et al., 2018) and MNIST (LeCun et al., 2010) containing images of Japanese characters and images of digits, respectively. It uses the MIT License (MIT).

**CIFAR-10 (Krizhevsky et al., 2009)** This is an image dataset where the goal is to classify a picture of objects into 10 classes (airplane, automobile, bird, cat, deer, dog, frog, horse, ship, truck). Each input is a $3 \times 32 \times 32$ tensor. The dataset contains $60,000$ samples. For OOD detection experiments, we use street view house numbers (SVHN) (Netzer et al., 2011) containing images of numbers and CelebA (Liu et al., 2015) containing images of celebrity faces. We do not use CIFAR100 (Krizhevsky et al., 2009) or TinyImageNet (Fei-Fei Li & Johnson, 2017) as OOD as they also contain images of vehicles and animals similar to CIFAR10. This rightly questions to what extent are these datasets really OOD for CIFAR10. Furthermore, we generate the corrupted CIFAR-10 dataset (Hendrycks & Dietterich, 2019) with 15 corruption types per image, each with 5 different severities. It uses the MIT License (MIT).

**Bike Sharing (Fanaee-T & Gama, 2014)** This is a tabular dataset where the goal is to predict the total number of rentals within an hour. Each input is composed of 15 attributes. We removed features related to the year period (i.e. record index, date, season, months) which would make OOD detection trivial, leading to 11 attributes. The dataset contains $17,389$ samples in total. For OOD detection, we removed the attribute season from the input data and only trained on the summer season. The samples related to winter, spring and autumn were used as OOD datasets.

**Concrete (Dua & Graff, 2017)** This is a tabular dataset where the goal is to predict the compressive strength of high-performance concrete. Each input is composed of 8 attributes. The dataset contains $1,030$ samples in total. For OOD detection, we use the Energy and Kin8nm datasets which have the same input size.

**Kin8nm (Dua & Graff, 2017)** This is a tabular dataset where the goal is to predict the forward kinematics of an 8-link robot arm. Each input is composed of 8 attributes. The dataset contains $8,192$ samples in total. For OOD detection, we the Concrete and Energy datasets which have the same input size.

**NYU Depth v2 (Nathan Silberman & Fergus, 2012)** This is an image dataset where the goal is to predict the depth of room images at each pixel position. All inputs are of shape 3x640x480 tensors while we rescale outputs to be 320x240 tensors at both training and test time. This setting is slightly different from Kendall & Gal (2017) and Nathan Silberman & Fergus (2012). Indeed, (Kendall & Gal, 2017) up-scales the model output to 640x480 at training and test time while (Nathan Silberman & Fergus, 2012) up-scales the model output to 640x480 at test time only. The dataset contains $50,000$

---

[2]https://www.daml.in.tum.de/natpn

samples in total available on the DenseDepth GitHub [3]. For OOD detection, we use the KITTI (Geiger et al., 2013) dataset containing images of driving cars and two out of the 20 categories from the LSUN (Yu et al., 2015) dataset.

# G  MODEL DETAILS

We train all models using 5 seeds except for the large NYU dataset where we use a single randomly selected seed. All models are optimized with the Adam optimizer without further learning rate scheduling. We perform early stopping by checking loss improvement every epoch and a patience $p$ selected per dataset (Sensorless Drive: $p = 15$, MNIST: $p = 15$, CIFAR10: $p = 20$, Bike Sharing: $p = 50$, Concrete: $p = 50$, Kin8nm: $p = 30$, NYU Depth v2: $p = 2$). We train all models on a single GPU (NVIDIA GTX 1080 Ti or NVIDIA GTX 2080 Ti, 11 GB memory). All models are trained after a grid search for the learning rate in $[1e^{-2}, 5e^{-4}]$. The backbone architecture is shared across models and selected per dataset to match the task needs (Sensorless Drive: 3 lin. layers with 64 hidden dim, MNIST: 6 conv. layers with 32/32/32/64/64/64 filters + 3 lin. layers with hidden dim 1024/128/64, CIFAR10: 8 conv. layers with 32/32/32/64/64/128/128/128 filters + 3 lin. layers with hidden dim 1024/128/64, Bike Sharing: 3 lin. layers with 16/16/16 hidden dim, Concrete: 2 lin. layers with 16/16 hidden dim, Kin8nm: 2 lin. layers with 16/16 hidden dim, NYU Depth v2: DenseDepth + 4 upsampling layers with convolutions and skip connections). For the NYU Depth v2 dataset, we use a pretrained DenseNet for initialization of the backbone architecture which was fine-tuned during training. The remaining layers are trained from scratch. All architectures use LeakyReLU activations. For further details, we provide the code at the project page [4].

**Baselines.** For the dropout models, we use the best drop out rate $p_{\text{drop}}$ per dataset after a grid search in $\{0.1, 0.25, 0.4\}$ and sample 5 times for uncertainty estimation. Similarly, we use $m = 5$ for the ensemble baseline and the distribution distillation. Note that Ovadia et al. (2019) found that a relative small ensemble size (e.g. $m = 5$) may indeed be sufficient in practice. We also train Prior Networks where we set $\beta_{\text{in}} = 1e^2$ as suggested in the original papers (Malinin & Gales, 2018; 2019). Prior Networks use Fashion-MNIST and SVHN as training OOD datasets for MNIST and CIFAR-10, respectively. As there is no available OOD dataset for the Sensorless Drive dataset, we use Gaussian noise as training OOD data.

**Natural Posterior Network.** We perform a grid search for the entropy regularizer $\lambda$ in the range $[1e^{-5}, 0]$, for the latent dimension $H$ in $\{4, 8, 16, 32\}$, for the certainty budget $N_H$ in $\{e^{\frac{1}{2}H}, e^H, e^{\log(\sqrt{4\pi})H}\}$, and for normalizing flow type between radial flows (Rezende & Mohamed, 2015) with 8, 16 layers and Masked Autoregressive flows (Germain et al., 2015; Papamakarios et al., 2017) with 4, 8, 16 layers. Further results on latent dimensions, density types, number of normalizing flow layers and certainty budget are presented in Sec. I.6. We use "warm-up" training for the normalizing flows for all datasets except for the simple Concrete and Kin8nm datasets, and the NYU Depth v2 dataset which starts from a pretrained encoder. We use "fine-tuning" for the normalizing flows for all datasets except for the simple Concrete and Kin8nm datasets. As prior parameters, we set $\chi^{\text{prior}} = \mathbf{1}_C/C, n^{\text{prior}} = C$ for classification, $\chi^{\text{prior}} = (0, 100)^T, n^{\text{prior}} = 1$ for regression and $\chi^{\text{prior}} = 1, n^{\text{prior}} = 1$ for count prediction. Note that the mean of these prior distributions correspond to an equiprobable Categorical distribution $\text{Cat}(\mathbf{1}_C/C)$, a Normal distribution with large variance $\mathcal{N}(0, 10)$ and a Poisson distribution with a unitary mean $\text{Poi}(1)$. Those prior target distributions represent the safe default prediction when no evidence is predicted.

# H  EXPERIMENT DETAILS

**Target Error Metric.** For classification, we use the standard accuracy $\frac{1}{N} \sum_i \mathbb{I}[\boldsymbol{y}^{*,(i)} = \boldsymbol{y}^{(i)}]$ where $\boldsymbol{y}^{*,(i)}$ is the one-hot true label and $\boldsymbol{y}^{(i)}$ is the one-hot predicted label. For regression, we use the standard Root Mean Square Error $\sqrt{\frac{1}{N} \sum_i^N (y^{*,(i)} - y^{(i)})^2}$.

**Calibration Metric.** For classification, we use the Brier score which is computed as $\frac{1}{C} \sum_i^N ||\boldsymbol{p}^{(i)} - \boldsymbol{y}^{(i)}||_2$ where $\boldsymbol{p}^{(i)}$ is the predicted softmax probability and $\boldsymbol{y}^{(i)}$ is the one-hot encoded ground-truth

---

[3] https://github.com/ialhashim/DenseDepth
[4] https://www.daml.in.tum.de/natpn

label. For regression and count prediction, we use the absolute difference between the percentile $p$ and the percentage of target lying in the confidence interval $I_p = [0, \frac{p}{2}] \cup [1 - \frac{p}{2}, 1]$ under the predicted target distribution. Formally, we compute $p_{\text{pred}} = \frac{1}{N} \sum_i \mathbb{I}[F_{\boldsymbol{\theta}^{(i)}}(y^{*,(i)})) \in I_p]$ where $F_{\boldsymbol{\theta}^{(i)}}(y^{*,(i)}) = \mathbb{P}(y \leq y^{*,(i)} | \boldsymbol{\theta}^{(i)})$ is the cumulative function of the predicted target distribution evaluated at the true target. For example, the percentile $p = 0.1$ would be compared to $p_{\text{pred}} = \frac{1}{N} \sum_i \mathbb{I}[F_{\boldsymbol{\theta}^{(i)}}(y^{*,(i)})) \in [0, 0.05] \cup [0.95, 1]]$ which should be close to $0.10$ for calibrated predictions. We compute a single calibration score by summing the square difference for $p \in \{0.1, \ldots, 0.9\}$ i.e. $\sqrt{\sum_p (p - p_{\text{pred}})^2}$ (Kuleshov et al., 2018).

**OOD Metric.** The OOD detection task can be evaluated as a binary classification. Hence, we assign class 1 to ID data and class 0 to OOD data task and use the aleatoric and epistemic uncertainty estimates as scores for OOD data. It enables to compute final scores using the area under the precision-recall curve (AUC-PR) and the area under the receiver operating characteristic curve (AUC-ROC). Both metrics have been scaled by 100. We obtain numbers in $[0, 100]$ for all scores instead of $[0, 1]$. Results for AUC-ROC are reported in Sec. I.7. For the aleatoric uncertainty, we use the negative entropy of the predicted target distribution. For the epistemic uncertainty, we use the predicted evidence for models parametrizing conjugate-prior, or the variance of the predicted winning probability class for classification and the variance of the mean for regression and class count prediction for ensemble or dropout.

**Inference Time Metric.** We measure inference time of models in ms and used NVIDIA GTX 1080 Ti GPUs. We evaluate the inference for one classification dataset (CIFAR-10) and one regression dataset (NYU Depth v2). For evaluation, we use a randomly initialized model and simply push random data through the model with batch size of 4,096 CIFAR10 and batch size of 4 for NYU Depth v2. The final numbers are averaged over 100 batches excluding the first batch due to GPU initialization. Compared models shared the same backbone architecture.

# I  ADDITIONAL EXPERIMENTS

## I.1  MNIST, FASHION MNIST, CIFAR10 AND BIKE SHARING RESULTS

Table 8: Results on MNIST (classification with Categorical target distribution). Best scores among all single-pass models are in bold. Best scores among all models are starred. Gray numbers indicate that R-PriorNet has seen samples from the FMNIST dataset during training.

| | Accuracy | Brier | K. Alea. | K. Epist. | F. Alea. | F. Epist. | OODom Alea. | OODom Epist. |
|---|---|---|---|---|---|---|---|---|
| **Dropout** | $99.45 \pm 0.01$ | $1.07 \pm 0.05$ | $98.27 \pm 0.05$ | $97.82 \pm 0.08$ | $*99.40 \pm 0.03$ | $98.01 \pm 0.14$ | $43.86 \pm 1.62$ | $74.09 \pm 0.92$ |
| **Ensemble** | $99.46 \pm 0.02$ | $1.02 \pm 0.02$ | $98.39 \pm 0.07$ | $98.43 \pm 0.05$ | $99.33 \pm 0.06$ | $98.73 \pm 0.08$ | $40.98 \pm 1.80$ | $66.54 \pm 0.58$ |
| **NatPE** | $*99.55 \pm 0.01$ | $*0.84 \pm 0.03$ | $96.39 \pm 0.73$ | $*99.61 \pm 0.02$ | $97.49 \pm 0.85$ | $*99.70 \pm 0.04$ | $*100.00 \pm 0.00$ | $*100.00 \pm 0.00$ |
| **StandardNet** | $98.91 \pm 0.06$ | $1.81 \pm 0.14$ | $95.81 \pm 0.44$ | – | $96.29 \pm 1.04$ | – | $47.53 \pm 3.44$ | – |
| **SNGP** | $99.34 \pm 0.03$ | $2.62 \pm 0.04$ | $98.85 \pm 0.11$ | – | $98.04 \pm 0.34$ | – | $*100.00 \pm 0.00$ | – |
| **R-PriorNet** | $99.35 \pm 0.04$ | $0.97 \pm 0.03$ | $*99.33 \pm 0.18$ | $99.28 \pm 0.25$ | $100.00 \pm 0.00$ | $100.00 \pm 0.00$ | $97.48 \pm 0.66$ | $31.03 \pm 0.13$ |
| **EnD$^2$** | $99.24 \pm 0.05$ | $6.19 \pm 0.13$ | $98.36 \pm 0.15$ | $98.76 \pm 0.13$ | $\mathbf{99.25 \pm 0.16}$ | $99.35 \pm 0.14$ | $48.09 \pm 1.38$ | $31.60 \pm 0.39$ |
| **PostNet** | $99.36 \pm 0.02$ | $1.33 \pm 0.04$ | $98.88 \pm 0.05$ | $98.79 \pm 0.07$ | $98.89 \pm 0.23$ | $98.85 \pm 0.23$ | $*100.00 \pm 0.00$ | $*100.00 \pm 0.00$ |
| **NatPN** | $\mathbf{99.47 \pm 0.02}$ | $1.09 \pm 0.03$ | $99.20 \pm 0.20$ | $\mathbf{99.39 \pm 0.08}$ | $99.16 \pm 0.28$ | $\mathbf{99.54 \pm 0.09}$ | $99.99 \pm 0.01$ | $*100.00 \pm 0.00$ |

Table 9: Results on FMNIST (classification with Categorical target distribution). Best scores among all single-pass models are in bold. Best scores among all models are starred. Gray numbers indicate that R-PriorNet has seen samples from the KMNIST dataset during training.

| | Accuracy | Brier | M. Alea. | M. Epist. | K. Alea. | K. Epist. | OODom Alea. | OODom Epist. |
|---|---|---|---|---|---|---|---|---|
| **Dropout** | $92.44 \pm 0.17$ | $13.89 \pm 0.31$ | $60.75 \pm 1.41$ | $75.85 \pm 1.73$ | $76.57 \pm 1.30$ | $92.48 \pm 0.46$ | $39.97 \pm 0.69$ | $90.90 \pm 1.74$ |
| **Ensemble** | $92.64 \pm 0.10$ | $13.63 \pm 0.25$ | $77.14 \pm 1.49$ | $90.78 \pm 0.75$ | $86.20 \pm 0.76$ | $95.16 \pm 0.35$ | $37.30 \pm 0.83$ | $82.93 \pm 0.96$ |
| **NatPE** | $*92.89 \pm 0.06$ | $14.44 \pm 0.06$ | $82.56 \pm 0.33$ | $96.38 \pm 0.29$ | $92.12 \pm 0.17$ | $*98.79 \pm 0.09$ | $*100.00 \pm 0.00$ | $*100.00 \pm 0.00$ |
| **StandardNet** | $90.28 \pm 0.24$ | $17.12 \pm 0.53$ | $71.81 \pm 2.43$ | – | $82.28 \pm 0.97$ | – | $32.82 \pm 0.73$ | – |
| **SNGP** | $91.38 \pm 0.08$ | $16.73 \pm 0.46$ | $89.40 \pm 1.66$ | – | $95.31 \pm 0.42$ | – | $100.00 \pm 0.00$ | – |
| **R-PriorNet** | $91.53 \pm 0.10$ | $*12.21 \pm 0.20$ | $*98.83 \pm 0.49$ | $*99.54 \pm 0.18$ | $99.96 \pm 0.02$ | $99.99 \pm 0.00$ | $72.23 \pm 6.32$ | $48.84 \pm 6.09$ |
| **EnD$^2$** | $\mathbf{91.84 \pm 0.03}$ | $29.23 \pm 0.79$ | $79.32 \pm 1.39$ | $91.61 \pm 1.04$ | $91.99 \pm 0.06$ | $98.36 \pm 0.20$ | $43.70 \pm 3.37$ | $36.73 \pm 3.74$ |
| **PostNet** | $91.04 \pm 0.10$ | $16.11 \pm 0.30$ | $90.56 \pm 1.25$ | $92.10 \pm 1.77$ | $*96.65 \pm 0.33$ | $97.06 \pm 0.42$ | $*100.00 \pm 0.00$ | $*100.00 \pm 0.00$ |
| **NatPN** | $91.65 \pm 0.14$ | $14.88 \pm 0.30$ | $81.12 \pm 2.77$ | $96.51 \pm 0.81$ | $93.03 \pm 1.00$ | $\mathbf{98.38 \pm 0.23}$ | $99.99 \pm 0.01$ | $*100.00 \pm 0.00$ |

Table 10: Classification results on CIFAR-10 with Categorical target distribution. Best scores among all single-pass models are in bold. Best scores among all models are starred. Gray numbers indicate that R-PriorNet has seen samples from the SVHN dataset during training.

| | Accuracy | Brier | SVHN Alea. | SVHN Epist. | CelebA Alea. | CelebA Epist. | OODom Alea. | OODom Epist. |
|---|---|---|---|---|---|---|---|---|
| **Dropout** | 88.15 ± 0.20 | 19.59 ± 0.41 | 80.63 ± 1.59 | 73.09 ± 1.51 | 71.84 ± 4.28 | 71.04 ± 3.92 | 18.42 ± 1.11 | 49.69 ± 9.10 |
| **Ensemble** | *89.95 ± 0.11 | 17.33 ± 0.17 | 85.26 ± 0.84 | 82.51 ± 0.63 | 76.20 ± 0.87 | 74.23 ± 0.78 | 25.30 ± 4.02 | 89.21 ± 7.55 |
| **NatPE** | 89.21 ± 0.09 | 17.41 ± 0.12 | 85.66 ± 0.34 | *83.16 ± 0.67 | *78.95 ± 1.15 | *82.06 ± 1.30 | 87.27 ± 1.79 | *98.88 ± 0.26 |
| **SNGP** | 84.06 ± 1.68 | 30.49 ± 2.99 | 79.95 ± 1.82 | – | 67.82 ± 3.67 | – | *96.00 ± 1.67 | – |
| **R-PriorNet** | **88.94 ± 0.23** | *15.99 ± 0.32 | 99.87 ± 0.02 | 99.94 ± 0.01 | 67.74 ± 4.86 | 59.55 ± 7.90 | 42.21 ± 8.77 | 38.25 ± 9.82 |
| **EnD²** | 84.03 ± 0.25 | 40.84 ± 0.36 | *86.47 ± 0.66 | **81.84 ± 0.92** | 75.54 ± 1.79 | 75.94 ± 1.82 | 42.19 ± 8.77 | 15.79 ± 0.27 |
| **PostNet** | 87.95 ± 0.20 | 20.19 ± 0.40 | 82.35 ± 0.68 | 79.24 ± 1.49 | 72.96 ± 2.33 | 75.84 ± 1.61 | 85.89 ± 4.10 | 92.30 ± 2.18 |
| **NatPN** | 87.90 ± 0.16 | 19.99 ± 0.46 | 82.29 ± 1.11 | 77.83 ± 1.22 | **76.01 ± 1.18** | **76.87 ± 3.38** | 93.67 ± 3.03 | **94.90 ± 3.09** |

Table 11: Results on the Bike Sharing Dataset with Normal $\mathcal{N}$ and Poison Poi target distributions. Best scores among all single-pass models are in bold. Best scores among all models are starred.

| | RMSE | Calibration | Winter Epist. | Spring Epist. | Autumn Epist. | OODom Epist. |
|---|---|---|---|---|---|---|
| **Dropout-$\mathcal{N}$** | 70.20 ± 1.30 | 6.05 ± 0.77 | 15.26 ± 0.51 | 13.66 ± 0.16 | 15.11 ± 0.46 | 99.99 ± 0.01 |
| **Ensemble-$\mathcal{N}$** | *48.02 ± 2.78 | 5.88 ± 1.00 | 42.46 ± 2.29 | 21.28 ± 0.38 | 21.97 ± 0.58 | *100.00 ± 0.00 |
| **StandardNet-$\mathcal{N}$** | 58.49 ± 4.37 | 2.32 ± 0.88 | – | – | – | – |
| **EvReg-$\mathcal{N}$** | **49.58 ± 1.51** | 3.77 ± 0.81 | 17.19 ± 0.76 | 15.54 ± 0.65 | 14.75 ± 0.29 | 34.99 ± 17.02 |
| **NatPN-$\mathcal{N}$** | 49.85 ± 1.38 | *1.95 ± 0.34 | *55.04 ± 6.81 | *23.25 ± 1.20 | *27.78 ± 2.47 | *100.00 ± 0.00 |

## I.2 UNCERTAINTY VISUALIZATION ON TOY DATASETS

We visualize the aleatoric and the epistemic uncertainty for two toy datasets with three classes with the same number of training examples for the three classes (see Fig. 4) and different number of training examples for the three classes (see Fig. 5). The predictions are more aleatorically certain close to training samples. The preidctions are more epistemically uncertain close to fewer training examples and very epistemically uncertain for region far from all training data.

Figure 4: Visualization of the aleatoric and epistemic uncertainty on a 2D toy dataset with 3 classes with 900 training samples for each class.

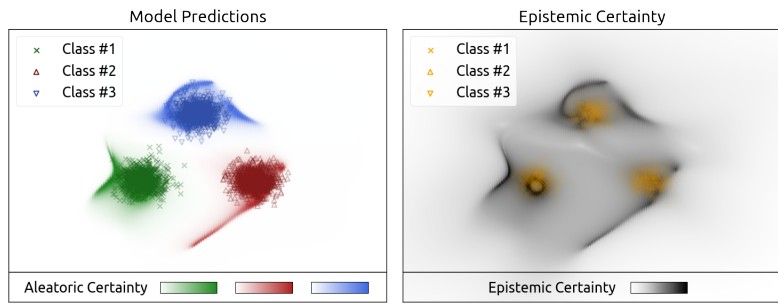

Figure 5: Visualization of the aleatoric and epistemic uncertainty on a 2D toy dataset with 3 classes with 900 training samples for class 1 (green), 600 training samples for class 2 (red) and 300 training samples for class 2 (blue).

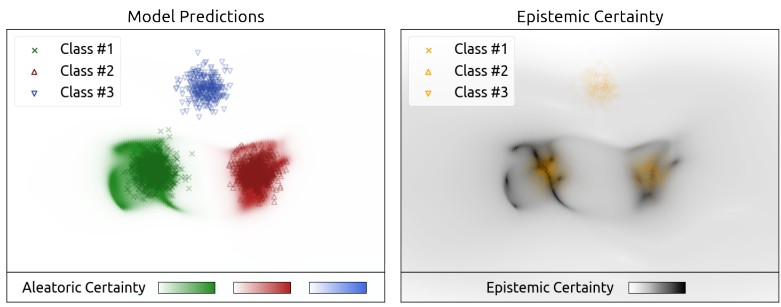

### I.3 Latent Space Visualizations

We propose additional visualizations of the latent space for MNIST with t-SNE (Chan et al., 2019) with different perplexities (see Fig. 6). For all perplexities, we clearly observe ten green clusters corresponding to the ten classes for MNIST. The KNMIST (OOD) samples in red can easily be separated from the MNIST (ID) samples in green. As desired, NatPN assigns higher log-probabilities used in evidence computation to ID samples from MNIST.

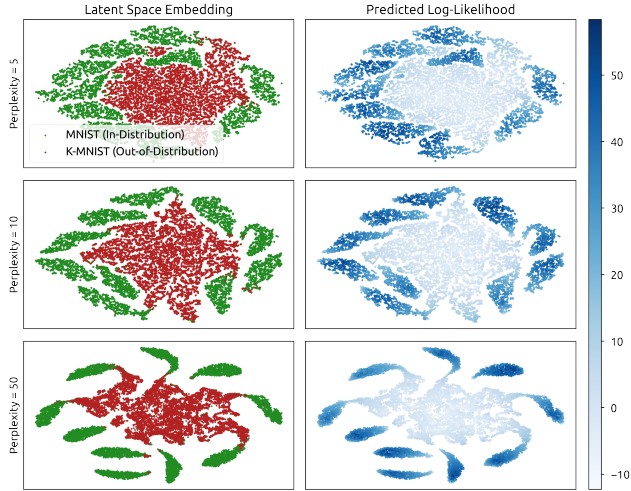

Figure 6: t-SNE visualization of the latent space of NatPN on MNIST (ID) vs KMNIST (OOD). On the left, The ID data (MNIST in green) can easily be distinguished from the OOD data (KMNIST in red). On the right, NatPN correctly assigns higher likelihood to ID data.

### I.4 Histogram of Uncertainty Estimates

We visualize the histogram distribution of the entropy of the posterior distribution accounting for predictive uncertainty for ID (MNIST/NYU) and OOD (KNMIST and Fashion-MNIST/LSUN classroom and LSUN church + KITTI) (see Fig. 7). We clearly observe lower predictive entropy for ID data than for OOD data for both MNIST and NYU datasets. On one hand, the entropy clearly differentiates between ID data (MNIST) and any other OOD datasets (KMNIST, Fashion MNIST, OODom) for classification. We intuitively explain this clear distinction since the samples from the OOD datasets are irrelevant for the digit classification task. On the other hand, the entropy is still a good indicator of ID (NYU) and OOD datasets (LSUN classroom and LSUN church + KITTI) for regression although the distinction between ID and OOD datasets is less strong compared to MNIST. We intuitively explain this behavior since the task of depth estimation is still relevant to LSUN classroom and LSUN church + KITTI.

Figure 7: Histogram of the entropy of the posterior distribution accounting for the predictive uncertainty of NatPN on MNIST (ID) vs KMNIST, Fashion-MNIST, Out-Of-Domain (OOD) and NYU (ID) vs LSUN classroom and LSUN church + KITTI (OOD). In both cases, low entropy is a good indicator of in-distribution data.

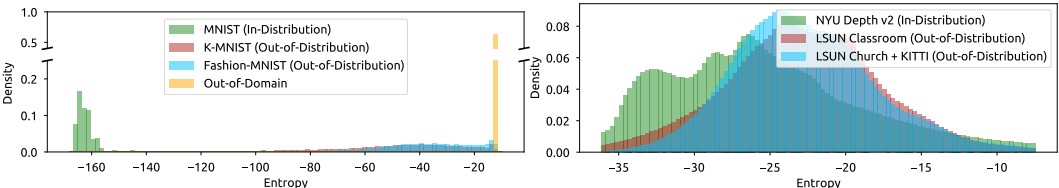

## I.5 Uncertainty Visualization on NYU Depth v2 Dataset

We visualize the prediction and the predictive uncertainty per pixel for the NYU Depth v2 dataset (see Fig. 8). We observe accurate target predictions compared to the ground truth depth of the images. Further NatPN assigns higher uncertainty for pixels close to object edges, which is reasonable since the depth abruptly change at these locations.

Figure 8: Visualization of the predicted depth and predictive uncertainty estimates of NatPN per pixel on the NYU Depth v2 dataset. NatPN predicts accurate depth uncertainty and reasonably assigns higher uncertainty to object edges.

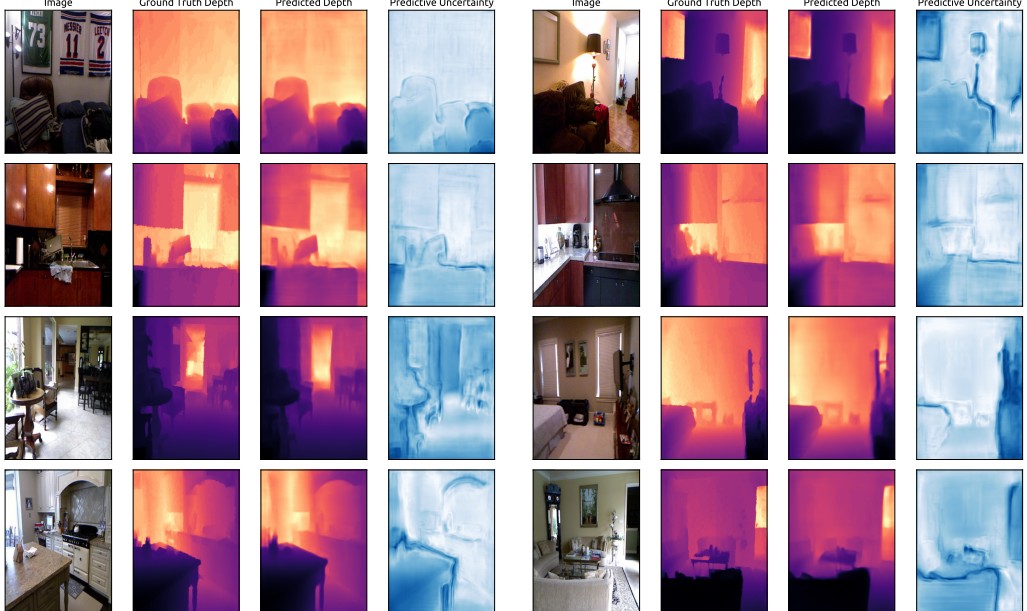

## I.6 Hyper-Parameter Study

As a ablation study, we also report the results of the grid search on the latent dimension, normalizing flow types and number of normalizing flow layers for MNIST, CIFAR-10 and Bike Sharing datasets in Tab. 12, 13, 14, 15. While most models converge to fairly good uncertainty estimates, we notice that 16 layers of simple radial flows on latent spaces of 16 dimensions were achieving very good results in practice.

Changing the flow type or the number of normalizing flow layers does not lead to strong variations of the results except for Bike sharing with Poisson target distributions. In this case, more complex MAF normalizing flows improve NatPN performance.

The latent dimension appears to be a more important choice for the model convergence. As an example, a higher latent dimension of 16 or 32 leads to significantly better performances than a latent dimension of 4 on MNIST, CIFAR10 and Bike Sharing datasets. We hypothesize that too low latent dimensions are less able to encode the necessary information for the prediction task, leading to worse target errors.

Further, we compare three different variants of the certainty budget $N_H$ used in the evidence computation $n^{(i)} = N_H \, \mathbb{P}(\boldsymbol{z}^{(i)} \,|\, \boldsymbol{\omega})$: a constant unit budget (i.e. $N_H = 1$ (I)) corresponding to a fixed budget regardless of the number of training data and the latent dimension, a budget equals to the number of training data (i.e. $N_H = N$ (II)) similarly to Charpentier et al. (2020), or a budget which scales exponentially with the number of latent dimensions (e.g. $N_H$ equal to $e^{\frac{1}{2}H}$ (III), $e^H$ (IV) or $e^{\log(\sqrt{4\pi})H}$ (V)). We observe that scaling the budget w.r.t. the latent space dimension ($H$-budget) is more stable in practice than constant budget (1-budget) and budget related to the number of training data ($N$-budget) (see. Tab. 12,13,14,15). In aprticular, the $H$-budgets achieve more better results on

higher latent dimensions and performance on par with the other certainty budget scheme otherwise. The intuition is that due to the curse of dimensionality, the expected value of a probability density function $\mathbb{E}_{\boldsymbol{z}}[\mathbb{P}(\boldsymbol{z})]$ tends to decrease exponentially fast. For example, we have $\mathbb{E}_{\boldsymbol{z}}[\mathbb{P}(\boldsymbol{z})] = \frac{1}{(\sqrt{4\pi})^H}$ when $\boldsymbol{z} \sim \mathcal{N}(\boldsymbol{0}, \boldsymbol{1})$ in a $H$-dimensional space. Increasing the certainty budget $N_H$ exponentially w.r.t. to the dimension $H$ avoids numerical issues by allocating close to $0$ evidence to latent representations. In our experiments, we use a grid search in different exponential scaling $N_H$ equal to $e^{\frac{1}{2}H}$ (III), $e^H$ (IV), $e^{\log(\sqrt{4\pi})H}$ (V).

## I.7 OOD DETECTION WITH AUC-ROC SCORES

In addition to the AUC-APR scores, we report the OOD detection results for MNIST, CIFAR10 and Bike Sharing datasets in Tab. 16, 17, 18 with AUC-ROC scores. Similarly as with AUC-APR scores, NatPN and NatPE achieve very competitive performances compare to the baselines. It particular, they outperform all baselines to detect challenging OODom data.

Table 12: MNIST comparison (<latent dim> – <certainty budget> – <radial layers>/<MAF layers>). Bold and starred number indicate best score among all models.

| | Accuracy | Brier | K. Alea. | K. Epist. | F. Alea. | F. Epist. | OODom Alea. | OODom Epist. |
|---|---|---|---|---|---|---|---|---|
| $H = 4, N_H = $ I, Flow = 8/0 | 61.92 ± 3.79 | 64.33 ± 3.98 | 88.70 ± 1.65 | 87.66 ± 2.47 | 90.07 ± 3.43 | 89.52 ± 2.25 | *100.00 ± 0.00 | *100.00 ± 0.00 |
| $H = 4, N_H = $ I, Flow = 16/0 | 77.46 ± 6.26 | 59.11 ± 1.96 | 91.38 ± 2.50 | 91.48 ± 0.85 | 92.46 ± 1.72 | 93.03 ± 1.60 | 99.91 ± 0.04 | *100.00 ± 0.00 |
| $H = 4, N_H = $ I, Flow = 0/4 | 75.04 ± 10.10 | 59.46 ± 9.92 | 92.36 ± 3.34 | 92.64 ± 4.06 | 93.28 ± 2.66 | 91.90 ± 4.71 | 99.82 ± 0.05 | *100.00 ± 0.00 |
| $H = 4, N_H = $ I, Flow = 0/8 | 80.97 ± 4.29 | 58.35 ± 1.75 | 91.87 ± 2.55 | 92.65 ± 2.52 | 93.58 ± 2.93 | 96.50 ± 1.32 | 99.85 ± 0.04 | *100.00 ± 0.00 |
| $H = 4, N_H = $ II, Flow = 8/0 | 99.33 ± 0.05 | 3.43 ± 0.08 | 92.45 ± 0.33 | 67.84 ± 2.95 | 96.55 ± 1.23 | 71.57 ± 4.00 | *100.00 ± 0.00 | *100.00 ± 0.00 |
| $H = 4, N_H = $ II, Flow = 16/0 | 99.36 ± 0.03 | 2.72 ± 0.08 | 94.87 ± 0.49 | 86.13 ± 1.47 | 95.03 ± 0.44 | 88.62 ± 1.80 | *100.00 ± 0.00 | *100.00 ± 0.00 |
| $H = 4, N_H = $ II, Flow = 0/4 | 98.86 ± 0.05 | 4.26 ± 0.90 | 98.93 ± 0.16 | 98.04 ± 0.38 | 99.16 ± 0.23 | 98.52 ± 0.52 | 99.71 ± 0.02 | *100.00 ± 0.00 |
| $H = 4, N_H = $ II, Flow = 0/8 | 98.77 ± 0.15 | 5.44 ± 1.57 | 98.43 ± 0.26 | 97.87 ± 0.40 | 98.85 ± 0.26 | 98.35 ± 0.44 | 99.76 ± 0.03 | *100.00 ± 0.00 |
| $H = 4, N_H = $ III, Flow = 8/0 | 78.14 ± 5.55 | 48.46 ± 4.78 | 88.79 ± 1.05 | 87.28 ± 1.96 | 90.56 ± 1.10 | 89.69 ± 1.88 | *100.00 ± 0.00 | 99.99 ± 0.01 |
| $H = 4, N_H = $ III, Flow = 16/0 | 79.56 ± 5.27 | 41.44 ± 4.86 | 93.92 ± 1.25 | 93.99 ± 0.39 | 95.61 ± 0.88 | 95.64 ± 0.58 | 99.92 ± 0.07 | *100.00 ± 0.00 |
| $H = 4, N_H = $ III, Flow = 0/4 | 76.08 ± 7.67 | 58.56 ± 11.60 | 93.72 ± 3.51 | 94.58 ± 2.66 | 94.34 ± 3.39 | 96.60 ± 1.86 | 99.86 ± 0.05 | *100.00 ± 0.00 |
| $H = 4, N_H = $ III, Flow = 0/8 | 79.91 ± 10.84 | 50.78 ± 10.99 | 94.91 ± 2.23 | 96.44 ± 1.49 | 95.97 ± 1.37 | 98.35 ± 0.54 | 99.78 ± 0.07 | *100.00 ± 0.00 |
| $H = 4, N_H = $ IV, Flow = 8/0 | 85.31 ± 3.89 | 34.01 ± 5.20 | 89.65 ± 1.24 | 86.81 ± 1.36 | 92.78 ± 0.40 | 86.93 ± 2.47 | *100.00 ± 0.00 | 99.99 ± 0.00 |
| $H = 4, N_H = $ IV, Flow = 16/0 | 89.36 ± 0.15 | 26.48 ± 2.64 | 94.56 ± 0.65 | 93.48 ± 0.88 | 96.03 ± 1.01 | 94.74 ± 1.26 | 99.98 ± 0.02 | *100.00 ± 0.00 |
| $H = 4, N_H = $ IV, Flow = 0/4 | 91.64 ± 6.70 | 22.61 ± 13.67 | 96.63 ± 1.92 | 93.07 ± 1.49 | 98.43 ± 0.52 | 94.55 ± 2.01 | 99.81 ± 0.06 | *100.00 ± 0.00 |
| $H = 4, N_H = $ IV, Flow = 0/8 | 98.14 ± 0.46 | 12.73 ± 5.11 | 98.68 ± 0.25 | 97.61 ± 0.64 | 98.93 ± 0.20 | 97.73 ± 0.63 | 99.78 ± 0.05 | *100.00 ± 0.00 |
| $H = 4, N_H = $ V, Flow = 8/0 | 93.59 ± 2.32 | 20.40 ± 3.68 | 92.06 ± 1.39 | 87.15 ± 2.25 | 93.55 ± 1.03 | 87.11 ± 2.97 | *100.00 ± 0.00 | 99.99 ± 0.00 |
| $H = 4, N_H = $ V, Flow = 16/0 | 97.19 ± 1.97 | 14.66 ± 3.13 | 94.67 ± 1.45 | 92.82 ± 1.04 | 95.80 ± 1.17 | 95.27 ± 0.70 | *100.00 ± 0.00 | *100.00 ± 0.00 |
| $H = 4, N_H = $ V, Flow = 0/4 | 96.69 ± 1.56 | 16.51 ± 6.21 | 97.78 ± 0.63 | 94.58 ± 1.06 | 96.95 ± 1.76 | 92.94 ± 1.74 | 99.80 ± 0.03 | *100.00 ± 0.00 |
| $H = 4, N_H = $ V, Flow = 0/8 | 98.34 ± 0.19 | 6.72 ± 1.54 | 98.85 ± 0.15 | 98.02 ± 0.49 | 99.08 ± 0.08 | 98.41 ± 0.34 | 99.69 ± 0.04 | *100.00 ± 0.00 |
| $H = 16, N_H = $ I, Flow = 8/0 | 99.45 ± 0.04 | 1.49 ± 0.11 | 97.75 ± 0.32 | 89.16 ± 0.34 | 98.13 ± 0.27 | 89.16 ± 0.41 | *100.00 ± 0.00 | *100.00 ± 0.00 |
| $H = 16, N_H = $ I, Flow = 16/0 | 99.48 ± 0.01 | 1.41 ± 0.07 | *99.32 ± 0.11 | 99.34 ± 0.05 | *99.52 ± 0.07 | *99.59 ± 0.05 | *100.00 ± 0.00 | *100.00 ± 0.00 |
| $H = 16, N_H = $ I, Flow = 0/4 | 99.18 ± 0.05 | 1.83 ± 0.08 | 98.78 ± 0.11 | 97.92 ± 0.15 | 99.39 ± 0.07 | 98.32 ± 0.15 | 99.93 ± 0.02 | *100.00 ± 0.00 |
| $H = 16, N_H = $ I, Flow = 0/8 | 99.13 ± 0.06 | 1.97 ± 0.09 | 98.97 ± 0.04 | 98.33 ± 0.07 | *99.52 ± 0.04 | 99.28 ± 0.09 | 99.88 ± 0.02 | *100.00 ± 0.00 |
| $H = 16, N_H = $ II, Flow = 8/0 | 99.42 ± 0.02 | 1.42 ± 0.10 | 96.62 ± 0.37 | 82.95 ± 1.46 | 97.34 ± 0.59 | 82.85 ± 1.37 | *100.00 ± 0.00 | *100.00 ± 0.00 |
| $H = 16, N_H = $ II, Flow = 16/0 | 99.39 ± 0.02 | 1.52 ± 0.23 | 98.19 ± 0.38 | 95.92 ± 1.30 | 98.48 ± 0.37 | 96.20 ± 1.30 | *100.00 ± 0.00 | *100.00 ± 0.00 |
| $H = 16, N_H = $ II, Flow = 0/4 | 99.24 ± 0.04 | 1.74 ± 0.08 | 98.65 ± 0.12 | 97.49 ± 0.20 | 99.29 ± 0.09 | 98.62 ± 0.20 | 99.94 ± 0.02 | *100.00 ± 0.00 |
| $H = 16, N_H = $ II, Flow = 0/8 | 99.26 ± 0.04 | 1.73 ± 0.09 | 98.89 ± 0.06 | 98.09 ± 0.12 | 99.33 ± 0.11 | 98.73 ± 0.27 | 99.95 ± 0.01 | *100.00 ± 0.00 |
| $H = 16, N_H = $ III, Flow = 8/0 | 99.47 ± 0.02 | 1.90 ± 0.50 | 95.08 ± 1.14 | 83.27 ± 0.43 | 96.03 ± 1.33 | 82.95 ± 0.30 | *100.00 ± 0.00 | *100.00 ± 0.00 |
| $H = 16, N_H = $ III, Flow = 16/0 | 99.47 ± 0.02 | *1.09 ± 0.03 | 99.20 ± 0.20 | *99.39 ± 0.08 | 99.16 ± 0.28 | 99.54 ± 0.09 | 99.99 ± 0.01 | *100.00 ± 0.00 |
| $H = 16, N_H = $ III, Flow = 0/4 | 99.25 ± 0.03 | 1.65 ± 0.07 | 98.72 ± 0.07 | 97.95 ± 0.15 | 99.44 ± 0.04 | 98.96 ± 0.17 | 99.95 ± 0.01 | *100.00 ± 0.00 |
| $H = 16, N_H = $ III, Flow = 0/8 | 99.26 ± 0.05 | 1.78 ± 0.07 | 98.89 ± 0.08 | 98.24 ± 0.20 | 99.23 ± 0.28 | 98.70 ± 0.37 | 99.95 ± 0.01 | *100.00 ± 0.00 |
| $H = 16, N_H = $ IV, Flow = 8/0 | 99.33 ± 0.04 | 1.58 ± 0.03 | 97.08 ± 0.35 | 77.40 ± 1.79 | 98.15 ± 0.61 | 77.46 ± 1.82 | *100.00 ± 0.00 | *100.00 ± 0.00 |
| $H = 16, N_H = $ IV, Flow = 16/0 | 99.37 ± 0.03 | 1.47 ± 0.11 | 97.52 ± 0.50 | 93.53 ± 0.89 | 98.04 ± 0.55 | 94.28 ± 0.60 | *100.00 ± 0.00 | *100.00 ± 0.00 |
| $H = 16, N_H = $ IV, Flow = 0/4 | 99.23 ± 0.06 | 1.74 ± 0.10 | 98.64 ± 0.06 | 97.00 ± 0.38 | 99.31 ± 0.07 | 98.24 ± 0.35 | 99.93 ± 0.02 | *100.00 ± 0.00 |
| $H = 16, N_H = $ IV, Flow = 0/8 | 99.17 ± 0.03 | 1.83 ± 0.04 | 98.54 ± 0.10 | 97.43 ± 0.23 | 99.15 ± 0.15 | 98.28 ± 0.34 | 99.93 ± 0.01 | *100.00 ± 0.00 |
| $H = 16, N_H = $ V, Flow = 8/0 | 99.36 ± 0.03 | 1.46 ± 0.02 | 97.60 ± 0.63 | 72.45 ± 3.67 | 98.25 ± 0.83 | 72.69 ± 3.58 | *100.00 ± 0.00 | *100.00 ± 0.00 |
| $H = 16, N_H = $ V, Flow = 16/0 | 99.38 ± 0.02 | 1.58 ± 0.12 | 96.17 ± 0.61 | 90.44 ± 1.80 | 96.80 ± 0.33 | 91.12 ± 1.87 | *100.00 ± 0.00 | *100.00 ± 0.00 |
| $H = 16, N_H = $ V, Flow = 0/4 | 99.15 ± 0.04 | 1.89 ± 0.08 | 98.57 ± 0.13 | 96.06 ± 0.34 | 99.08 ± 0.11 | 97.79 ± 0.42 | 99.91 ± 0.01 | *100.00 ± 0.00 |
| $H = 16, N_H = $ V, Flow = 0/8 | 99.23 ± 0.04 | 1.98 ± 0.08 | 98.53 ± 0.04 | 97.03 ± 0.19 | 98.94 ± 0.13 | 97.83 ± 0.20 | 99.90 ± 0.01 | *100.00 ± 0.00 |
| $H = 32, N_H = $ I, Flow = 8/0 | *99.49 ± 0.03 | 2.28 ± 0.89 | 92.36 ± 1.75 | 76.18 ± 2.10 | 93.90 ± 1.59 | 76.38 ± 2.11 | *100.00 ± 0.00 | *100.00 ± 0.00 |
| $H = 32, N_H = $ I, Flow = 16/0 | 99.47 ± 0.01 | 1.47 ± 0.26 | 96.20 ± 0.25 | 95.95 ± 1.21 | 95.96 ± 0.77 | 95.97 ± 1.14 | *100.00 ± 0.00 | *100.00 ± 0.00 |
| $H = 32, N_H = $ I, Flow = 0/4 | 99.29 ± 0.02 | 1.42 ± 0.04 | 99.02 ± 0.05 | 98.14 ± 0.16 | 99.46 ± 0.02 | 98.57 ± 0.11 | 99.95 ± 0.01 | *100.00 ± 0.00 |
| $H = 32, N_H = $ I, Flow = 0/8 | 99.27 ± 0.04 | 1.45 ± 0.06 | 98.95 ± 0.08 | 98.52 ± 0.08 | 99.35 ± 0.06 | 98.80 ± 0.04 | 99.99 ± 0.00 | *100.00 ± 0.00 |
| $H = 32, N_H = $ II, Flow = 8/0 | 99.44 ± 0.05 | 1.29 ± 0.18 | 96.03 ± 1.88 | 73.89 ± 2.45 | 96.83 ± 1.13 | 73.98 ± 2.41 | *100.00 ± 0.00 | *100.00 ± 0.00 |
| $H = 32, N_H = $ II, Flow = 16/0 | 99.46 ± 0.03 | 1.94 ± 0.79 | 96.57 ± 1.17 | 94.13 ± 1.47 | 97.82 ± 0.84 | 94.69 ± 1.39 | *100.00 ± 0.00 | *100.00 ± 0.00 |
| $H = 32, N_H = $ II, Flow = 0/4 | 99.26 ± 0.01 | 1.48 ± 0.02 | 98.90 ± 0.05 | 97.98 ± 0.06 | 99.29 ± 0.05 | 98.44 ± 0.09 | 99.94 ± 0.01 | *100.00 ± 0.00 |
| $H = 32, N_H = $ II, Flow = 0/8 | 99.30 ± 0.06 | 1.39 ± 0.09 | 98.89 ± 0.07 | 98.14 ± 0.14 | 99.45 ± 0.06 | 98.64 ± 0.13 | 99.98 ± 0.01 | *100.00 ± 0.00 |
| $H = 32, N_H = $ III, Flow = 8/0 | *99.49 ± 0.01 | 1.20 ± 0.10 | 97.91 ± 0.79 | 73.77 ± 3.06 | 98.73 ± 0.57 | 73.79 ± 2.97 | *100.00 ± 0.00 | *100.00 ± 0.00 |
| $H = 32, N_H = $ III, Flow = 16/0 | 99.44 ± 0.02 | 1.90 ± 0.69 | 96.16 ± 1.53 | 90.95 ± 1.20 | 97.42 ± 1.12 | 91.46 ± 1.24 | *100.00 ± 0.00 | *100.00 ± 0.00 |
| $H = 32, N_H = $ III, Flow = 0/4 | 99.31 ± 0.02 | 1.38 ± 0.03 | 98.89 ± 0.10 | 97.73 ± 0.06 | 99.29 ± 0.11 | 98.41 ± 0.08 | 99.97 ± 0.01 | *100.00 ± 0.00 |
| $H = 32, N_H = $ III, Flow = 0/8 | 99.28 ± 0.04 | 1.33 ± 0.05 | 98.80 ± 0.09 | 98.16 ± 0.05 | 99.27 ± 0.08 | 98.52 ± 0.13 | 99.98 ± 0.01 | *100.00 ± 0.00 |
| $H = 32, N_H = $ IV, Flow = 8/0 | 99.36 ± 0.03 | 1.47 ± 0.05 | 97.97 ± 0.39 | 71.87 ± 1.31 | 99.00 ± 0.10 | 72.28 ± 1.30 | *100.00 ± 0.00 | *100.00 ± 0.00 |
| $H = 32, N_H = $ IV, Flow = 16/0 | 99.38 ± 0.02 | 2.41 ± 0.66 | 95.45 ± 1.73 | 83.06 ± 1.62 | 96.91 ± 1.24 | 83.48 ± 1.65 | *100.00 ± 0.00 | *100.00 ± 0.00 |
| $H = 32, N_H = $ IV, Flow = 0/4 | 99.22 ± 0.04 | 1.50 ± 0.09 | 98.77 ± 0.08 | 97.49 ± 0.11 | 99.27 ± 0.05 | 98.33 ± 0.09 | 99.97 ± 0.01 | *100.00 ± 0.00 |
| $H = 32, N_H = $ IV, Flow = 0/8 | 99.25 ± 0.04 | 1.34 ± 0.07 | 98.62 ± 0.09 | 97.70 ± 0.15 | 99.24 ± 0.06 | 98.09 ± 0.34 | 99.99 ± 0.00 | *100.00 ± 0.00 |
| $H = 32, N_H = $ V, Flow = 8/0 | 99.35 ± 0.02 | 1.43 ± 0.08 | 98.43 ± 0.29 | 64.15 ± 3.79 | 99.22 ± 0.16 | 64.37 ± 3.84 | *100.00 ± 0.00 | *100.00 ± 0.00 |
| $H = 32, N_H = $ V, Flow = 16/0 | 99.33 ± 0.03 | 1.76 ± 0.33 | 97.41 ± 0.93 | 84.15 ± 1.84 | 98.26 ± 0.60 | 85.01 ± 2.01 | *100.00 ± 0.00 | *100.00 ± 0.00 |
| $H = 32, N_H = $ V, Flow = 0/4 | 99.18 ± 0.05 | 1.58 ± 0.08 | 98.61 ± 0.09 | 96.71 ± 0.24 | 99.11 ± 0.07 | 97.34 ± 0.17 | 99.94 ± 0.01 | *100.00 ± 0.00 |
| $H = 32, N_H = $ V, Flow = 0/8 | 99.19 ± 0.03 | 1.58 ± 0.06 | 98.58 ± 0.10 | 96.98 ± 0.15 | 99.19 ± 0.08 | 96.99 ± 0.28 | 99.99 ± 0.00 | *100.00 ± 0.00 |

Table 13: CIFAR10 comparison (<latent dim> – <certainty budget> – <radial layers>/<MAF layers>). Bold and starred number indicate best score among all models.

| | Accuracy | Brier | SVHN Alea. | SVHN Epist. | CelebA Alea. | CelebA Epist. | OODom Alea. | OODom Epist. |
|---|---|---|---|---|---|---|---|---|
| $H = 4$, $N_H = $ I, Flow = 8/0 | 42.73 ± 6.40 | 81.56 ± 2.72 | 67.03 ± 3.15 | 49.84 ± 6.09 | 56.86 ± 2.13 | 39.01 ± 7.65 | 95.33 ± 4.48 | 97.68 ± 2.32 |
| $H = 4$, $N_H = $ I, Flow = 16/0 | 46.39 ± 6.64 | 77.73 ± 3.57 | 66.43 ± 2.39 | 46.63 ± 5.92 | 68.22 ± 3.82 | 41.98 ± 6.52 | 98.93 ± 0.85 | 99.61 ± 0.39 |
| $H = 4$, $N_H = $ I, Flow = 0/4 | 53.79 ± 2.51 | 80.83 ± 4.54 | 59.77 ± 5.82 | 35.96 ± 2.59 | 64.95 ± 1.92 | 48.27 ± 5.85 | 97.76 ± 1.90 | 99.86 ± 0.07 |
| $H = 4$, $N_H = $ I, Flow = 0/8 | 51.11 ± 3.67 | 79.13 ± 2.10 | 58.22 ± 4.19 | 33.22 ± 8.03 | 63.97 ± 3.12 | 42.75 ± 9.47 | 84.62 ± 10.88 | 88.24 ± 11.76 |
| $H = 4$, $N_H = $ II, Flow = 8/0 | 88.02 ± 0.12 | 22.61 ± 0.18 | 75.40 ± 2.82 | 40.98 ± 3.98 | 62.73 ± 6.72 | 35.68 ± 3.66 | 82.17 ± 3.49 | 60.66 ± 7.27 |
| $H = 4$, $N_H = $ II, Flow = 16/0 | 87.80 ± 0.09 | 22.43 ± 0.33 | 78.00 ± 1.68 | 60.02 ± 2.96 | 63.55 ± 2.08 | 46.15 ± 4.20 | 88.55 ± 3.72 | 88.82 ± 2.85 |
| $H = 4$, $N_H = $ II, Flow = 0/4 | 87.10 ± 0.10 | 22.42 ± 0.16 | 83.35 ± 0.72 | 67.89 ± 4.11 | 75.76 ± 1.33 | 63.04 ± 4.04 | 89.20 ± 3.60 | 91.43 ± 3.81 |
| $H = 4$, $N_H = $ II, Flow = 0/8 | 86.28 ± 0.74 | 24.15 ± 1.52 | 83.69 ± 0.48 | 67.20 ± 3.22 | 75.30 ± 1.66 | 63.56 ± 5.25 | 80.92 ± 8.99 | 83.04 ± 7.45 |
| $H = 4$, $N_H = $ III, Flow = 8/0 | 68.54 ± 4.06 | 64.35 ± 3.30 | 71.61 ± 2.06 | 57.05 ± 5.90 | 70.14 ± 2.66 | 51.87 ± 3.56 | 93.39 ± 6.00 | 95.61 ± 3.82 |
| $H = 4$, $N_H = $ III, Flow = 16/0 | 69.41 ± 4.00 | 59.98 ± 4.79 | 74.52 ± 1.11 | 52.59 ± 6.96 | 73.48 ± 2.43 | 51.59 ± 7.09 | 93.43 ± 3.42 | 93.45 ± 4.86 |
| $H = 4$, $N_H = $ III, Flow = 0/4 | 64.80 ± 4.59 | 61.82 ± 5.62 | 67.99 ± 3.37 | 37.00 ± 6.95 | 66.88 ± 5.61 | 38.00 ± 10.01 | 99.70 ± 0.02 | 99.97 ± 0.02 |
| $H = 4$, $N_H = $ III, Flow = 0/8 | 61.49 ± 5.78 | 66.53 ± 6.29 | 57.58 ± 5.84 | 34.37 ± 3.79 | 62.22 ± 6.68 | 39.86 ± 9.87 | 94.73 ± 3.13 | 98.71 ± 1.02 |
| $H = 4$, $N_H = $ IV, Flow = 8/0 | 84.45 ± 0.44 | 38.86 ± 1.13 | 73.65 ± 1.66 | 53.64 ± 1.71 | 71.68 ± 2.13 | 52.30 ± 1.49 | 69.65 ± 5.61 | 66.36 ± 7.76 |
| $H = 4$, $N_H = $ IV, Flow = 16/0 | 83.16 ± 0.83 | 35.51 ± 1.55 | 80.65 ± 2.67 | 59.70 ± 3.43 | 76.69 ± 3.23 | 50.09 ± 2.76 | 80.24 ± 7.59 | 83.05 ± 5.77 |
| $H = 4$, $N_H = $ IV, Flow = 0/4 | 72.13 ± 2.90 | 48.67 ± 3.63 | 72.17 ± 1.95 | 27.43 ± 3.60 | 71.77 ± 2.24 | 27.09 ± 2.25 | 99.22 ± 0.53 | 99.85 ± 0.09 |
| $H = 4$, $N_H = $ IV, Flow = 0/8 | 72.34 ± 1.84 | 51.55 ± 3.08 | 68.02 ± 3.39 | 28.48 ± 2.16 | 69.02 ± 2.91 | 32.72 ± 4.89 | 99.13 ± 0.44 | 98.91 ± 0.68 |
| $H = 4$, $N_H = $ V, Flow = 8/0 | 86.32 ± 0.64 | 32.24 ± 1.38 | 73.15 ± 4.29 | 50.79 ± 3.87 | 66.37 ± 6.81 | 47.11 ± 7.35 | 73.58 ± 7.20 | 63.29 ± 5.99 |
| $H = 4$, $N_H = $ V, Flow = 16/0 | 86.18 ± 0.39 | 27.92 ± 0.56 | 79.67 ± 2.46 | 56.96 ± 4.80 | 66.45 ± 4.29 | 45.29 ± 5.32 | 77.59 ± 10.23 | 80.75 ± 8.14 |
| $H = 4$, $N_H = $ V, Flow = 0/4 | 80.39 ± 0.94 | 37.33 ± 2.23 | 77.81 ± 2.52 | 31.59 ± 3.34 | 74.14 ± 1.57 | 31.93 ± 3.75 | 93.63 ± 5.76 | 93.85 ± 5.61 |
| $H = 4$, $N_H = $ V, Flow = 0/8 | 77.84 ± 0.71 | 42.01 ± 1.03 | 75.27 ± 2.23 | 40.87 ± 6.42 | 73.46 ± 1.81 | 33.18 ± 5.09 | 98.59 ± 0.72 | 99.08 ± 0.61 |
| $H = 16$, $N_H = $ I, Flow = 8/0 | 82.83 ± 0.84 | 30.89 ± 1.71 | 81.35 ± 0.60 | 65.51 ± 2.65 | 75.48 ± 1.65 | 61.89 ± 3.89 | 99.73 ± 0.24 | 99.82 ± 0.18 |
| $H = 16$, $N_H = $ I, Flow = 16/0 | 84.83 ± 0.60 | 26.24 ± 1.06 | 82.39 ± 1.11 | 75.54 ± 2.52 | 75.30 ± 0.72 | 69.12 ± 2.15 | 99.90 ± 0.06 | 99.96 ± 0.04 |
| $H = 16$, $N_H = $ I, Flow = 0/4 | 83.71 ± 1.37 | 27.35 ± 2.34 | 81.38 ± 0.94 | 56.29 ± 4.77 | 75.88 ± 0.51 | 51.73 ± 4.32 | 99.17 ± 0.65 | 99.92 ± 0.07 |
| $H = 16$, $N_H = $ I, Flow = 0/8 | 84.13 ± 0.53 | 27.02 ± 1.02 | 80.24 ± 1.88 | 67.03 ± 2.59 | 74.59 ± 0.68 | 58.67 ± 8.57 | 99.93 ± 0.01 | *100.00 ± 0.00 |
| $H = 16$, $N_H = $ II, Flow = 8/0 | 86.82 ± 0.46 | 22.45 ± 0.88 | 83.73 ± 1.26 | 56.60 ± 3.15 | 76.98 ± 1.67 | 63.55 ± 2.26 | 85.41 ± 11.05 | 87.59 ± 7.40 |
| $H = 16$, $N_H = $ II, Flow = 16/0 | 86.40 ± 0.46 | 23.06 ± 0.82 | 83.60 ± 0.96 | 74.04 ± 2.78 | 76.87 ± 0.92 | 74.00 ± 1.92 | 82.09 ± 10.05 | 92.84 ± 3.70 |
| $H = 16$, $N_H = $ II, Flow = 0/4 | 88.02 ± 0.12 | 19.83 ± 0.21 | 81.19 ± 0.77 | 72.46 ± 1.88 | 74.19 ± 1.54 | 65.32 ± 2.02 | 99.14 ± 0.59 | 99.90 ± 0.07 |
| $H = 16$, $N_H = $ II, Flow = 0/8 | 87.73 ± 0.11 | 19.97 ± 0.33 | 83.29 ± 0.70 | 71.93 ± 1.90 | 74.14 ± 2.13 | 63.20 ± 3.16 | 99.94 ± 0.01 | *100.00 ± 0.00 |
| $H = 16$, $N_H = $ III, Flow = 8/0 | 86.54 ± 0.55 | 23.44 ± 1.10 | 80.97 ± 1.57 | 59.46 ± 1.33 | 73.31 ± 2.36 | 58.10 ± 3.38 | 93.95 ± 2.43 | 91.07 ± 3.26 |
| $H = 16$, $N_H = $ III, Flow = 16/0 | 86.89 ± 0.20 | 22.07 ± 0.41 | 83.11 ± 0.43 | 72.37 ± 1.29 | 75.52 ± 0.83 | 72.66 ± 3.46 | 96.20 ± 2.51 | 98.21 ± 1.06 |
| $H = 16$, $N_H = $ III, Flow = 0/4 | 88.16 ± 0.16 | 19.54 ± 0.22 | 82.50 ± 1.29 | 62.54 ± 1.59 | 74.62 ± 0.98 | 57.76 ± 2.44 | 99.89 ± 0.01 | *100.00 ± 0.00 |
| $H = 16$, $N_H = $ III, Flow = 0/8 | 87.67 ± 0.17 | 20.23 ± 0.38 | 84.03 ± 0.93 | 70.91 ± 1.97 | 73.43 ± 1.33 | 61.20 ± 1.55 | 99.90 ± 0.03 | 99.99 ± 0.00 |
| $H = 16$, $N_H = $ IV, Flow = 8/0 | 87.97 ± 0.07 | 19.98 ± 0.17 | *84.42 ± 0.82 | 62.10 ± 0.56 | 72.25 ± 1.85 | 65.17 ± 5.34 | 90.96 ± 4.50 | 81.63 ± 9.18 |
| $H = 16$, $N_H = $ IV, Flow = 16/0 | 87.90 ± 0.16 | 19.99 ± 0.46 | 82.29 ± 1.11 | *77.83 ± 1.22 | 76.01 ± 1.18 | *76.87 ± 3.38 | 93.67 ± 3.03 | 94.90 ± 3.09 |
| $H = 16$, $N_H = $ IV, Flow = 0/4 | 87.92 ± 0.22 | 19.93 ± 0.38 | 82.26 ± 1.22 | 67.03 ± 2.75 | 73.28 ± 1.25 | 63.06 ± 2.78 | 99.91 ± 0.02 | *100.00 ± 0.00 |
| $H = 16$, $N_H = $ IV, Flow = 0/8 | 87.90 ± 0.19 | 19.70 ± 0.28 | 81.72 ± 0.44 | 64.76 ± 4.18 | 74.90 ± 0.94 | 66.46 ± 3.04 | 99.82 ± 0.12 | 99.99 ± 0.01 |
| $H = 16$, $N_H = $ V, Flow = 8/0 | 88.17 ± 0.17 | 19.78 ± 0.29 | 83.67 ± 0.66 | 56.34 ± 0.85 | 74.32 ± 2.32 | 62.47 ± 1.08 | 95.06 ± 1.51 | 82.09 ± 6.85 |
| $H = 16$, $N_H = $ V, Flow = 16/0 | 88.17 ± 0.16 | 19.81 ± 0.34 | 81.76 ± 1.19 | 68.45 ± 1.60 | 72.98 ± 1.93 | 71.08 ± 4.11 | 83.74 ± 6.25 | 86.85 ± 3.37 |
| $H = 16$, $N_H = $ V, Flow = 0/4 | 88.24 ± 0.12 | 19.30 ± 0.25 | 83.57 ± 0.67 | 66.44 ± 3.12 | 74.16 ± 1.94 | 63.04 ± 2.90 | 99.10 ± 0.84 | 99.97 ± 0.02 |
| $H = 16$, $N_H = $ V, Flow = 0/8 | 88.10 ± 0.20 | 19.32 ± 0.40 | 81.22 ± 1.29 | 71.88 ± 2.05 | 75.42 ± 0.96 | 66.06 ± 2.02 | 99.74 ± 0.18 | 99.99 ± 0.00 |
| $H = 32$, $N_H = $ I, Flow = 8/0 | 10.05 ± 0.28 | 95.02 ± 0.03 | 27.35 ± 1.29 | 23.46 ± 1.45 | 35.30 ± 1.88 | 34.31 ± 1.47 | 97.06 ± 0.64 | *100.00 ± 0.00 |
| $H = 32$, $N_H = $ I, Flow = 16/0 | 83.94 ± 0.61 | 27.83 ± 1.22 | 81.71 ± 0.62 | 74.15 ± 2.21 | 75.62 ± 1.48 | 75.68 ± 4.15 | 99.94 ± 0.02 | 99.97 ± 0.03 |
| $H = 32$, $N_H = $ I, Flow = 0/4 | 85.30 ± 0.86 | 24.67 ± 1.56 | 82.70 ± 1.54 | 66.68 ± 2.56 | 76.15 ± 1.88 | 62.01 ± 0.96 | 99.93 ± 0.02 | *100.00 ± 0.00 |
| $H = 32$, $N_H = $ I, Flow = 0/8 | 85.58 ± 0.32 | 24.04 ± 0.73 | 82.77 ± 1.10 | 57.03 ± 2.71 | 75.48 ± 1.25 | 56.44 ± 1.77 | 99.84 ± 0.11 | *100.00 ± 0.00 |
| $H = 32$, $N_H = $ II, Flow = 8/0 | 86.11 ± 0.41 | 23.61 ± 0.82 | 83.38 ± 0.71 | 52.99 ± 2.55 | 76.14 ± 1.43 | 60.70 ± 1.29 | 95.80 ± 2.55 | 55.44 ± 2.79 |
| $H = 32$, $N_H = $ II, Flow = 16/0 | 86.28 ± 0.31 | 23.11 ± 0.54 | 83.00 ± 0.53 | 63.42 ± 3.35 | 73.87 ± 1.19 | 72.90 ± 3.19 | 97.87 ± 2.04 | 99.92 ± 0.05 |
| $H = 32$, $N_H = $ II, Flow = 0/4 | 87.98 ± 0.12 | 19.62 ± 0.22 | 82.72 ± 0.77 | 58.53 ± 4.37 | 74.45 ± 1.96 | 57.56 ± 4.88 | 99.93 ± 0.04 | *100.00 ± 0.00 |
| $H = 32$, $N_H = $ II, Flow = 0/8 | 87.41 ± 0.45 | 20.73 ± 0.78 | 81.91 ± 1.56 | 63.56 ± 2.61 | 75.30 ± 0.85 | 58.86 ± 3.76 | 99.97 ± 0.01 | *100.00 ± 0.00 |
| $H = 32$, $N_H = $ III, Flow = 8/0 | 86.65 ± 0.08 | 22.53 ± 0.18 | 82.38 ± 0.45 | 51.04 ± 1.13 | 74.93 ± 0.97 | 62.11 ± 1.92 | 95.20 ± 2.24 | 92.17 ± 5.43 |
| $H = 32$, $N_H = $ III, Flow = 16/0 | 85.71 ± 0.47 | 24.29 ± 0.88 | 82.65 ± 0.59 | 66.63 ± 3.54 | 76.55 ± 1.22 | 73.27 ± 1.09 | 98.27 ± 1.24 | 98.75 ± 1.14 |
| $H = 32$, $N_H = $ III, Flow = 0/4 | 88.05 ± 0.17 | 19.54 ± 0.34 | 81.85 ± 0.94 | 63.19 ± 2.75 | 76.21 ± 0.84 | 61.44 ± 4.30 | 99.75 ± 0.21 | *100.00 ± 0.00 |
| $H = 32$, $N_H = $ III, Flow = 0/8 | 88.23 ± 0.37 | 19.48 ± 0.62 | 84.08 ± 1.42 | 64.08 ± 3.71 | *78.26 ± 1.27 | 64.08 ± 1.55 | *99.98 ± 0.01 | *100.00 ± 0.00 |
| $H = 32$, $N_H = $ IV, Flow = 8/0 | 88.02 ± 0.19 | 19.78 ± 0.32 | 83.24 ± 0.65 | 50.51 ± 4.65 | 77.52 ± 1.20 | 60.08 ± 2.16 | 95.24 ± 2.10 | 91.41 ± 3.61 |
| $H = 32$, $N_H = $ IV, Flow = 16/0 | 88.29 ± 0.14 | 19.36 ± 0.31 | 82.62 ± 1.72 | 59.43 ± 5.41 | 75.52 ± 1.33 | 59.20 ± 5.22 | 97.56 ± 1.00 | 99.57 ± 0.39 |
| $H = 32$, $N_H = $ IV, Flow = 0/4 | 87.83 ± 0.14 | 19.92 ± 0.24 | 83.23 ± 0.71 | 57.29 ± 5.16 | 74.99 ± 1.71 | 65.64 ± 3.16 | 99.93 ± 0.04 | *100.00 ± 0.00 |
| $H = 32$, $N_H = $ IV, Flow = 0/8 | 88.24 ± 0.28 | *19.25 ± 0.44 | 82.82 ± 1.17 | 64.69 ± 2.96 | 73.58 ± 2.12 | 60.43 ± 3.12 | 99.27 ± 0.43 | 99.99 ± 0.01 |
| $H = 32$, $N_H = $ V, Flow = 8/0 | 88.25 ± 0.11 | 19.43 ± 0.25 | 83.02 ± 0.60 | 48.85 ± 3.13 | 73.55 ± 2.21 | 54.24 ± 3.53 | 94.02 ± 2.56 | 83.94 ± 5.81 |
| $H = 32$, $N_H = $ V, Flow = 16/0 | 88.30 ± 0.17 | 19.41 ± 0.34 | 83.35 ± 1.37 | 64.63 ± 3.58 | 75.89 ± 2.13 | 72.53 ± 3.19 | 95.68 ± 0.75 | 95.89 ± 2.84 |
| $H = 32$, $N_H = $ V, Flow = 0/4 | *88.41 ± 0.15 | 19.34 ± 0.26 | 84.03 ± 0.94 | 59.12 ± 3.95 | 74.31 ± 1.96 | 63.33 ± 1.89 | 99.73 ± 0.13 | 99.99 ± 0.00 |
| $H = 32$, $N_H = $ V, Flow = 0/8 | 88.26 ± 0.10 | 19.28 ± 0.10 | 83.68 ± 0.69 | 60.66 ± 2.92 | 73.08 ± 1.57 | 61.10 ± 5.28 | 99.28 ± 0.28 | *100.00 ± 0.00 |

Table 14: Bike Sharing (Normal $\mathcal{N}$) comparison (<latent dim> – <certainty budget> – <radial layers>/<MAF layers>). Bold and starred number indicate best score among all models.

| | RMSE | Calibration | Winter Epist. | Spring Epist. | Autumn Epist. | OODom Epist. |
|---|---|---|---|---|---|---|
| $H$ = 4, $N_H$ = I, Flow = 8/0 | $60.12 \pm 4.17$ | $21.15 \pm 2.06$ | $21.11 \pm 2.59$ | $14.97 \pm 0.48$ | $17.61 \pm 1.02$ | *$100.00 \pm 0.00$ |
| $H$ = 4, $N_H$ = I, Flow = 16/0 | $53.68 \pm 3.45$ | $21.92 \pm 1.65$ | $26.80 \pm 2.61$ | $17.54 \pm 0.87$ | $17.96 \pm 1.20$ | *$100.00 \pm 0.00$ |
| $H$ = 4, $N_H$ = I, Flow = 0/4 | $83.51 \pm 8.56$ | $23.94 \pm 1.94$ | $30.25 \pm 6.17$ | $17.49 \pm 1.50$ | $18.79 \pm 1.30$ | *$100.00 \pm 0.00$ |
| $H$ = 4, $N_H$ = I, Flow = 0/8 | $75.16 \pm 8.97$ | $25.93 \pm 1.01$ | $29.86 \pm 4.07$ | $17.71 \pm 0.78$ | $17.98 \pm 0.41$ | *$100.00 \pm 0.00$ |
| $H$ = 4, $N_H$ = II, Flow = 8/0 | $53.90 \pm 3.72$ | $22.76 \pm 0.93$ | $24.59 \pm 2.43$ | $16.69 \pm 0.76$ | $18.30 \pm 1.10$ | *$100.00 \pm 0.00$ |
| $H$ = 4, $N_H$ = II, Flow = 16/0 | $56.08 \pm 4.96$ | $26.30 \pm 1.32$ | $26.93 \pm 4.34$ | $17.11 \pm 1.06$ | $20.41 \pm 1.84$ | *$100.00 \pm 0.00$ |
| $H$ = 4, $N_H$ = II, Flow = 0/4 | $65.03 \pm 8.07$ | $23.10 \pm 1.68$ | $27.99 \pm 3.46$ | $17.04 \pm 1.06$ | $22.55 \pm 2.60$ | *$100.00 \pm 0.00$ |
| $H$ = 4, $N_H$ = II, Flow = 0/8 | $74.25 \pm 7.89$ | $26.51 \pm 1.09$ | $31.66 \pm 3.21$ | $18.31 \pm 0.90$ | $20.35 \pm 2.26$ | *$100.00 \pm 0.00$ |
| $H$ = 4, $N_H$ = III, Flow = 8/0 | $51.68 \pm 1.25$ | $19.04 \pm 1.75$ | $25.06 \pm 2.21$ | $17.05 \pm 0.47$ | $17.44 \pm 1.50$ | *$100.00 \pm 0.00$ |
| $H$ = 4, $N_H$ = III, Flow = 16/0 | $55.23 \pm 4.45$ | $19.04 \pm 1.13$ | $27.77 \pm 2.88$ | $16.69 \pm 0.60$ | $18.74 \pm 0.19$ | *$100.00 \pm 0.00$ |
| $H$ = 4, $N_H$ = III, Flow = 0/4 | $79.25 \pm 11.47$ | $25.54 \pm 0.94$ | $26.66 \pm 2.88$ | $16.73 \pm 0.92$ | $19.28 \pm 0.68$ | *$100.00 \pm 0.00$ |
| $H$ = 4, $N_H$ = III, Flow = 0/8 | $92.99 \pm 3.68$ | $26.48 \pm 0.41$ | $28.84 \pm 1.87$ | $17.26 \pm 0.43$ | $22.35 \pm 2.26$ | *$100.00 \pm 0.00$ |
| $H$ = 4, $N_H$ = IV, Flow = 8/0 | $51.71 \pm 1.50$ | $18.36 \pm 2.24$ | $23.72 \pm 2.95$ | $16.19 \pm 0.75$ | $16.77 \pm 0.38$ | *$100.00 \pm 0.00$ |
| $H$ = 4, $N_H$ = IV, Flow = 16/0 | $60.81 \pm 5.56$ | $21.01 \pm 1.35$ | $22.02 \pm 1.85$ | $15.43 \pm 0.48$ | $17.31 \pm 0.61$ | *$100.00 \pm 0.00$ |
| $H$ = 4, $N_H$ = IV, Flow = 0/4 | $57.74 \pm 4.29$ | $22.78 \pm 1.02$ | $26.75 \pm 1.99$ | $17.57 \pm 0.36$ | $18.85 \pm 1.22$ | *$100.00 \pm 0.00$ |
| $H$ = 4, $N_H$ = IV, Flow = 0/8 | $78.64 \pm 8.29$ | $23.29 \pm 1.91$ | $34.19 \pm 2.72$ | $19.14 \pm 0.86$ | $20.46 \pm 0.70$ | *$100.00 \pm 0.00$ |
| $H$ = 4, $N_H$ = V, Flow = 8/0 | $53.74 \pm 2.95$ | $23.27 \pm 1.51$ | $30.83 \pm 4.89$ | $17.34 \pm 0.25$ | $20.26 \pm 1.24$ | *$100.00 \pm 0.00$ |
| $H$ = 4, $N_H$ = V, Flow = 16/0 | $56.01 \pm 2.32$ | $23.35 \pm 0.94$ | $27.03 \pm 2.64$ | $17.52 \pm 1.04$ | $17.07 \pm 0.81$ | *$100.00 \pm 0.00$ |
| $H$ = 4, $N_H$ = V, Flow = 0/4 | $83.87 \pm 8.67$ | $24.20 \pm 2.03$ | $27.64 \pm 3.59$ | $16.48 \pm 0.78$ | $21.24 \pm 2.11$ | *$100.00 \pm 0.00$ |
| $H$ = 4, $N_H$ = V, Flow = 0/8 | $90.87 \pm 5.11$ | $22.54 \pm 0.99$ | $30.61 \pm 2.20$ | $17.46 \pm 0.72$ | $19.84 \pm 0.93$ | *$100.00 \pm 0.00$ |
| $H$ = 16, $N_H$ = I, Flow = 8/0 | $58.52 \pm 5.08$ | $9.86 \pm 3.10$ | $22.85 \pm 3.51$ | $15.57 \pm 1.36$ | $19.51 \pm 3.12$ | *$100.00 \pm 0.00$ |
| $H$ = 16, $N_H$ = I, Flow = 16/0 | $58.33 \pm 4.80$ | $10.29 \pm 4.02$ | $38.20 \pm 3.14$ | $19.10 \pm 0.91$ | $21.55 \pm 1.90$ | *$100.00 \pm 0.00$ |
| $H$ = 16, $N_H$ = I, Flow = 0/4 | $52.02 \pm 1.93$ | $7.98 \pm 2.43$ | $48.51 \pm 5.78$ | $22.11 \pm 1.76$ | $31.46 \pm 3.21$ | *$100.00 \pm 0.00$ |
| $H$ = 16, $N_H$ = I, Flow = 0/8 | $59.43 \pm 5.49$ | $9.20 \pm 3.32$ | $55.93 \pm 8.21$ | $25.67 \pm 2.12$ | $32.76 \pm 5.61$ | *$100.00 \pm 0.00$ |
| $H$ = 16, $N_H$ = II, Flow = 8/0 | $58.02 \pm 2.35$ | $4.25 \pm 1.32$ | $23.71 \pm 1.27$ | $16.69 \pm 0.69$ | $17.95 \pm 0.53$ | *$100.00 \pm 0.00$ |
| $H$ = 16, $N_H$ = II, Flow = 16/0 | $58.92 \pm 6.56$ | $4.69 \pm 1.23$ | $33.81 \pm 5.95$ | $18.33 \pm 1.66$ | $21.59 \pm 2.26$ | *$100.00 \pm 0.00$ |
| $H$ = 16, $N_H$ = II, Flow = 0/4 | $51.87 \pm 2.45$ | $8.01 \pm 1.89$ | $48.39 \pm 10.55$ | $23.12 \pm 3.23$ | $25.23 \pm 2.61$ | *$100.00 \pm 0.00$ |
| $H$ = 16, $N_H$ = II, Flow = 0/8 | $58.00 \pm 5.68$ | $4.57 \pm 1.15$ | $51.38 \pm 8.78$ | $24.60 \pm 2.22$ | $26.31 \pm 3.59$ | *$100.00 \pm 0.00$ |
| $H$ = 16, $N_H$ = III, Flow = 8/0 | $60.73 \pm 4.01$ | $4.45 \pm 0.89$ | $30.21 \pm 2.21$ | $17.74 \pm 0.74$ | $20.59 \pm 2.19$ | *$100.00 \pm 0.00$ |
| $H$ = 16, $N_H$ = III, Flow = 16/0 | $59.87 \pm 5.68$ | $5.70 \pm 1.47$ | $36.65 \pm 6.92$ | $18.95 \pm 1.61$ | $21.12 \pm 3.18$ | *$100.00 \pm 0.00$ |
| $H$ = 16, $N_H$ = III, Flow = 0/4 | $58.80 \pm 4.49$ | $7.34 \pm 2.08$ | $56.72 \pm 3.51$ | $25.77 \pm 1.72$ | $27.21 \pm 3.06$ | *$100.00 \pm 0.00$ |
| $H$ = 16, $N_H$ = III, Flow = 0/8 | $54.08 \pm 2.95$ | $8.56 \pm 2.70$ | $55.37 \pm 8.22$ | $25.82 \pm 2.32$ | $31.20 \pm 4.73$ | *$100.00 \pm 0.00$ |
| $H$ = 16, $N_H$ = IV, Flow = 8/0 | $52.49 \pm 1.77$ | $5.54 \pm 1.22$ | $33.26 \pm 4.84$ | $16.93 \pm 1.23$ | $22.32 \pm 1.87$ | *$100.00 \pm 0.00$ |
| $H$ = 16, $N_H$ = IV, Flow = 16/0 | $53.29 \pm 3.17$ | $4.15 \pm 1.67$ | $30.48 \pm 4.05$ | $17.72 \pm 1.15$ | $20.31 \pm 1.66$ | *$100.00 \pm 0.00$ |
| $H$ = 16, $N_H$ = IV, Flow = 0/4 | $56.11 \pm 3.47$ | $5.48 \pm 1.63$ | $54.10 \pm 10.31$ | $23.80 \pm 3.44$ | $25.87 \pm 2.60$ | *$100.00 \pm 0.00$ |
| $H$ = 16, $N_H$ = IV, Flow = 0/8 | $56.03 \pm 2.51$ | $3.74 \pm 1.59$ | $55.41 \pm 7.83$ | $23.78 \pm 2.41$ | $30.82 \pm 2.93$ | *$100.00 \pm 0.00$ |
| $H$ = 16, $N_H$ = V, Flow = 8/0 | $55.11 \pm 3.53$ | $6.72 \pm 1.17$ | $39.26 \pm 7.71$ | $19.03 \pm 1.41$ | $23.03 \pm 3.74$ | *$100.00 \pm 0.00$ |
| $H$ = 16, $N_H$ = V, Flow = 16/0 | $57.19 \pm 4.36$ | $6.13 \pm 1.77$ | $30.47 \pm 3.14$ | $17.24 \pm 1.30$ | $22.40 \pm 3.35$ | *$100.00 \pm 0.00$ |
| $H$ = 16, $N_H$ = V, Flow = 0/4 | $54.33 \pm 2.57$ | $4.31 \pm 1.03$ | $40.28 \pm 7.97$ | $18.99 \pm 1.47$ | $24.85 \pm 3.95$ | *$100.00 \pm 0.00$ |
| $H$ = 16, $N_H$ = V, Flow = 0/8 | $55.97 \pm 4.00$ | $2.09 \pm 0.55$ | $57.11 \pm 9.02$ | $26.22 \pm 3.44$ | $25.91 \pm 4.23$ | *$100.00 \pm 0.00$ |
| $H$ = 32, $N_H$ = I, Flow = 8/0 | $55.82 \pm 1.49$ | $3.19 \pm 0.78$ | $33.14 \pm 4.82$ | $18.14 \pm 1.42$ | $19.36 \pm 0.80$ | *$100.00 \pm 0.00$ |
| $H$ = 32, $N_H$ = I, Flow = 16/0 | $60.20 \pm 4.97$ | $3.11 \pm 0.78$ | $40.89 \pm 5.92$ | $19.51 \pm 1.50$ | $19.08 \pm 1.31$ | *$100.00 \pm 0.00$ |
| $H$ = 32, $N_H$ = I, Flow = 0/4 | $59.49 \pm 2.94$ | $2.90 \pm 0.54$ | $40.57 \pm 8.66$ | $19.14 \pm 2.25$ | $26.81 \pm 2.90$ | *$100.00 \pm 0.00$ |
| $H$ = 32, $N_H$ = I, Flow = 0/8 | $61.86 \pm 4.98$ | $1.94 \pm 0.33$ | *$72.07 \pm 8.34$ | *$28.46 \pm 1.65$ | $32.84 \pm 3.41$ | *$100.00 \pm 0.00$ |
| $H$ = 32, $N_H$ = II, Flow = 8/0 | $57.46 \pm 3.24$ | $5.08 \pm 1.96$ | $30.09 \pm 2.78$ | $17.55 \pm 0.67$ | $18.25 \pm 0.78$ | *$100.00 \pm 0.00$ |
| $H$ = 32, $N_H$ = II, Flow = 16/0 | $55.34 \pm 2.78$ | $2.15 \pm 0.41$ | $35.14 \pm 2.49$ | $19.24 \pm 1.00$ | $20.53 \pm 2.12$ | *$100.00 \pm 0.00$ |
| $H$ = 32, $N_H$ = II, Flow = 0/4 | $58.50 \pm 7.43$ | $2.23 \pm 0.29$ | $57.94 \pm 5.29$ | $24.43 \pm 0.60$ | $29.14 \pm 3.00$ | *$100.00 \pm 0.00$ |
| $H$ = 32, $N_H$ = II, Flow = 0/8 | $54.68 \pm 4.01$ | $2.26 \pm 0.60$ | $45.24 \pm 5.14$ | $19.82 \pm 1.48$ | $29.27 \pm 2.28$ | *$100.00 \pm 0.00$ |
| $H$ = 32, $N_H$ = III, Flow = 8/0 | $53.35 \pm 3.39$ | $2.88 \pm 0.81$ | $32.52 \pm 1.58$ | $18.28 \pm 1.29$ | $20.08 \pm 1.05$ | *$100.00 \pm 0.00$ |
| $H$ = 32, $N_H$ = III, Flow = 16/0 | $54.91 \pm 2.39$ | $4.51 \pm 0.92$ | $35.21 \pm 4.26$ | $18.68 \pm 1.00$ | $19.81 \pm 1.46$ | *$100.00 \pm 0.00$ |
| $H$ = 32, $N_H$ = III, Flow = 0/4 | $58.84 \pm 1.61$ | *$1.49 \pm 0.20$ | $50.90 \pm 11.50$ | $21.83 \pm 2.72$ | $33.23 \pm 5.33$ | *$100.00 \pm 0.00$ |
| $H$ = 32, $N_H$ = III, Flow = 0/8 | *$51.12 \pm 1.93$ | $2.17 \pm 0.35$ | $55.22 \pm 5.21$ | $24.55 \pm 2.43$ | $28.35 \pm 2.18$ | *$100.00 \pm 0.00$ |
| $H$ = 32, $N_H$ = IV, Flow = 8/0 | $52.58 \pm 2.37$ | $6.09 \pm 1.30$ | $28.88 \pm 5.31$ | $16.77 \pm 1.24$ | $20.28 \pm 2.63$ | *$100.00 \pm 0.00$ |
| $H$ = 32, $N_H$ = IV, Flow = 16/0 | $58.58 \pm 5.19$ | $5.05 \pm 1.43$ | $28.78 \pm 2.57$ | $16.47 \pm 0.83$ | $21.19 \pm 1.64$ | *$100.00 \pm 0.00$ |
| $H$ = 32, $N_H$ = IV, Flow = 0/4 | $51.88 \pm 1.57$ | $2.72 \pm 0.74$ | $55.32 \pm 7.42$ | $24.11 \pm 2.10$ | *$35.59 \pm 3.99$ | *$100.00 \pm 0.00$ |
| $H$ = 32, $N_H$ = IV, Flow = 0/8 | $53.84 \pm 2.82$ | $1.85 \pm 0.33$ | $49.44 \pm 3.65$ | $21.06 \pm 2.11$ | $30.14 \pm 2.12$ | *$100.00 \pm 0.00$ |
| $H$ = 32, $N_H$ = V, Flow = 8/0 | $53.52 \pm 1.64$ | $8.24 \pm 0.64$ | $30.51 \pm 2.22$ | $18.50 \pm 0.67$ | $20.98 \pm 2.00$ | *$100.00 \pm 0.00$ |
| $H$ = 32, $N_H$ = V, Flow = 16/0 | $58.21 \pm 9.11$ | $6.91 \pm 1.16$ | $35.75 \pm 3.59$ | $18.07 \pm 0.91$ | $19.09 \pm 1.29$ | *$100.00 \pm 0.00$ |
| $H$ = 32, $N_H$ = V, Flow = 0/4 | $56.22 \pm 4.42$ | $1.98 \pm 0.45$ | $40.97 \pm 5.06$ | $20.21 \pm 1.98$ | $23.40 \pm 1.69$ | *$100.00 \pm 0.00$ |
| $H$ = 32, $N_H$ = V, Flow = 0/8 | $54.06 \pm 3.18$ | $2.09 \pm 0.32$ | $35.75 \pm 7.51$ | $18.86 \pm 2.13$ | $22.27 \pm 1.83$ | *$100.00 \pm 0.00$ |

Table 15: Bike Sharing (Poisson Poi) comparison (<latent dim> – <certainty budget> – <radial layers>/<MAF layers>). Bold and starred number indicate best score among all models.

| | RMSE | Winter Epist. | Spring Epist. | Autumn Epist. | OODom Epist. |
|---|---|---|---|---|---|
| $H$ = 4, $N_H$ = I, Flow = 8/0 | 937.56 ± 238.40 | 33.50 ± 4.83 | 18.04 ± 0.58 | 20.44 ± 1.53 | *100.00 ± 0.00 |
| $H$ = 4, $N_H$ = I, Flow = 16/0 | 777.22 ± 328.71 | 22.53 ± 2.16 | 17.00 ± 0.74 | 18.30 ± 1.25 | *100.00 ± 0.00 |
| $H$ = 4, $N_H$ = I, Flow = 0/4 | 33780.69 ± 19607.73 | 45.22 ± 7.96 | 21.35 ± 2.09 | 37.89 ± 4.13 | *100.00 ± 0.00 |
| $H$ = 4, $N_H$ = I, Flow = 0/8 | 22686.67 ± 5815.18 | 53.51 ± 5.83 | 21.80 ± 1.14 | 39.25 ± 2.59 | *100.00 ± 0.00 |
| $H$ = 4, $N_H$ = II, Flow = 8/0 | 60383.07 ± 21389.07 | 15.20 ± 0.57 | 13.50 ± 0.22 | 15.28 ± 0.77 | *100.00 ± 0.00 |
| $H$ = 4, $N_H$ = II, Flow = 16/0 | 32982.63 ± 17819.36 | 33.09 ± 8.79 | 17.55 ± 2.25 | 19.17 ± 1.19 | *100.00 ± 0.00 |
| $H$ = 4, $N_H$ = II, Flow = 0/4 | 21467.31 ± 17019.59 | 44.60 ± 8.61 | 20.27 ± 1.97 | 33.25 ± 8.69 | *100.00 ± 0.00 |
| $H$ = 4, $N_H$ = II, Flow = 0/8 | 11231.30 ± 4784.62 | 42.64 ± 7.58 | 21.06 ± 2.63 | 23.93 ± 2.05 | *100.00 ± 0.00 |
| $H$ = 4, $N_H$ = III, Flow = 8/0 | 2048.78 ± 538.12 | 22.84 ± 1.90 | 16.59 ± 0.73 | 17.21 ± 0.50 | *100.00 ± 0.00 |
| $H$ = 4, $N_H$ = III, Flow = 16/0 | 5181.92 ± 3581.91 | 25.42 ± 2.91 | 15.24 ± 0.53 | 17.00 ± 1.34 | *100.00 ± 0.00 |
| $H$ = 4, $N_H$ = III, Flow = 0/4 | 35092.52 ± 10813.54 | 47.42 ± 5.98 | 22.49 ± 1.88 | 28.50 ± 1.52 | *100.00 ± 0.00 |
| $H$ = 4, $N_H$ = III, Flow = 0/8 | 86946.86 ± 42792.69 | 53.87 ± 7.18 | 22.87 ± 2.17 | 33.78 ± 5.12 | *100.00 ± 0.00 |
| $H$ = 4, $N_H$ = IV, Flow = 8/0 | 10255.94 ± 6207.90 | 19.89 ± 1.82 | 14.93 ± 0.30 | 16.38 ± 0.57 | *100.00 ± 0.00 |
| $H$ = 4, $N_H$ = IV, Flow = 16/0 | 6665.70 ± 3160.53 | 29.95 ± 6.24 | 17.66 ± 1.06 | 17.33 ± 0.46 | *100.00 ± 0.00 |
| $H$ = 4, $N_H$ = IV, Flow = 0/4 | 119600.14 ± 85229.53 | 35.15 ± 5.20 | 19.58 ± 2.37 | 27.15 ± 2.87 | *100.00 ± 0.00 |
| $H$ = 4, $N_H$ = IV, Flow = 0/8 | 132950.39 ± 98199.93 | 56.85 ± 8.13 | 24.17 ± 1.99 | 40.23 ± 5.69 | *100.00 ± 0.00 |
| $H$ = 4, $N_H$ = V, Flow = 8/0 | 131051.70 ± 124947.53 | 23.89 ± 2.55 | 15.68 ± 0.71 | 17.94 ± 1.34 | *100.00 ± 0.00 |
| $H$ = 4, $N_H$ = V, Flow = 16/0 | 16481.96 ± 7339.53 | 26.55 ± 1.59 | 16.40 ± 0.25 | 26.36 ± 5.15 | *100.00 ± 0.00 |
| $H$ = 4, $N_H$ = V, Flow = 0/4 | 28238.41 ± 10202.38 | 39.68 ± 9.73 | 19.04 ± 2.13 | 29.09 ± 3.57 | *100.00 ± 0.00 |
| $H$ = 4, $N_H$ = V, Flow = 0/8 | 27167.10 ± 9698.30 | 46.59 ± 6.57 | 23.03 ± 1.22 | 27.77 ± 5.03 | *100.00 ± 0.00 |
| $H$ = 16, $N_H$ = I, Flow = 8/0 | 633.14 ± 237.64 | 35.20 ± 5.92 | 18.16 ± 0.91 | 22.16 ± 2.43 | *100.00 ± 0.00 |
| $H$ = 16, $N_H$ = I, Flow = 16/0 | 408.28 ± 246.12 | 45.32 ± 4.53 | 19.96 ± 1.66 | 34.91 ± 6.35 | *100.00 ± 0.00 |
| $H$ = 16, $N_H$ = I, Flow = 0/4 | 276.23 ± 151.72 | 64.86 ± 8.92 | 26.40 ± 3.85 | 37.05 ± 5.43 | *100.00 ± 0.00 |
| $H$ = 16, $N_H$ = I, Flow = 0/8 | 262.91 ± 126.12 | 80.30 ± 4.85 | 32.12 ± 2.86 | 38.83 ± 3.14 | *100.00 ± 0.00 |
| $H$ = 16, $N_H$ = II, Flow = 8/0 | 1325.94 ± 79.27 | 30.06 ± 3.95 | 18.63 ± 1.33 | 21.43 ± 1.82 | *100.00 ± 0.00 |
| $H$ = 16, $N_H$ = II, Flow = 16/0 | 1042.48 ± 413.69 | 45.96 ± 5.13 | 20.63 ± 1.63 | 24.25 ± 1.57 | *100.00 ± 0.00 |
| $H$ = 16, $N_H$ = II, Flow = 0/4 | 129.87 ± 79.26 | 71.03 ± 5.39 | 34.02 ± 4.65 | 36.16 ± 5.75 | *100.00 ± 0.00 |
| $H$ = 16, $N_H$ = II, Flow = 0/8 | 182.97 ± 129.05 | 81.19 ± 6.60 | 36.17 ± 5.02 | *43.97 ± 7.12 | *100.00 ± 0.00 |
| $H$ = 16, $N_H$ = III, Flow = 8/0 | 1233.19 ± 76.60 | 34.22 ± 4.42 | 19.09 ± 1.47 | 25.34 ± 1.63 | *100.00 ± 0.00 |
| $H$ = 16, $N_H$ = III, Flow = 16/0 | 881.87 ± 367.31 | 38.58 ± 3.13 | 21.66 ± 1.47 | 22.85 ± 1.57 | *100.00 ± 0.00 |
| $H$ = 16, $N_H$ = III, Flow = 0/4 | 89.12 ± 37.11 | 84.06 ± 3.99 | 37.00 ± 3.23 | 36.77 ± 5.20 | *100.00 ± 0.00 |
| $H$ = 16, $N_H$ = III, Flow = 0/8 | 93.06 ± 28.11 | 76.50 ± 6.64 | 30.94 ± 5.14 | 37.96 ± 5.69 | *100.00 ± 0.00 |
| $H$ = 16, $N_H$ = IV, Flow = 8/0 | 1893.36 ± 970.62 | 40.06 ± 6.68 | 20.01 ± 1.96 | 25.71 ± 4.91 | *100.00 ± 0.00 |
| $H$ = 16, $N_H$ = IV, Flow = 16/0 | 2212.13 ± 1084.13 | 41.60 ± 4.41 | 17.93 ± 0.92 | 29.77 ± 3.60 | *100.00 ± 0.00 |
| $H$ = 16, $N_H$ = IV, Flow = 0/4 | 56.60 ± 2.25 | 72.66 ± 6.46 | 32.09 ± 4.74 | 34.56 ± 5.52 | *100.00 ± 0.00 |
| $H$ = 16, $N_H$ = IV, Flow = 0/8 | 52.01 ± 2.10 | 79.58 ± 5.81 | 33.47 ± 2.84 | 36.19 ± 4.91 | *100.00 ± 0.00 |
| $H$ = 16, $N_H$ = V, Flow = 8/0 | 4434.32 ± 3059.38 | 31.63 ± 4.31 | 17.82 ± 1.02 | 20.77 ± 2.04 | *100.00 ± 0.00 |
| $H$ = 16, $N_H$ = V, Flow = 16/0 | 4115.67 ± 1891.56 | 47.58 ± 4.69 | 22.82 ± 2.46 | 26.51 ± 5.27 | *100.00 ± 0.00 |
| $H$ = 16, $N_H$ = V, Flow = 0/4 | 50.47 ± 1.54 | 83.71 ± 5.23 | *37.46 ± 5.13 | 42.63 ± 4.37 | *100.00 ± 0.00 |
| $H$ = 16, $N_H$ = V, Flow = 0/8 | 51.79 ± 0.78 | *85.15 ± 3.61 | 37.03 ± 2.35 | 42.73 ± 4.38 | *100.00 ± 0.00 |
| $H$ = 32, $N_H$ = I, Flow = 8/0 | 351.49 ± 157.14 | 38.59 ± 5.39 | 21.90 ± 2.62 | 25.23 ± 2.66 | *100.00 ± 0.00 |
| $H$ = 32, $N_H$ = I, Flow = 16/0 | 167.67 ± 116.18 | 45.10 ± 6.51 | 21.90 ± 2.76 | 24.84 ± 3.40 | *100.00 ± 0.00 |
| $H$ = 32, $N_H$ = I, Flow = 0/4 | 50.10 ± 1.55 | 73.09 ± 9.70 | 27.10 ± 2.42 | 40.78 ± 8.19 | *100.00 ± 0.00 |
| $H$ = 32, $N_H$ = I, Flow = 0/8 | 51.97 ± 2.57 | 58.80 ± 10.68 | 23.64 ± 2.46 | 30.77 ± 5.51 | *100.00 ± 0.00 |
| $H$ = 32, $N_H$ = II, Flow = 8/0 | 580.40 ± 250.82 | 38.80 ± 5.56 | 20.62 ± 1.89 | 25.80 ± 1.92 | *100.00 ± 0.00 |
| $H$ = 32, $N_H$ = II, Flow = 16/0 | 49.96 ± 1.60 | 46.52 ± 6.52 | 22.78 ± 1.69 | 27.16 ± 5.09 | *100.00 ± 0.00 |
| $H$ = 32, $N_H$ = II, Flow = 0/4 | *48.85 ± 0.92 | 60.12 ± 10.37 | 22.57 ± 2.44 | 37.51 ± 6.70 | *100.00 ± 0.00 |
| $H$ = 32, $N_H$ = II, Flow = 0/8 | 50.12 ± 2.29 | 69.33 ± 4.57 | 30.96 ± 2.55 | 35.11 ± 6.33 | *100.00 ± 0.00 |
| $H$ = 32, $N_H$ = III, Flow = 8/0 | 462.85 ± 169.59 | 43.86 ± 7.00 | 20.62 ± 2.41 | 30.58 ± 4.59 | *100.00 ± 0.00 |
| $H$ = 32, $N_H$ = III, Flow = 16/0 | 569.28 ± 219.42 | 54.12 ± 8.10 | 22.49 ± 1.98 | 31.49 ± 4.90 | *100.00 ± 0.00 |
| $H$ = 32, $N_H$ = III, Flow = 0/4 | 49.83 ± 1.25 | 67.93 ± 9.50 | 27.61 ± 3.98 | 31.87 ± 6.17 | *100.00 ± 0.00 |
| $H$ = 32, $N_H$ = III, Flow = 0/8 | 51.26 ± 1.53 | 70.68 ± 5.97 | 32.89 ± 3.07 | 28.56 ± 5.79 | *100.00 ± 0.00 |
| $H$ = 32, $N_H$ = IV, Flow = 8/0 | 50.79 ± 1.07 | 42.61 ± 8.46 | 18.84 ± 2.13 | 26.45 ± 3.68 | *100.00 ± 0.00 |
| $H$ = 32, $N_H$ = IV, Flow = 16/0 | 49.91 ± 1.54 | 45.15 ± 7.53 | 23.00 ± 2.26 | 27.90 ± 3.31 | *100.00 ± 0.00 |
| $H$ = 32, $N_H$ = IV, Flow = 0/4 | 49.82 ± 1.59 | 59.64 ± 7.63 | 26.64 ± 4.29 | 34.17 ± 7.65 | *100.00 ± 0.00 |
| $H$ = 32, $N_H$ = IV, Flow = 0/8 | 51.66 ± 1.79 | 65.31 ± 9.06 | 28.34 ± 3.83 | 38.84 ± 6.46 | *100.00 ± 0.00 |
| $H$ = 32, $N_H$ = V, Flow = 8/0 | 52.95 ± 1.36 | 39.37 ± 7.16 | 17.74 ± 1.36 | 28.73 ± 5.78 | *100.00 ± 0.00 |
| $H$ = 32, $N_H$ = V, Flow = 16/0 | 146.99 ± 94.70 | 56.50 ± 3.61 | 24.44 ± 2.00 | 34.28 ± 4.16 | *100.00 ± 0.00 |
| $H$ = 32, $N_H$ = V, Flow = 0/4 | 51.42 ± 1.68 | 74.23 ± 7.35 | 27.64 ± 1.85 | 35.90 ± 6.06 | *100.00 ± 0.00 |
| $H$ = 32, $N_H$ = V, Flow = 0/8 | 49.31 ± 1.81 | 76.36 ± 10.30 | 31.86 ± 4.07 | 41.32 ± 4.75 | *100.00 ± 0.00 |

Table 16: MNIST - OOD detection with AUC-ROC scores. Bold numbers indicate best score among single-pass models. Starred numbers indicate best scores among all models. Gray numbers indicate that R-PriorNet has seen samples from the Fashion-MNIST dataset during training.

|  | K. Alea. | K. Epist. | F. Alea. | F. Epist. | OODom Alea. | OODom Epist. |
|---|---|---|---|---|---|---|
| **Dropout** | $98.12 \pm 0.05$ | $97.16 \pm 0.11$ | $*99.26 \pm 0.03$ | $96.87 \pm 0.25$ | $15.19 \pm 1.70$ | $88.16 \pm 0.59$ |
| **Ensemble** | $98.17 \pm 0.07$ | $98.03 \pm 0.05$ | $99.15 \pm 0.07$ | $98.04 \pm 0.11$ | $11.83 \pm 1.81$ | $81.53 \pm 0.38$ |
| **NatPE** | $98.25 \pm 0.27$ | $99.48 \pm 0.03$ | $98.79 \pm 0.33$ | $*99.61 \pm 0.07$ | $*100.00 \pm 0.00$ | $*100.00 \pm 0.00$ |
| **R-PriorNet** | $*\textbf{99.44} \pm \textbf{0.09}$ | $*\textbf{99.59} \pm \textbf{0.08}$ | $100.00 \pm 0.00$ | $100.00 \pm 0.00$ | $99.44 \pm 0.16$ | $1.82 \pm 0.67$ |
| **EnD**$^2$ | $98.21 \pm 0.16$ | $98.65 \pm 0.15$ | $99.06 \pm 0.20$ | $99.21 \pm 0.17$ | $56.31 \pm 2.78$ | $4.60 \pm 1.89$ |
| **PostNet** | $98.73 \pm 0.05$ | $98.62 \pm 0.06$ | $98.65 \pm 0.35$ | $98.57 \pm 0.34$ | $*\textbf{100.00} \pm \textbf{0.00}$ | $*\textbf{100.00} \pm \textbf{0.00}$ |
| **NatPN** | $99.11 \pm 0.17$ | $99.25 \pm 0.09$ | $\textbf{99.13} \pm \textbf{0.24}$ | $\textbf{99.45} \pm \textbf{0.11}$ | $99.98 \pm 0.01$ | $*\textbf{100.00} \pm \textbf{0.00}$ |

Table 17: CIFAR-10 - OOD detection with AUC-ROC scores. Bold numbers indicate best score among single-pass models. Starred numbers indicate best scores among all models. Gray numbers indicate that R-PriorNet has seen samples from the SVHN dataset during training.

|  | SVHN Alea. | SVHN Epist. | CelebA Alea. | CelebA Epist. | OODom Alea. | OODom Epist. |
|---|---|---|---|---|---|---|
| **Dropout** | $84.67 \pm 1.42$ | $75.79 \pm 0.86$ | $75.95 \pm 3.61$ | $75.00 \pm 3.21$ | $21.75 \pm 6.36$ | $78.81 \pm 6.91$ |
| **Ensemble** | $88.08 \pm 0.85$ | $85.70 \pm 0.70$ | $78.80 \pm 0.82$ | $77.63 \pm 0.61$ | $40.53 \pm 9.95$ | $96.71 \pm 2.31$ |
| **NatPE** | $88.73 \pm 0.26$ | $*86.73 \pm 0.82$ | $*80.46 \pm 0.82$ | $*85.75 \pm 1.09$ | $92.45 \pm 1.37$ | $*99.56 \pm 0.11$ |
| **R-PriorNet** | $99.94 \pm 0.01$ | $99.98 \pm 0.00$ | $74.69 \pm 2.39$ | $70.63 \pm 6.14$ | $64.45 \pm 10.72$ | $59.61 \pm 13.23$ |
| **EnD**$^2$ | $*\textbf{89.56} \pm \textbf{0.67}$ | $\textbf{84.36} \pm \textbf{0.84}$ | $77.94 \pm 1.62$ | $78.14 \pm 1.66$ | $53.05 \pm 6.07$ | $4.42 \pm 2.57$ |
| **PostNet** | $85.52 \pm 0.58$ | $84.25 \pm 0.90$ | $75.68 \pm 2.05$ | $77.96 \pm 2.05$ | $93.00 \pm 2.46$ | $97.22 \pm 0.86$ |
| **NatPN** | $85.24 \pm 0.98$ | $81.74 \pm 1.05$ | $\textbf{77.98} \pm \textbf{1.22}$ | $\textbf{81.62} \pm \textbf{3.15}$ | $*\textbf{96.94} \pm \textbf{1.53}$ | $\textbf{97.41} \pm \textbf{1.63}$ |

Table 18: Bike Sharing - OOD detection with AUC-ROC scores. Bold numbers indicate best score among single-pass models. Starred numbers indicate best scores among all models. Normal and Poisson Regression are treated separately.

|  | Winter Epist. | Spring Epist. | Autumn Epist. | OODom Epist. |
|---|---|---|---|---|
| **Dropout-**$\mathcal{N}$ | $53.98 \pm 1.60$ | $51.24 \pm 0.96$ | $53.89 \pm 1.16$ | $*100.00 \pm 0.00$ |
| **Ensemble-**$\mathcal{N}$ | $81.53 \pm 1.11$ | $67.07 \pm 0.55$ | $67.78 \pm 1.31$ | $*100.00 \pm 0.00$ |
| **EvReg-**$\mathcal{N}$ | $55.26 \pm 2.14$ | $53.76 \pm 1.35$ | $52.39 \pm 1.31$ | $47.68 \pm 17.67$ |
| **NatPN-**$\mathcal{N}$ | $*\textbf{87.67} \pm \textbf{3.13}$ | $*\textbf{68.68} \pm \textbf{2.58}$ | $*\textbf{71.70} \pm \textbf{3.23}$ | $*\textbf{100.00} \pm \textbf{0.00}$ |
| **Dropout-**Poi | $55.30 \pm 0.58$ | $50.75 \pm 0.56$ | $59.05 \pm 1.15$ | $*\textbf{100.00} \pm \textbf{0.00}$ |
| **Ensemble-**Poi | $95.31 \pm 0.41$ | $75.62 \pm 0.85$ | $78.93 \pm 1.35$ | $*\textbf{100.00} \pm \textbf{0.00}$ |
| **NatPN-**Poi | $*\textbf{96.67} \pm \textbf{1.02}$ | $*\textbf{78.45} \pm \textbf{2.58}$ | $*\textbf{82.42} \pm \textbf{1.73}$ | $*\textbf{100.00} \pm \textbf{0.00}$ |

