# OpenReview forum: "Natural Posterior Network: Deep Bayesian Predictive Uncertainty for Exponential Family Distributions"
_ICLR.cc/2022/Conference — ICLR 2022 Spotlight_

### Official Review · Reviewer_qjTd · 2021-10-29

**Correctness:** 3
**Technical Novelty And Significance:** 2
**Empirical Novelty And Significance:** 2
**Recommendation:** 6
**Confidence:** 4

**Main Review:**

_Edit: Postrebuttal Update: The authors clarify most points in the rebuttal/include them in the new version. I increase my score from 5 $\to$ 6._
_________________

## Strengths
- While prior work in this direction of parameterizing a _'prior'_ distribution as part of the model focused mostly on the classification task offering the clear Categorical+Dirichler pair, this paper's focus on the exponential family allows for a clear generalization.

## Weaknesses
- While the paper title is phrased as targeting the predictive uncertainty, the paper's discussion and experiments solely focus on the epistemic/aleatoric decomposed setup (apart from some in-distribution calibration results). While the distinction between reducible and irreducible uncertainty is nice within a model, to better understand its performance or as a guiding signal for an active learning task, the paper explores no direction where decomposition of the predictive uncertainty actually is relevant.
- Related to the last point, the experimental comparisons are also all against methods that allow for a clean distinction instead of focusing on a broader class of models that target a predictive uncertainty independent of whether they allow for a theoretical/practical decomposition of such a term (e.g. a simple deterministic net with temperature scaling for calibration).
- The exponential family offers an extension away from the standard classification case. The paper has experiments with regression and count data, but as the authors acknowledge, the Poisson likelihood seems to perform poorly in the chosen data set. Having an experiment where the explicit modeling of the counts as counts was necessary would be a lot stronger.
- The paper is closely related to prior work by Charpentier et al. (2020), down to using the same input examples in Figure 2.
- ~~The related work discusses BNNs in the sampling-based paragraph but lacks a similar discussion of BNNs in the sampling-free methods~~ _Edit: Wrong claim from my side. I overlooked some references in the sampling-free paragraph as the keyword BNN was missing._


## Questions and minor comments
- Q1: In the contribution, the authors mention (point (3)) that the density is added "to the last predictor layer", a term that is never mentioned again. Can the authors comment on what they mean by that?
- Q2: As mentioned above on the epistemic/aleatoric point, a stated goal is to aim  "at accurately modelling both aleatoric and epistemic uncertainty". Can the authors comment on that goal and why they target it as the paper lacks discussions/explorations on the usefulness of the distinction between the two?
- The experiments compare against Amini et al. (2020) in the regression case but lack a comparison in the classification against Sensoy et al. (2018) Amini et al.'s closest relation and predecessor. That work also has the benefit of not OOD data during training as the PriorNets do.
- The BNN discussion comments on the huge computational budget of Izmailov et al. (2021) as a downside. This comment ignores that this cost was due to their goal of inferring the true posterior as optimally as possible via a costly HMC approach. However, that work includes many cheaper methods (be they variational inference-based or MCMC-based, such as SGLD, SGHMC) that demonstrate good performance for a much cheaper computational cost.
- Table 1 contains **(1),(2)** whose meaning is only resolved several pages later. At the same time, the page includes three different sets of (1),(2) (twice in two separate lists, and once each as equation numbers). A restructuring might be helpful to guide the reader towards the desired meaning in table 1.
- While the term is only taken from prior work, I would strongly encourage the authors to rename their objective to something other than "the Bayesian loss". It is an objective whose optimum is given by the posterior, which is nice, but apart from that, there is no such thing as _THE Bayesian loss_ or even "the principled Bayesian loss"..
- NatPN Ensembles are discussed, but the NatPE abbreviation is missing from the paragraph and never formally introduced
- In equation (6), as soon as $\lambda \neq 1$, L is not actually proportional to the right-hand side anymore.

### Minor Appendix comments
- App B: The ELBO loss refers again to the appendix
- App B: There appears a discussion on a prior over $y$, while talking about distributions over $\theta$
- In the appendix $P(\theta)$ and $P(y|\theta)$ seem to be used interchangeably
- The appendix states that the entropy of the posterior is used to estimate the predictive uncertainty, but the posterior predictive uncertainty should actually be computed after marginalization over the posterior.

### Typos
- Throughout the paper, please follow the ICLR style guide properly. E.g. captions belong above tables and below figures.
- Table 1 refers to eq (7) which is in the appendix. It should probably mean (5), i.e. the same loss just in the main paper
- End of page 3: Bishop (2006) -> (Bishop, 2006).
- page 4: predictive posterior distribution -> posterior predictive distribution
- page 5: using each NatPN member ~~all~~ separately
- The abbreviation OOD is introduced, but ID is not and needs to be guessed from the context


__________
Sensoy et al., Evidential Deep Learning to Quantify Classification Uncertainty, NeurIPS 2018


**Summary Of The Paper:**

The paper targets the task of getting useful predictive uncertainty, as measured by in-distribution calibration and out-of-distribution detection capability. Towards this, the authors focus on the distributions from the exponential family, which allow for a closed-form posterior form whose parameters are predicted via a neural net, together with a flexible density provided by a normalizing-flow net for OOD detection.


**Summary Of The Review:**

While the paper provides a useful extension and next step to the prior work by Charpentier et al. (2020), the theoretical contribution seems minor and without great empirical improvements. Explorations in areas where the proposed setup with the uncertainty decomposition becomes relevant could strengthen it a lot.

---

> ### Author Response · Authors · 2021-11-23
> **Answer to Reviewer qjTd**
>
> We would like to thank you for the valuable comments and suggestions. Following your concerns, we provided new results, provided further comments and suggestions and updated the manuscript. Most of the new results are in the appendix and in the rebuttal. If all the reviewers agree, we are happy to move them into the main paper. Finally, we hope that you agree with our comment and are happy to provide additional clarifications in case you have follow-up questions.
>
> **Aleatoric/Epistemic/Predictive Uncertainty.** The necessity of capturing different sources of uncertainty has already been studied ([Hora, 1996]( https://www.semanticscholar.org/paper/Aleatory-and-epistemic-uncertainty-in-probability-Hora/14c417ab194d47928e74f3728e394ae46a4ef793); Malinin et al., 2018). Further, epistemic uncertainty might not be the best uncertainty metric to select samples for active learning ([Zhao et al., 2021](https://openreview.net/pdf?id=Mu2ZxFctAI); [Mussmann et al.](https://arxiv.org/pdf/1806.06123.pdf)). Thus, the main goal of the paper does *not* include showing the usefulness of the distinction between aleatoric and epistemic uncertainty (see 'contribution' paragraph). In particular, we would like to emphasize that aleatoric uncertainty and epistemic uncertainty are not orthogonal uncertainty metrics for NatPN. Indeed, the expected sufficient statistic of the posterior distribution $\chi^{post, (i)} = E_{Q(\chi^{post, (i)}, n^{post, (i)})}[\theta^{(i)}]$ carries the inherent aleatoric uncertainty of the target distribution and $n^{post, (i)}$ carries the epistemic uncertainty (see Sec. 3.1). Importantly, it converges to the sufficient statistic of the prior distribution $\chi^{post, (i)}$ which should encode high aleatoric uncertainty according to the principle of maximum entropy (see Sec. 3.3). Thus, we do not expect a large difference between aleatoric and epistemic uncertainty in many cases.
>
> In practice, since the epistemic uncertainty is expected to represent the lack of knowledge for a prediction, it is expected to be particularly useful when comparing ID data with data very different training data where we have knowledge (i.e. OODom). As desired, we observe that the epistemic uncertainty achieves better results than aleatoric for OODom data on Tab. 2, 3, 8, 9, 12, 13, 16, 17. In contrast, all baselines which are not able to explcitly model epistemic uncertainty (e.g. Ensemble and Drop-Out) generally achieve lower performances than NatPN when applied to OOD (e.g. see Tab. 2, 3, 4, 5, 6). Specifically, standard deterministic neural networks achieve poor OOD performances (see new results in Tab. 8, 9, 11 in the appendix). This is not surprising since standard (ReLU) deterministic networks are known to be over-confident far from training data (Hein et al., 2019). Finally, we provide additional results in Fig. 4, 5 in the appendix where we visualize the aleatoric and the epistemic uncertainty for two new toy datasets with three classes. The first dataset has 900 training examples for the three classes (see Fig. 4) while the second dataset has 900, 600 and 300 training examples for the three classes (see Fig. 5). As desired, the predictions are more aleatorically certain close to training samples and more epistemically uncertain classes with fewer training examples. In particular, predictions are very epistemically uncertainty far from training data. This suggests that the epistemic uncertainty is well-calibrated.
>
> **Performance of Poisson Distribution.** While NatPN$-Poi$ is less calibrated than NatPN$-\mathcal{N}$ on the Bike sharing dataset, NatPN$-Poi$ significantly outperforms NatPN$-\mathcal{N}$ for OOD detection which also evaluate the uncertainty predictions. Indeed, NatPN$-Poi$ achieves $+30$%, $+14 $% and $+15$% absolute improvement for OOD detection against Winter, Spring and Autumn seasons compared to NatPN$-\mathcal{N}$ (see Tab. 5). Therefore, modeling counts with Poisson distributions is necessary and highly benefitial for this task.
>
> **Relation to PostNet.** We made the relation to PostNet (Charpentier et al. (2020)) clear in the last paragraph of Sec. 3.2. Both methods use normalizing flows and parametrize some conjugate prior. Howerver, NatPN is significantly more applicable since it can be used for *any* task described by an exponentaly family target distribution and is significantly *faster* (see Tab. 7). Input examples in Fig. 2 are images from the standard MNIST dataset selected to illustrate three interesting cases for uncertainty estimation.

---

> > ### Author Response · Authors · 2021-11-23
> > **Answer to Reviewer qjTd (2)**
> >
> > **Related Work BNNs.** First, we extensively cite BNNs sampling-free methods in the related work section. Specifically, we reference (Gast & Roth, 2018; Postels et al., 2019; Shekhovtsov & Flach, 2019; Wang et al., 2016) which model uncertainty on the weights and/or activation levels. Further, we precisely discuss the similarity with (Wang et al., 2016) which is a sampling free BNN which models uncertainty on the weights with exponential families distirbutions. We are happy to cite any other explicit references about sampling-free BNNs. Finally, we emphasize that NatPN is not a standard BNN since it parametrizes the *conjugate prior* of the *target* variable with *exponential family distributions*.
> >
> > Second, we explicitly mention that Izmailov et al. (2021) aim "...to achieve a more exact Bayesian inference..." i.e. inferring the true posterior (see paragraph 'Sampling-based methods' in Sec. 2). Further, Izmailov et al. (2021) explcitly mentione that SGMCMC (incl. SGLD and SGHMC) and Mean-field variatinal inference (MFVI) provide relatively distinct predictive distributions from HMC. Thus, we should be very careful when making judgements about true Bayesian neural networks based on the SGMCMC or MFVI performance. Finally, we already reference more scalable BNN methods such that (Dusenberry et al., 2020; Farquhar et al., 2020; Osawa et al., 2019) which are also valuable BNNs methods in practice.
> >
> > **Last Predictor Layer.** NatPN architecture mostly consists in one encoder/decoder architecture and one additional normalizing flow which fits a density $P(z^{(i)}|\omega)$ on the encoder output space. In practice, the encoder architecture $f_\theta$ is instantiated with some common architecture e.g. MLP, Conv, DenseDepth (Eigen et al., 2014) and the decoder $g_\phi$ is a linear layer. Hence, the core difference between NatPN and common architecures is that NatPN adds a single normalizing flow modelling $P(z^{(i)}|\omega)$ next to the predictor linear layer $g_\phi$ (see Fig. 2).
> >
> > **Additional Baselines.** We compare to three strong and recent baselines parametrizing Dirichlet conjugate prior (i.e. R-PriorNet, EnD$^2$ and PostNet). In particular, Evidential Networks (Sensoy et al. (2018)) have shown similar or worse performance than those baselines for uncertainty tasks like correct/wrong prediction detection, attack detection and OOD detection in ([Kopetzki at al., 2020](https://arxiv.org/pdf/2010.14986.pdf)).
> >
> > Nonetheless, we added the new results for standard deterministic networks and SNGP with the same core achitecture than other models in Tab. 8, 9, 10, 11 in the appendix. Standard deterministic networks and SNGP model aleaotric uncertainty but do not model epistemic uncertainty explicitly. NatPN is significantly better for OOD detection than Standard (ReLU) determininistic networks which are known to be overconfident far from training data (Hein et al., 2019). For SNGP, we used the hyper-parameters suggested by the original paper (Lakshminarayanan et al., 2020). In contrast with NatPN, SNGP is initially designed for classification tasks only. NatPN achieves comparable performance for OOD detection and improves calibration metrics compared to SNGP.
> >
> > **Loss Name.** We reused the name previously used in [9] which is particularly convenient and adapted to describe the formulation (5). Indeed the formulation (5) is a *loss* (potentially different from the ELBO) and has the *Bayesian* property to be optimal whenthe learned distribution is equal to the posterior (see Sec. 3.5 and App.B). We now removed the term "principled" and opt for the term "Bayesian formulation" when suited. We are happy to take any name suggestions accurately describing the *Bayesian* and *loss* nature of the formulation (5).
> >
> > **Minor.** We brought modifications to the paper based on the reviewer comments. We changed the notation to refer to different terms in formulation (5). We introduced the notation NatPE in Sec. 3.4. We changed the loss notation in equation (6).

---

### Official Review · Reviewer_QsKU · 2021-10-31

**Correctness:** 4
**Technical Novelty And Significance:** 4
**Empirical Novelty And Significance:** 4
**Recommendation:** 8
**Confidence:** 3

**Main Review:**

# Pros

- Utilizing normalizing flows for density estimation for calibration and OOD detection is intuitive
- NatPN can be used for both regression and classification tasks. The majority of previous works neglect regression tasks and focus solely on classification.
- NatPN maintains a fast inference time as compared to other sampling based BNN’s
- NatPN achieves high aleatoric uncertainty on OOD datasets, which performs well against baselines compared against.

# Cons

- Section 2: The authors state that modelling distributions over the weights results in ‘pathological’ behavior. This is vague and hard to understand, what is the pathology?
- Section 2 (sampling free methods): parameterizing conjugate prior distributions which $\rightarrow$ parameterizing conjugate prior distributions, which (even though I believe this fixes the problem with the sentence, it likely needs to be rewritten because the long list of references breaks the flow over three lines…)
- It is not immediately clear to me why the 2nd term of the loss in equation 5 is necessary. If each component of the model behaves as expected, wont $p( x | \omega )$ remove the need for entropy regularization?
- There are recent advances in single pass uncertainty such as SNGP [1] which are nor compared to. At least for classification tasks, I would be interested to see the performance difference between the proposed method and SNGP.
- Table 5 shows results for models which are trained on the Kin8nm and Concrete UCI datasets, and evaluated in terms of epistemic uncertainty on other UCI datasets. These datasets all have different input dimensions, so how does a model trained with a specific input dimension evaluate a dataset with another dimension?

# Minor

- section 4.2 or training OOD $\rightarrow$ or training on OOD


# References

[1] Liu, J. Z., Lin, Z., Padhy, S., Tran, D., Bedrax-Weiss, T., & Lakshminarayanan, B. (2020). Simple and principled uncertainty estimation with deterministic deep learning via distance awareness. arXiv preprint arXiv:2006.10108.

**Summary Of The Paper:**

The authors propose NatPN, a model which can estimate predictive uncertainty for classification and regression tasks. NatPN predicts the parameters of the posterior distribution which belongs to the exponential family. Contrary to other algorithms, NatPN requires no OOD data for training, and is able to evaluate the uncertainty in a single forward pass utilizing normalizing flows for density estimation.

**Summary Of The Review:**

There are a few concerns which are highlighted in the sections above, but overall the paper is a strong submission which solves a relevant problem in a novel and intuitive way. My score reflects everything stated in my review and is likely to be updated based on the authors response to my questions above.

---

> ### Author Response · Authors · 2021-11-23
> **Answer to Reviewer QsKU**
>
> We would like to thank you for the valuable comments and suggestions. Following your concerns, we provided new results, provided further comments and suggestions and updated the manuscript. Most of the new results are in the appendix and in the rebuttal. If all the reviewers agree, we are happy to move them into the main paper. Finally, we hope that you agree with our comment and are happy to provide additional clarifications in case you have follow-up questions.
>
> **BNN Clarification.** The term 'pathology' was taken from (Foong et al., 2020). They find simple cases where neither mean-field Gaussian nor Monte Carlo dropout method can have substantially increased uncertainty in between well-separated regions of low uncertainty. They also empirically show that this pathologies persist when performing variational inference in deep networks. Further, some BNN methods based on variational inference are limited to simple architectures and datasets (Graves, 2011). Finally, BNNs tend to show poor generalization under domain shifts, and scalable BNNs might provide relatively distinct predictive distributions from the computationally expensive Hamiltonian Monte Carlo method which is guaranteed to asymptotically produce samples from the true posterior (Izmailov et al., 2021).
>
> **Regularization Factor.** In theory, previous works (Bissiri et al., 2016; Shawe-Taylor & Williamson, 1997; Zellner, 1988) showed that the full loss (5) is guaranteed to converge to the true posterior distribution i.e. $Q^{\text{post},(i)} \approx Q^*(\theta | x^{(i)})$, thus motivating to include the entropy regularizer. By default, the learned posterior update of NatPN in Eq. (4) is not guaranteed to converge to the true posterior update by optimizing with any arbitrary loss.
>
> We show in Tab. A, B, C results for Categorical, Normal and Poisson target distributions for different regularizing factors $\lambda$. We observe that $\lambda=1e^{-5}$ generally achieves good performances. The larger MNIST dataset is less sensitive w.r.t the entropy regularizer than the smaller Bike Sharing dataset. We are happy to add these results to the hyper-parameter study in the appandix I.5.
>
> **Additional Baselines.** We added the new results for standard deterministic networks and SNGP with the same core achitecture than other models in Tab. 8, 9, 10, 11 in the appendix. Standard deterministic networks and SNGP model aleaotric uncertainty but do not model epistemic uncertainty explicitly. NatPN is significantly better for OOD detection than Standard (ReLU) determininistic networks which are known to be overconfident far from training data (Hein et al., 2019). For SNGP, we used the hyper-parameters suggested by the original paper (Lakshminarayanan et al., 2020). In contrast with NatPN, SNGP is initially designed for classification tasks only. NatPN achieves comparable performance for OOD detection and improves calibration metrics compared to SNGP. Finally, note that our work focuses on models parametrizing conjugate prior distributions which do not include SNGP or standard deterministic networks.
>
> **OOD for UCI Datasets.** The Concrete and Kin8nm UCI datasets have both the same input dimensions of $8$ (see appendix F). Therefore, there is no need to modify the input dimenion of this dataset to use it as OOD for the other dataset.
>
> **Minor.** We fixed typos and improved phrasing in the related work.

---

> > ### Author Response · Authors · 2021-11-23
> > **Answer to Reviewer QsKU (2)**
> >
> > Table A: Classification results on MNIST with Categorical target distribution for different regularizing factors $\lambda$ in Eq. (6).
> >
> > |   $\lambda$ |   Accuracy |   Brier |   K. Alea. |   K. Epist. |   OODom Alea. |   OODom Epist. |
> > |--------------:|----------------:|-------------------:|------------------------:|------------------------:|------------------------------:|------------------------------:|
> > |         0e+00 |           99.48 $\pm$ 0.01 |               1.09 $\pm$ 0.03 |                   99.21 $\pm$ 0.21 |                   99.33 $\pm$ 0.07 |                        100.00 $\pm$ 0.00 |                        100.00 $\pm$ 0.00 |
> > |         1e-05 |           99.45 $\pm$ 0.04 |               1.05 $\pm$ 0.06 |                   99.35 $\pm$ 0.09 |                   99.34 $\pm$ 0.13 |                        100.00 $\pm$ 0.00 |                        100.00 $\pm$ 0.00 |
> > |         1e-04 |           99.47 $\pm$ 0.03 |               1.14 $\pm$ 0.15 |                   98.58 $\pm$ 0.47 |                   98.40 $\pm$ 1.05 |                        100.00 $\pm$ 0.00 |                        100.00 $\pm$ 0.00 |
> > |         1e-03 |           81.51 $\pm$ 17.93 |              20.14 $\pm$ 18.68 |                   93.16 $\pm$ 4.56 |                   86.10 $\pm$ 10.11 |                         95.00 $\pm$ 5.00 |                        100.00 $\pm$ 0.00 |
> > |         1e-02 |           81.53 $\pm$ 17.93 |              25.89 $\pm$ 17.33 |                   88.65 $\pm$ 4.05 |                   81.81 $\pm$ 9.35 |                         95.00 $\pm$ 5.00 |                        100.00 $\pm$ 0.00 |
> > |         1e-01 |            9.80 $\pm$ 0.00 |              94.87 $\pm$ 0.00 |                   75.00 $\pm$ 0.00 |                   47.57 $\pm$ 0.82 |                         75.00 $\pm$ 0.00 |                        100.00 $\pm$ 0.00 |
> > |         1e+00 |            9.80 $\pm$ 0.00 |              94.87 $\pm$ 0.00 |                   75.00 $\pm$ 0.00 |                   47.57 $\pm$ 0.81 |                         75.00 $\pm$ 0.00 |                        100.00 $\pm$ 0.00 |
> >
> > Table B: Regression results on Bike Sharing with Normal target distribution for different regularizing factors $\lambda$ in Eq. (6).
> >
> > |   $\lambda$ |   RMSE |   Calibration |   Winter Epist. |   Spring Epist. |   Autumn Epist. |   OODom EPist. |
> > |--------------:|------------:|------------------------:|------------------------:|------------------------:|------------------------:|------------------------------:|
> > |         0e+00 |       56.86 $\pm$ 4.64 |                    3.34 $\pm$ 0.83 |                   59.70 $\pm$ 3.94 |                   27.26 $\pm$ 2.02 |                   28.76 $\pm$ 3.42 |                        100.00 $\pm$ 0.00 |
> > |         1e-05 |       50.38 $\pm$ 1.22 |                    1.81 $\pm$ 0.25 |                   50.96 $\pm$ 5.87 |                   22.70 $\pm$ 0.98 |                   28.52 $\pm$ 3.06 |                        100.00 $\pm$ 0.00 |
> > |         1e-04 |       56.79 $\pm$ 5.93 |                    1.64 $\pm$ 0.12 |                   57.85 $\pm$ 8.46 |                   24.19 $\pm$ 2.45 |                   35.43 $\pm$ 5.81 |                        100.00 $\pm$ 0.00 |
> > |         1e-03 |       52.03 $\pm$ 3.51 |                    3.67 $\pm$ 1.19 |                   53.47 $\pm$ 11.18 |                   21.98 $\pm$ 1.73 |                   26.80 $\pm$ 4.48 |                        100.00 $\pm$ 0.00 |
> > |         1e-02 |       57.33 $\pm$ 6.69 |                    8.71 $\pm$ 1.92 |                   42.60 $\pm$ 10.13 |                   21.63 $\pm$ 3.06 |                   22.84 $\pm$ 1.99 |                        100.00 $\pm$ 0.00 |
> > |         1e-01 |      194.11 $\pm$ 1.46 |                    9.10 $\pm$ 1.18 |                   89.99 $\pm$ 5.38 |                   44.83 $\pm$ 6.63 |                   44.26 $\pm$ 9.81 |                        100.00 $\pm$ 0.00 |
> > |         1e+00 |      199.93 $\pm$ 1.41 |                   17.19 $\pm$ 2.49 |                   94.93 $\pm$ 2.57 |                   44.17 $\pm$ 10.42 |                   47.37 $\pm$ 5.43 |                        100.00 $\pm$ 0.00 |

---

> > > ### Author Response · Authors · 2021-11-23
> > > **Answer to Reviewer QsKU (3)**
> > >
> > > Table C: Count prediction results on Bike Sharing with Poisson target distribution for different regularizing factors $\lambda$ in Eq. (6).
> > >
> > > |   $\lambda$ |    RMSE |   Calibration |   Winter Epist. |   Spring Epist. |   Autumn Epist. |   OODom EPist. |
> > > |--------------:|------------:|------------------------:|------------------------:|------------------------:|------------------------:|------------------------------:|
> > > |         0e+00 |       52.48 $\pm$ 1.08 |                   31.15 $\pm$ 1.79 |                   88.90 $\pm$ 4.24 |                   37.02 $\pm$ 2.88 |                   47.05 $\pm$ 5.45 |                        100.00 $\pm$ 0.00 |
> > > |         1e-05 |       53.22 $\pm$ 2.31 |                   30.50 $\pm$ 2.09 |                   74.05 $\pm$ 2.73 |                   31.06 $\pm$ 0.85 |                   34.94 $\pm$ 5.13 |                        100.00 $\pm$ 0.00 |
> > > |         1e-04 |       53.51 $\pm$ 0.78 |                   29.97 $\pm$ 1.48 |                   87.09 $\pm$ 1.86 |                   35.78 $\pm$ 1.81 |                   39.52 $\pm$ 4.22 |                        100.00 $\pm$ 0.00 |
> > > |         1e-03 |       52.13 $\pm$ 1.37 |                   30.47 $\pm$ 1.50 |                   88.05 $\pm$ 2.79 |                   41.02 $\pm$ 3.87 |                   43.18 $\pm$ 2.82 |                        100.00 $\pm$ 0.00 |
> > > |         1e-02 |       50.74 $\pm$ 1.05 |                   32.41 $\pm$ 1.22 |                   77.13 $\pm$ 5.95 |                   34.44 $\pm$ 4.08 |                   35.91 $\pm$ 5.47 |                        100.00 $\pm$ 0.00 |
> > > |         1e-01 |       53.92 $\pm$ 1.15 |                   36.51 $\pm$ 0.68 |                   64.15 $\pm$ 12.81 |                   26.05 $\pm$ 4.37 |                   38.50 $\pm$ 5.69 |                        100.00 $\pm$ 0.00 |
> > > |         1e+00 |      111.08 $\pm$ 10.69 |                   47.16 $\pm$ 1.55 |                   83.77 $\pm$ 3.90 |                   37.08 $\pm$ 3.81 |                   43.83 $\pm$ 5.87 |                        100.00 $\pm$ 0.00 |

---

> > > > ### Comment · Reviewer_QsKU · 2021-11-25
> > > > **Thanks for the clarifications**
> > > >
> > > > Thanks for the clarifications and extra results. While I think SNGP would likely perform well on many of the experiments in the main section, I think the authors have made a sizable contribution in generalizing and improving upon PN.
> > > >
> > > > I still think the sentence with the word 'pathology' is extremely vague and remains present in the updated version of the paper. I would strongly urge the authors to update this sentence in the camera ready version of the paper to be more concrete, as I would be clueless and confused as to the meaning without the above explanation in the author response.
> > > >
> > > > Thank you for the response, I will adjust my score accordingly.

---

### Official Review · Reviewer_cKhx · 2021-11-01

**Correctness:** 3
**Technical Novelty And Significance:** 3
**Empirical Novelty And Significance:** 2
**Recommendation:** 8
**Confidence:** 3

**Main Review:**

Strengths:

- The paper is well written and clear.
- The theoretical contribution, Theorem 1, is a nice addition to the empirical results.
- NatPN performs on par or better than similar prior methods from the literature.
- NatPN extends the Posterior Network to the exponential family distributions which encompasses important tasks such as regression with Gaussian likelihood and counting with Poisson likelihood.
- NatPN uses a single normalizing flow for classification rather than one flow per class as per Posterior Networks, which in theory enables scaling to datasets with many classes.

Weaknesses:

- NatPN has the theoretical advantage of being able to scale to many classes given that it uses a single normalizing flow. This is a significant potential advantage over Posterior Networks. However the paper only evaluates on classifiation tasks with 10 classes: CIFAR-10, MNIST, FMNIST. It would be very useful to verify the scalability of the method to > 10 classes on datasets such as CIFAR-100 and Imagenet which have well defined associated OOD tasks e.g. CIFAR-100 vs. SVHN, CIFAR-100 corrupted, Imagenet-C/A/R/V2 and are standard benchmarks in the OOD literature.
- Please also include OOD performance e.g. accuracy, RMSE, etc. for the near OOD datasets e.g. CIFAR-10 corrupted. Methods which are can provide good OOD detection should also be capable of generalizing to near OOD data points.
- The work is nice, but somewhat incremental vs. the Posterior Network. The extension to exponential families is natural. The use of a single normalizing flow for classification seems more significant to me, but it is unclear if this causes new training difficulties (see questions below).
-  The empirical results do not show a clear gain from NatPN vs. similar competing methods.

Additional suggested improvements:

- Comparing against other deterministic/single-pass methods from the literature that do not require normalizing flows would be a significant improvement to the experiments, in particular SNGP (Liu et al., 2020) and DUQ (van Amersfoort et al., 2020).
- When comparing NatPN to the Posterior Network it is hard to disentangle the performance changes for classification coming from the use of a single normalizing flow rather than C normalizing flows vs. the new update rules. Could you add an ablation to disentangle these contributions? For example, could you maintain the old Posterior Network update rules but use a single normalizing flow or could you use C normalizing flows but use the NatPN update rules per flow? This would help assessing the relative importance of the contributions of this paper for classification tasks.

Questions:

- Could the authors comment on the disadvantages of not scaling the number of normalizing flows with the number of classes as per PostNet?
- The second sentence quoted is not clear to me: "On the other hand, a single normalized density is trained to output the evidence update n (i) = NH P(z (i) | ω) accounting for the epistemic uncertainty. The intuition is that increasing the evidence on training data during training forces the evidence everywhere else (incl. far from training data) to decrease thanks to the density normalization constraint." Could the authors clarify?
- It is not clear to me where the Bayesian NatPN Ensemble posterior update for n^{post, (i)} comes from? How is this not effectively (m - 1) times overcounting the effective training samples given that each ensemble member is presumably trained on the same dataset from a different random initialization?
- Please comment further or point to the point in the appendix where this is clarified: “Additionally, we observed that “warm-up” training and “fine-tuning” of the density helped to improve uncertainty estimation for more complex flows and datasets. Thus, we trained the normalizing flow density to maximize the likelihood of the latent representations before and after the joint optimization while keeping all other parameters fixed.” If a significant change in the training procedure for NatPN vs. Posterior Networks and other baselines is required then this is a significant disadvantage of the method.
- What is the definition of the “predicted evidence” for all models? In particular for baseline models: dropout and deep ensembles?

Nits:

- For comparison purposes it would be useful to include results for a standard/deterministic network trained without dropout (with standard MAP parameter estimation).
- NatPE is never defined. Assumed it is the Bayesian NatPN Ensemble.
- NatPE should be included in the batched inference time table. Also nit: this should be a Table {number} not Figure 4.
- I am surprised the Dropout baseline inference time is so slow in Figure 4. It seems reasonable to compute the inference time assuming that the sampling of the dropout masks can be parallelized along the batch dimension. From the numbers it looks like what is being measured is the inference time if the 5 dropout masks are sampled serially. In addition, given this efficient implementation, 5 dropout masks seems low for a fair comparison to a deep ensemble, a useful additional result would be to include a Dropout baseline with num_samples=128 or similar.

**Summary Of The Paper:**

The paper introduces a new method Natural Posterior Network (NatPN), an extension to the Posterior Network (Charpentier et al., 2020). NatPN aims to tackle the problem of enabling calibrated uncertainty estimates for in and out-of-distribution inputs for exponential family likelihoods parameterized by deep neural networks. In particular NatPN maps input samples into a latent space on which a normalizing flow is defined. A mapping from this latent space and its corresponding likelihood under the normalizing flow to the parameters of exponential family update rule is defined. The exponential family posterior tends towards the prior parameters for data points far from the training data and prior has little influence for data points within the training distribution.

NatPN makes two significant changes to the Posterior Network 1) it generalizes the method to the exponentially family, allowing for regression and counting tasks beyond just classification as per the Posterior Network and 2) it uses a single normalizing flow rather than class specific normalizing flows as per the Posterior Network, in theory this aids the scalability of the method to datasets with many classes. In addition a minor change is made to the update rule of the posterior parameters.

The paper presents extensive empirical results on classification, regression and counting tasks across several relatively small scale datasets. NatPN is shown to perform on par or slightly better than similar methods from the literature.

**Summary Of The Review:**

The paper is well written and seems technically correct. The proposed method NatPN extends the Posterior Network to the exponential family distributions and has the advantage of using a single normalizing flow rather than C flows for classification tasks. The empirical results are backed up with an interesting theoretical guarantee under what seem to be reasonable conditions.

A number of aspects of the paper are unclear to me, I have listed these questions above.

Given the slightly incremental contribution of the extension of Posterior Networks to the exponential family distributions the paper would benefit from additional empirical validation e.g. with the above suggested ablation, the addition of SNGP and DUQ baselines and the inclusion of OOD performance metrics. Additionally, given that I see the use of a single normalizing flow as a main contribution of this paper I would like to have the question regarding the modified training procedure clarified and I would like to see empirical validation of the scaling to CIFAR-100 and/or Imagenet.

Nonetheless, the paper is a solid contribution and I believe worthy of acceptance given the information currently available.

---

> ### Author Response · Authors · 2021-11-23
> **ANswer to Reviewer cKhx**
>
> We would like to thank you for the valuable comments and suggestions. Following your concerns, we provided new results, provided further comments and suggestions and updated the manuscript. Most of the new results are in the appendix and in the rebuttal. If all the reviewers agree, we are happy to move them into the main paper. Finally, we hope that you agree with our comment and are happy to provide additional clarifications in case you have follow-up questions.
>
> **Performance Gain over Competitors.** For regression, NatPN outperforms other single-pass models for 23/26 scores for regression, thus showing a clear gain over competing methods (see Sec.4.2 'Regression'). For classification, excluding PostNet where we do not expect strong improvement due to their akin design (see Sec. 4.2 'Classification'), NatPN achieves 19/30 top-1 scores and 28/30 top-2 scores among single-pass models while NatPE achieves 28/30 top-1 scores among multiple-pass models. In particular, it achieves very strong gain over all other baselines for OODom (see Tab. 2-6) and tabular data (see Tab. 2). Further, NatPN also demonstrates a clear gain over PostNet since it is significantly faster and is applicable to other tasks than classification (e.g. see Tab. 7). Finally, the major empirical gain of NatPN is that it works for *all* tasks described by an exponential family target distributions which is not the case for all the other baselines parametrizing conjugate prior distributions.
>
> **Number of NF.** Both PostNet and NatPN use the input-dependent Bayesian udpate as update rule for classification. Thus, transfering PostNet to use a single normalizing flow would result in the NatPN model, and uncertainty performance results in Tab.2-4 and speed results in Fig. 7 alraedy assess the relative importance of using a single normalizing flow against $C$ normalizing flows for classification tasks. Specifically, we do not expect a strong improvement over PostNet due to their akin design (see Sec. 4.2 'Classification') but NatPN is significantly faster than PostNet (see Tab. 7). We provide new additional results where NatPN uses a mixture of 10 normalizing flows to model the density $P(z^{(i)}|\omega)$ instead of a single normalizing flow in Tab. A, B. While a potential advantage of using $C$ normalizing flows instead of a single normalizing flow could be that it learns more complex distributions, both model using a mixture or a single normalizing flow achieve similar performances.
>
> **NF Clarification.** Because of the normalization constraint of the density (i.e. $\int P(z | \omega) dz = 1$), the density value $P(z | \omega)$ cannot be large for all possible latent representations $z$. It implies that increasing the evidence $n^{(i)} = N_H P(z^{(i)} | \omega)$ for latent representations of (training) In-Distribution inputs $x^{(i)}$ -- by increasing the density values $P(z | \omega)$ -- enforces the evidence $\tilde{n}^{(j)} = N_H P(\tilde{z}^{(j)} | \omega)$ for latent representations of Out-Of-Distribution inputs $\tilde{x}^{(j)}$ to decrease. In particular, this applied to input data far from training data (i.e. $||\tilde{x}|| \rightarrow + \infty$) as stated by Thm. 1.
>
> **NatPE Clarification.** The predicted evidence of an ensemble of $m$ NatPN is indeed likely to be $m$ times larger than the evidence of a single NatPN model. However, The uncertainty estimates of model for an input $x^{(i)}$ should be compared to the uncertainty of the *same* model on another input $x^{(j)}$. In particular, the evidence $n^{(i)} = n^{prior} + \sum_k^{m} n_k^{(i)}$ of an ensemble of $m$ NatPN models for an input $x^{(i)}$ should be compared to the evidence $n^{(j)} = n^{prior} + \sum_k^{m} n_k^{(j)}$ of the *same* ensemble of $m$ NatPN models for another input $x^{(j)}$. Hence, NatPE is not overcounting the evidence for one input sample $x^{(i)}$ compared to another input sample $x^{(j)}$. In particular, NatPE is also guaranteed to predict low evidence for inputs very differen from training data (i.e. $||\tilde{x}|| \rightarrow + \infty$) because each NatPN member follows Thm. 1. Finally, we followed the reviewer's suggestion and added a reference to the notation NatPE in Sec. 3.5 and report the NatPE batched inference time in Tab. 7. An ensemble of $5$ NatPN is as expected approximately $\times 5$ slower than a single NatPN model.
>
> **Additional Datasets.** The results for CIFAR-10 corrupted are provided in Fig.3. Further, our experiments already include 8 datasets for training including tabular and images input data with categorical, continuous and count target values, and uses four different types of OOD samples including 6 additional datasets and left-out data. Hence, this extensive evaluation shows that the unified exponential family framework of NatPN provides well-calibrated uncertainty estimates for multiple data types which is the main goal of the paper. Scaling NatPN to larger datasets with more classes is an interesting direction for future work.

---

> > ### Author Response · Authors · 2021-11-23
> > **Answer to Reviewer cKhx (2)**
> >
> > **Warm-up/Fine-tuning Clarification.** The warm-up and fine-tuning simply correspond to train the parameters of the Normalizing flow only by maximizing the likelihood of the training latent representations i.e. $\arg\max_{\omega} \log P(z^{(i)}| \omega)$. This does not add a significant complexity to the training procedure and is common practice in Machine Learning  (Ash & Adams, 2020; Käding et al., 2016).
> >
> > **Predicted Evidence.** Thank you for pointing out this sentence where our phrasing is imprecise. For models parametrizing conjugate-prior (i.e. R-PriorNet, EnD$^2$, PostNet, NatPN), the epistemic unceraitny is measured using the predicted evidence which corresponds to the evidence parameter $n$ of the predicted conjugate prior. For the ensemble and drop-out models, the epistemic uncertainty is measured using the variance of the predicted mean (i.e. the variance of the winning probability class for classification and the variance of the mean for regression and class count prediction). We clarified this point in Sec.4.1 and in the appendix.
> >
> > **Drop-Out Model.** We show results of dropout with 128 samples in Tab. C, D, E. As observed by (Ovadia et al., 2019), higher number of samples does not lead to significant performance improvement. The mask sampling cannot be parallized along the batch dimension since it would require a larger GPU to process the batch of $4086$ CIFAR10 inputs and the batch of $4$ of NYU inputs, thus giving the drop-out model an unfair advantage.
> >
> > **Additional Baselines.** We added the new results for standard deterministic networks and SNGP with the same core achitecture than other models in Tab. 8, 9, 10, 11 in the appendix. Standard deterministic networks and SNGP model aleaotric uncertainty but do not model epistemic uncertainty explicitly. NatPN is significantly better for OOD detection than Standard (ReLU) determininistic networks which are known to be overconfident far from training data (Hein et al., 2019). For SNGP, we used the hyper-parameters suggested by the original paper (Lakshminarayanan et al., 2020). In contrast with NatPN, SNGP is initially designed for classification tasks only. NatPN achieves comparable performance for OOD detection and improves calibration metrics compared to SNGP. Finally, note that our work focuses on models parametrizing conjugate prior distributions which do not include SNGP or standard deterministic networks.
> >
> > Table A: Classification results on MNIST with Categorical target distribution using Mixture of $10$ normalizing flows to model $P(z^{(i)}|\omega)$.
> >
> > |    |   Accuracy |   Brier |   K. Alea. |   K. Epist. | F. Alea. | F. Epist. |  OODom Alea. |   OODom Epist. |
> > |--------------:|----------------:|-------------------:|------------------------:|------------------------:|---:|---:|------------------------------:|------------------------------:|
> > |         NatPN (Mixture NF) |           99.45 $\pm$ 0.02 |               1.09 $\pm$ 0.05 |                   99.37 $\pm$ 0.05 |                   99.31 $\pm$ 0.06 | 99.27 $\pm$ 0.23 | 99.51 $\pm$ 0.07 |                         99.99 $\pm$ 0.01 |                        100.00 $\pm$ 0.00 |
> > |         NatPN (Single NF) |           99.47 $\pm$ 0.02 |               1.09 $\pm$ 0.03 |                   99.20 $\pm$ 0.20 |                   99.39 $\pm$ 0.08 | 99.16 $\pm$ 0.28 | 99.54 $\pm$ 0.09 |                         99.99 $\pm$ 0.01 |                        100.00 $\pm$ 0.00 |
> >
> > Table B: Regression results on Bike Sharing with Normal target distribution using Mixture of $10$ normalizing flows to model $P(z^{(i)}|\omega)$.
> >
> > |    |   RMSE |   Calibration |   Winter Epist. |   Spring Epist. |   Autumn Epist. |   OODom Epist. |
> > |--------------:|------------:|------------------------:|------------------------:|------------------------:|------------------------:|------------------------------:|
> > |         NatPN (Mixture NF) |       59.54 $\pm$ 4.76 |                    2.47 $\pm$ 0.25 |                   56.91 $\pm$ 4.84 |                   22.78 $\pm$ 1.46 |                   29.19 $\pm$ 2.72 |                        100.00 $\pm$ 0.00 |
> > |         NatPN (Single NF) |       49.85 $\pm$ 1.38 |                    1.95 $\pm$ 0.34 |                   55.04 $\pm$ 6.81 |                   23.25 $\pm$ 1.20 |                   27.78 $\pm$ 2.47 |                        100.00 $\pm$ 0.00 |

---

> > > ### Author Response · Authors · 2021-11-23
> > > **Answer to Reviewer cKhx (3)**
> > >
> > > Table C: Classification results on MNIST with Categorical target distribution for drop-out with 5 and 128 samples.
> > >
> > > |    |   Accuracy |   Brier |   K. Alea. |   K. Epist. |   F. Alea. |   F. Epist. |   OODom Alea. |   OODom Epist. |
> > > |----------------:|----------------:|-------------------:|------------------------:|------------------------:|-------------------------------:|-------------------------------:|------------------------------:|------------------------------:|
> > > |               Dropout (5 samples) |           99.45 $\pm$ 0.01 |               1.07 $\pm$ 0.05 |                   98.27 $\pm$ 0.05 |                   97.82 $\pm$ 0.08 |                          99.40 $\pm$ 0.03 |                          98.01 $\pm$ 0.14 |                         43.86 $\pm$ 1.62 |                         74.09 $\pm$ 0.92 |
> > > |             Dropout (128 samples) |           99.45 $\pm$ 0.02 |               1.01 $\pm$ 0.06 |                   98.44 $\pm$ 0.06 |                   98.44 $\pm$ 0.08 |                          99.47 $\pm$ 0.03 |                          98.23 $\pm$ 0.14 |                         44.11 $\pm$ 1.65 |                         93.62 $\pm$ 0.40 |
> > >
> > > Table D: Classification results on Fashion MNIST with Categorical target distribution for drop-out with 5 and 128 samples.
> > >
> > > |                       |   Accuracy |   Brier |   MNIST Alea. |   MNIST Epist. |   K. Alea. |   K. Epist. |   OODom Alea. |   OODom Epist. |
> > > |:----------------------------|----------------:|-------------------:|-----------------------:|-----------------------:|------------------------:|------------------------:|------------------------------:|------------------------------:|
> > > | Dropout (5 samples) |           92.44 $\pm$ 0.17 | 13.89 $\pm$ 0.31 | 60.75 $\pm$ 1.41 | 75.85 $\pm$ 1.73 | 76.57 $\pm$ 1.30 | 92.48 $\pm$ 0.46 | 39.97 $\pm$ 0.69 | 90.90 $\pm$ 1.74
> > > | Dropout (128 samples) |           92.57 $\pm$ 0.13 |              13.71 $\pm$ 0.31 |                  59.58 $\pm$ 1.84 |                  84.13 $\pm$ 1.31 |                   76.28 $\pm$ 1.46 |                   96.46 $\pm$ 0.24 |                         39.97 $\pm$ 0.64 |                         99.35 $\pm$ 0.35 |
> > >
> > >
> > > Table E: Regression results on Bike Sharing with Normal target distribution for drop-out with 5 and 128 samples.
> > >
> > > |   |   RMSE |   Calibration |   Winter Epist. |   Spring Epist. |   Autumn Epist. |   OODom Epist. |
> > > |----------------:|------------:|------------------------:|------------------------:|------------------------:|------------------------:|------------------------------:|
> > > |             Dropout (5 samples) |       70.20 $\pm$ 1.30 | 6.05 $\pm$ 0.77 | 15.26 $\pm$ 0.51 | 13.66 $\pm$ 0.16 | 15.11 $\pm$ 0.46 | 99.99 $\pm$ 0.01 |
> > > |             Dropout (128 samples) |       68.80 $\pm$ 1.28 |                    7.24 $\pm$ 0.96 |                   16.69 $\pm$ 0.52 |                   13.92 $\pm$ 0.20 |                   15.48 $\pm$ 0.49 |                        100.00 $\pm$ 0.00 |

---

> > > > ### Comment · Reviewer_cKhx · 2021-11-24
> > > > **Response**
> > > >
> > > > Thanks to the authors for their detailed response. I would have **really liked to see NatPN applied to larger-scale datasets** e.g. ImageNet or even CIFAR-100. I highly encourage the authors to add this experiment to **this** paper for the camera ready. Nonetheless the authors have clarified a number of my questions and have added SNGP and the deterministic method as a baseline, as well as adding the requested Dropout ablation. Therefore I have updated my score accordingly.

---

### Official Review · Reviewer_pzkS · 2021-11-02

**Correctness:** 3
**Technical Novelty And Significance:** 3
**Empirical Novelty And Significance:** 3
**Recommendation:** 8
**Confidence:** 3

**Main Review:**

**Update:** changed my score (see comment) from 6 to 8.

**Strengths**:

The paper makes three claims.
Firstly, NatPN can be applied to all problems with an exponential family likelihood distribution. The paper provides strong evidence that this is the case. They perform multiple experiments on classification tasks and two experiments on regression and count prediction. The experiments are thorough and tested on multiple datasets each.
Secondly, they claim that empirically NatPNs outperform other state-of-the-art methods in different uncertainty-related metrics such as calibration and OOD detection.
Across all experiments, their method beats most competitors or provides competitive estimates across multiple metrics. Their experiments provide more than enough evidence for this claim.
Thirdly, they provide a theoretical statement on the OOD behaviour of NatPNs in the limit. Their theorom shows that under mild assumptions NatPNs recover the prior far away from the training data.

Overall, the experiments convince me of the main goal of the paper, namely that NatPN provides well-calibrated uncertainty estimates for multiple data types.

Furthermore, there are other miscellaneous strengths. The paper is well written and the figures are (mostly) of high quality. The paper doesn’t shy away from talking about its limitations. The method is fast and easy to set up which is especially beneficial for practitioners. All of these are important strengths of the paper that are often undervalued.

All in all, the paper provides a significant contribution to an important problem.

**Weaknesses:**

There are two main (and easily fixable) weaknesses.

a) I think the role of the normalizing flow is underexplained. It is stated multiple times that the normalizing flow provides the evidence updates and its purpose is to estimate epistemic uncertainty. The remaining questions for me are 1. From which space to which does the NF map the latent variable z? 2. Why is the arrow in Figure 2 from a Gaussian space into the latent space, rather than from the latent space to n^(i)? I thought the main purpose was to influence n^(i)? 3. Which experiments show that the normalizing flow contributes meaningfully to the epistemic uncertainty (see b))?

b) Figure 1 does a good job of showing the intuition behind NatPNs but it lacks some components and a discussion in the text. The authors choose to show aleatoric (un-)certainty and predictive certainty respectively but don’t show epistemic (un-)certainty. Technically, you could deduce epistemic uncertainty from aleatoric and predictive uncertainty but it would be easier to compare and follow your argument if it was made explicit. Furthermore, I would like to see an explicit discussion of the results. Why, for example, is the difference between aleatoric and predictive uncertainty so low? Is there no or little epistemic uncertainty in this setting? There are two things that would convince me more regarding this problem: a) an additional toy experiment similar to Figure 1 which includes more epistemic uncertainty, e.g. with fewer data points. This could show that the epistemic uncertainty is well-calibrated. b) An argument for why the epistemic uncertainty is (presumably) so low in your setting. a) and b) are not mutually exclusive, doing both would convince me more.

**There are a couple of minor improvements:**
Figure 1 is not referenced in the main text.
I find it hard to spot the difference w.r.t the symbols in Figure 1. Maybe just making it less crowded would already improve visibility.
In the last paragraph of 3.1, you mention “warm-up” and “fine-tuning”. It would be helpful to explain these concepts briefly in one additional sentence or provide references.

**What would raise my score?**

I would raise my score by 1 or 2 points if my main weaknesses are well addressed or if evidence is provided that my criticism is the consequence of a misunderstanding.

I would raise my score even further if I’m convinced of the high significance of this work. This will be mostly dependent on the estimate of more expert reviewers but I’m also open to arguments by the authors.



**Summary Of The Paper:**

This paper proposes Natural Posterior Networks (NatPNs), a technique to provide uncertainty-estimation to tasks where the likelihood is an exponential family, e.g. classification or regression. NatPNs leverage the properties of exponential families, i.e. conjugacy and closed-form posterior predictive for fast and elegant Bayesian Inference. This is combined with a Bayesian treatment of the loss function that allows for end-to-end training. Furthermore, it uses a normalizing flow to account for the epistemic uncertainty. The normalizing flow density is supposed to increase the evidence on the training data and thereby decrease it everywhere else. The posterior distribution is on the outputs not on the weights of the NN which makes it easier to train. The paper provides one theorem for why the posterior uncertainty is meaningful and multiple experiments showing practical improvements for multiple exponential families and datasets.

NatPNs are an extension of Posterior Networks. However, the differences are significant and well explained in the paper. From my perspective, the contributions are clearly novel enough to justify this new paper.


**Summary Of The Review:**

**Recommendation:**

6: marginally above the acceptance threshold; Tendency towards 8, if concerns addressed.

**Summary:**

The paper presents a novel method and provides empirical and theoretical evidence for its effectiveness. There are two parts of the paper that are currently insufficiently explained. Firstly, the technical setup of the normalizing flow and how it is used to create epistemic uncertainty is underexplained IMO. Secondly, the toy experiments in Figure 1 should be explained and discussed in more detail. I'm specifically currently uncertain why the epistemic uncertainty is not shown and presumably quite low in this setting.

I think these concerns can be easily addressed by the authors and would be willing to increase my score to an acceptance (8) if they are addressed well.

---

> ### Author Response · Authors · 2021-11-23
> **Answer to Reviewer pzkS**
>
> We would like to thank you for the valuable comments and suggestions. Following your concerns, we provided new results, provided further comments and suggestions and updated the manuscript. Most of the new results are in the appendix and in the rebuttal. If all the reviewers agree, we are happy to move them into the main paper. Finally, we hope that you agree with our comment and are happy to provide additional clarifications in case you have follow-up questions.
>
> **Normalizing Flow Clarification.** The NF learns a more complex transformed distribution $P(z^{(i)}|\omega)$ on the *latent space* by transforming a simple base distribution (which is usually set to a Gaussian) in the NF *base space*. Therefore the black NF arrow indicates the transformation from the NF base space to the transformed latent space. The colored arrows indicate the evidence computation which combines the learned distribution defined on the latent space with the certainty budget (i.e. $n^{(i)} = N_H P(z^{(i)}|\omega)$), and the expected sufficient statistic computation from the latent representation (i.e. $\chi^{(i)} = g_\psi(z^{(i)})$).
>
> In practice, since the epistemic uncertainty is expected to represent the lack of knowledge for a prediction, it is expected to be particularly useful when comparing ID data with data very different training data the model trained on (e.g. OODom). As desired, we observe in our experiments that the epistemic uncertainty achieves better results than aleatoric for OODom data (see Tab. 2, 3, 8, 9, 12, 13, 16, 17). Further, (Charpentier et al., 2020) showed already that using NF densities for pseudo-count computation was significantly improving calibration and OOD performance compared to using no densities or mixture of Gaussians densities. Finally, NatPN using a mixture of 10 normalizing flows to model the density $P(z^{(i)}|\omega)$ instead of a single normalizing flow does not consitently improve performance (see new additional results in Tab. A, B). These results suggest that a single NF density contributes to meaningful epistemic uncertainty.
>
> In theory, Thm. 1 shows that learning a density contributes to meaningful epistemic uncertainty contrary to standard ReLU networks. Indeed, the evidence parameter is guaranteed become very low indicating a high epistemic uncertainty for data far from training data (i.e $n^{(i)} \rightarrow 0$ when $||x^{(i)}|| \rightarrow \infty$). This theorem suggests that the NF density contributes to meaningful epistemic uncertainty.
>
> **Epistemic Uncertainty.** (a) We provide additional results in Fig. 4, 5 in the appendix where we visualize the aleatoric and the epistemic uncertainty for two new toy datasets with three classes. The first dataset has 900 training examples for the three classes (see Fig. 4) while the second dataset has 900, 600 and 300 training examples for the three classes (see Fig. 5). As desired, the predictions are more aleatorically certain close to training samples and more epistemically uncertain for classes with fewer training examples. In particular, predictions are very epistemically uncertainty far from training data. This suggests that the epistemic uncertainty is well-calibrated.
>
> (b) In Fig.1, the epistemic uncertainty is also well calibrated. Indeed, the epistemic uncertainty is again high far from training data and low close to training data. We would like to emphasize that aleatoric uncertainty and epistemic uncertainty are not orthogonal uncertainty metrics for NatPN. Indeed, the expected sufficient statistic of the posterior distribution $\chi^{post, (i)} = E_{Q(\chi^{post, (i)}, n^{post, (i)})}[\theta^{(i)}]$ carries the inherent aleatoric uncertainty of the target distribution (see Sec. 3.1). Importantly, it converges to the sufficient statistic of the prior distribution $\chi^{post, (i)}$ which should encode high aleatoric uncertainty according to the principle of maximum entropy (see Sec. 3.3). Thus, we do not expect a large difference between aleatoric and epistemic uncertainty in many cases.
>
> **Minor.** Fig.1 is referenced at the end of the contribution paragraph. Further, we added new references regarding warm-up training and fine-tuning which are common practices in Machine Learning. Warm-up and fine-tuning steps correspond to train the NF parameters by maximizing $\log P(z^{(i)}|\omega)$.

---

> > ### Author Response · Authors · 2021-11-23
> > **Answer to Reviewer pzkS (2)**
> >
> > Table A: Classification results on MNIST with Categorical target distribution using Mixture of $10$ normalizing flows to model $P(z^{(i)}|\omega)$.
> >
> > |    |   Accuracy |   Brier |   K. Alea. |   K. Epist. | F. Alea. | F. Epist. |  OODom Alea. |   OODom Epist. |
> > |--------------:|----------------:|-------------------:|------------------------:|------------------------:|---:|---:|------------------------------:|------------------------------:|
> > |         NatPN (Mixture NF) |           99.45 $\pm$ 0.02 |               1.09 $\pm$ 0.05 |                   99.37 $\pm$ 0.05 |                   99.31 $\pm$ 0.06 | 99.27 $\pm$ 0.23 | 99.51 $\pm$ 0.07 |                         99.99 $\pm$ 0.01 |                        100.00 $\pm$ 0.00 |
> > |         NatPN (Single NF) |           99.47 $\pm$ 0.02 |               1.09 $\pm$ 0.03 |                   99.20 $\pm$ 0.20 |                   99.39 $\pm$ 0.08 | 99.16 $\pm$ 0.28 | 99.54 $\pm$ 0.09 |                         99.99 $\pm$ 0.01 |                        100.00 $\pm$ 0.00 |
> >
> > Table B: Regression results on Bike Sharing with Normal target distribution using Mixture of $10$ normalizing flows to model $P(z^{(i)}|\omega)$.
> >
> > |    |   RMSE |   Calibration |   Winter Epist. |   Spring Epist. |   Autumn Epist. |   OODom Epist. |
> > |--------------:|------------:|------------------------:|------------------------:|------------------------:|------------------------:|------------------------------:|
> > |         NatPN (Mixture NF) |       59.54 $\pm$ 4.76 |                    2.47 $\pm$ 0.25 |                   56.91 $\pm$ 4.84 |                   22.78 $\pm$ 1.46 |                   29.19 $\pm$ 2.72 |                        100.00 $\pm$ 0.00 |
> > |         NatPN (Single NF) |       49.85 $\pm$ 1.38 |                    1.95 $\pm$ 0.34 |                   55.04 $\pm$ 6.81 |                   23.25 $\pm$ 1.20 |                   27.78 $\pm$ 2.47 |                        100.00 $\pm$ 0.00 |

---

> > > ### Comment · Reviewer_pzkS · 2021-11-24
> > > **Good rebuttal, changed score from 6 to 8**
> > >
> > > Thanks for the answers.
> > >
> > > Both of my concerns have been addressed. Firstly, the exact details of the normalizing flow are now clear to me.
> > > Secondly, the additional experiments in the appendix convince me that the aleatoric and epistemic uncertainty is well-calibrated.
> > >
> > > I have changed my score from 6 to 8 accordingly. Nice work!

---

### Official Review · Reviewer_x9Pt · 2021-11-02

**Correctness:** 3
**Technical Novelty And Significance:** 3
**Empirical Novelty And Significance:** 3
**Recommendation:** 6
**Confidence:** 4

**Main Review:**

**Relevance**:
Uncertainty estimation is a highly active research area. Methods that don’t require calculating an average over multiple forward passes are arguably particularly relevant for practical applications. While most machine learning problems, at least in a research setting, are typically classification or regression problems, generalizations beyond those settings are of interest for the community.

**Novelty**:
While the paper builds off PostNet (Charpentier et al., 2020), which is clearly referenced, the generalization to non-classification tasks is novel. The paper further proposes an alternative scheme of using a single normalizing flow for density estimation on the features, which is computationally more scalable for a higher number of classes. The overall contribution seems solid in terms of novelty, although there might be related work that I’m not aware of, as I have been following this branch of the literature only superficially.

**Clarity**:
Overall the paper is clear regarding the proposed method, although I found it helpful to also read the PostNet paper. However, the typesetting is too dense, presumably due to the space constraints. In particular in section 3 I would suggest using subsection and subsubsections to structure the text instead of paragraphs. Similarly pages 7 and 8 could benefit from more spacing, the use of bold font inside of paragraphs does not seem ideal to me (in particular when some terms also get underlined, visually I find this page distracting). I’m not sure if an additional page will be available during the discussion period, otherwise I would suggest moving most of the detailed discussion of the setup into the appendix, most of the datasets and metrics are fairly standard.

**Empirical evaluation**:
The classification experiments and metrics are mostly standard and the regression problems appear to overlap with (Amini et al., 2020). I would have ideally liked to see the paper push a little bit further in the direction of problems that are not classification or regression, but it is good to see that there is a count prediction problem and that OOD detection is improved by a Poisson observation model.

In an orthogonal direction, I would also be interested in seeing results for a classification problem with a higher number of classes, as this is the setting where the difference to PostNet of using a single normalizing flow instead of one per class becomes more relevant. I would be interested in seeing if either approach runs into limitations in terms of fitting the data. Ideally this would be shown for ImageNet (if PostNet can be scaled to 1000 classes), but I understand that this may be too much to ask over the discussion period and would be happy with Cifar100 results -- I assume this should be runnable with negligible overhead based off the Cifar10 experiments.

**Other notes/questions**:
* How sensitive is the method to the value of $\lambda$ (the entropy hyperparameter)? An ablation study would be useful, I would ideally like to see a plot showing how the different metrics vary across values of $\lambda$ with some dataset for each likelihood model (normal/categorical/poisson).
* Please place table captions above the tables as per the formatting instructions.
* Page 7, final line: missing transition between sentences?
* There are some minor inconsistencies in the reference formatting (full first names vs only initials, title capitalization).

**Summary Of The Paper:**

This paper proposes a single-pass uncertainty estimation method for neural networks by predicting the update to the natural parameters of the conjugate prior to the likelihood of the data distribution and optimizing a corresponding ‘Bayesian loss’. In that it generalizes PostNet (Charpentier et al., 2020) beyond classification. The paper reports competitive or improved performance across a range of datasets and against both standard and recent baselines.

**Summary Of The Review:**

The paper presents a solid and well-motivated generalization of prior work. There is some room for improvement in terms of the writing as well as some additional experiments, however, I would already consider this a solid submission as it is. Therefore I am **leaning towards an accept**

---

> ### Author Response · Authors · 2021-11-23
> **Answer to Reviewer x9Pt**
>
> We would like to thank you for the valuable comments and suggestions. Following your concerns, we provided new results, provided further comments and suggestions and updated the manuscript. Most of the new results are in the appendix and in the rebuttal. If all the reviewers agree, we are happy to move them into the main paper. Finally, we hope that you agree with our comment and are happy to provide additional clarifications in case you have follow-up questions.
>
> **Layout.** We followed the reviewer suggestion by making subsections in section 3, removing bold test inside paragraphs, moving the detailed set-up description to the appendix, adjusting table captions and fixing inconsitencies in the reference formatting. We are happy to further improve the paper formatting based on reviewers comments.
>
> **Additional Datasets.** Our experiments already include 8 datasets for training including tabular and images input data with categorical, continuous and count target values. Further, the uncertainty evaluation uses four different types of OOD samples including 6 additional datasets, left-out data and CIFAR-10 corrupted. Hence, this extensive evaluation shows that the unified exponential family framework of NatPN provides well-calibrated uncertainty estimates for multiple data types which is the main goal of the paper. We agree that scaling NatPN to larger datasets with more classes is an interesting direction for future work.
>
> **Regularization Factor.** As suggested, we show in Tab. A, B, C new results for Categorical, Normal and Poisson target distributions for different regularizing factors $\lambda$. We observe that $\lambda=1e^{-5}$ generally achieves good performances. The larger MNIST dataset is less sensitive w.r.t the entropy regularizer than the smaller Bike Sharing dataset. We are happy to add these results to the hyper-parameter study in the appendix I.5.
>
> Table A: Classification results on MNIST with Categorical target distribution for different regularizing factors $\lambda$ in Eq. (6).
>
> |   $\lambda$ |   Accuracy |   Brier |   K. Alea. |   K. Epist. |   OODom Alea. |   OODom Epist. |
> |--------------:|----------------:|-------------------:|------------------------:|------------------------:|------------------------------:|------------------------------:|
> |         0e+00 |           99.48 $\pm$ 0.01 |               1.09 $\pm$ 0.03 |                   99.21 $\pm$ 0.21 |                   99.33 $\pm$ 0.07 |                        100.00 $\pm$ 0.00 |                        100.00 $\pm$ 0.00 |
> |         1e-05 |           99.45 $\pm$ 0.04 |               1.05 $\pm$ 0.06 |                   99.35 $\pm$ 0.09 |                   99.34 $\pm$ 0.13 |                        100.00 $\pm$ 0.00 |                        100.00 $\pm$ 0.00 |
> |         1e-04 |           99.47 $\pm$ 0.03 |               1.14 $\pm$ 0.15 |                   98.58 $\pm$ 0.47 |                   98.40 $\pm$ 1.05 |                        100.00 $\pm$ 0.00 |                        100.00 $\pm$ 0.00 |
> |         1e-03 |           81.51 $\pm$ 17.93 |              20.14 $\pm$ 18.68 |                   93.16 $\pm$ 4.56 |                   86.10 $\pm$ 10.11 |                         95.00 $\pm$ 5.00 |                        100.00 $\pm$ 0.00 |
> |         1e-02 |           81.53 $\pm$ 17.93 |              25.89 $\pm$ 17.33 |                   88.65 $\pm$ 4.05 |                   81.81 $\pm$ 9.35 |                         95.00 $\pm$ 5.00 |                        100.00 $\pm$ 0.00 |
> |         1e-01 |            9.80 $\pm$ 0.00 |              94.87 $\pm$ 0.00 |                   75.00 $\pm$ 0.00 |                   47.57 $\pm$ 0.82 |                         75.00 $\pm$ 0.00 |                        100.00 $\pm$ 0.00 |
> |         1e+00 |            9.80 $\pm$ 0.00 |              94.87 $\pm$ 0.00 |                   75.00 $\pm$ 0.00 |                   47.57 $\pm$ 0.81 |                         75.00 $\pm$ 0.00 |                        100.00 $\pm$ 0.00 |

---

> > ### Author Response · Authors · 2021-11-23
> > **Answer to Reviewer x9Pt (2)**
> >
> > Table B: Regression results on Bike Sharing with Normal target distribution for different regularizing factors $\lambda$ in Eq. (6).
> >
> > |   $\lambda$ |   RMSE |   Calibration |   Winter Epist. |   Spring Epist. |   Autumn Epist. |   OODom EPist. |
> > |--------------:|------------:|------------------------:|------------------------:|------------------------:|------------------------:|------------------------------:|
> > |         0e+00 |       56.86 $\pm$ 4.64 |                    3.34 $\pm$ 0.83 |                   59.70 $\pm$ 3.94 |                   27.26 $\pm$ 2.02 |                   28.76 $\pm$ 3.42 |                        100.00 $\pm$ 0.00 |
> > |         1e-05 |       50.38 $\pm$ 1.22 |                    1.81 $\pm$ 0.25 |                   50.96 $\pm$ 5.87 |                   22.70 $\pm$ 0.98 |                   28.52 $\pm$ 3.06 |                        100.00 $\pm$ 0.00 |
> > |         1e-04 |       56.79 $\pm$ 5.93 |                    1.64 $\pm$ 0.12 |                   57.85 $\pm$ 8.46 |                   24.19 $\pm$ 2.45 |                   35.43 $\pm$ 5.81 |                        100.00 $\pm$ 0.00 |
> > |         1e-03 |       52.03 $\pm$ 3.51 |                    3.67 $\pm$ 1.19 |                   53.47 $\pm$ 11.18 |                   21.98 $\pm$ 1.73 |                   26.80 $\pm$ 4.48 |                        100.00 $\pm$ 0.00 |
> > |         1e-02 |       57.33 $\pm$ 6.69 |                    8.71 $\pm$ 1.92 |                   42.60 $\pm$ 10.13 |                   21.63 $\pm$ 3.06 |                   22.84 $\pm$ 1.99 |                        100.00 $\pm$ 0.00 |
> > |         1e-01 |      194.11 $\pm$ 1.46 |                    9.10 $\pm$ 1.18 |                   89.99 $\pm$ 5.38 |                   44.83 $\pm$ 6.63 |                   44.26 $\pm$ 9.81 |                        100.00 $\pm$ 0.00 |
> > |         1e+00 |      199.93 $\pm$ 1.41 |                   17.19 $\pm$ 2.49 |                   94.93 $\pm$ 2.57 |                   44.17 $\pm$ 10.42 |                   47.37 $\pm$ 5.43 |                        100.00 $\pm$ 0.00 |
> >
> > Table C: Count prediction results on Bike Sharing with Poisson target distribution for different regularizing factors $\lambda$ in Eq. (6).
> >
> > |   $\lambda$ |    RMSE |   Calibration |   Winter Epist. |   Spring Epist. |   Autumn Epist. |   OODom EPist. |
> > |--------------:|------------:|------------------------:|------------------------:|------------------------:|------------------------:|------------------------------:|
> > |         0e+00 |       52.48 $\pm$ 1.08 |                   31.15 $\pm$ 1.79 |                   88.90 $\pm$ 4.24 |                   37.02 $\pm$ 2.88 |                   47.05 $\pm$ 5.45 |                        100.00 $\pm$ 0.00 |
> > |         1e-05 |       53.22 $\pm$ 2.31 |                   30.50 $\pm$ 2.09 |                   74.05 $\pm$ 2.73 |                   31.06 $\pm$ 0.85 |                   34.94 $\pm$ 5.13 |                        100.00 $\pm$ 0.00 |
> > |         1e-04 |       53.51 $\pm$ 0.78 |                   29.97 $\pm$ 1.48 |                   87.09 $\pm$ 1.86 |                   35.78 $\pm$ 1.81 |                   39.52 $\pm$ 4.22 |                        100.00 $\pm$ 0.00 |
> > |         1e-03 |       52.13 $\pm$ 1.37 |                   30.47 $\pm$ 1.50 |                   88.05 $\pm$ 2.79 |                   41.02 $\pm$ 3.87 |                   43.18 $\pm$ 2.82 |                        100.00 $\pm$ 0.00 |
> > |         1e-02 |       50.74 $\pm$ 1.05 |                   32.41 $\pm$ 1.22 |                   77.13 $\pm$ 5.95 |                   34.44 $\pm$ 4.08 |                   35.91 $\pm$ 5.47 |                        100.00 $\pm$ 0.00 |
> > |         1e-01 |       53.92 $\pm$ 1.15 |                   36.51 $\pm$ 0.68 |                   64.15 $\pm$ 12.81 |                   26.05 $\pm$ 4.37 |                   38.50 $\pm$ 5.69 |                        100.00 $\pm$ 0.00 |
> > |         1e+00 |      111.08 $\pm$ 10.69 |                   47.16 $\pm$ 1.55 |                   83.77 $\pm$ 3.90 |                   37.08 $\pm$ 3.81 |                   43.83 $\pm$ 5.87 |                        100.00 $\pm$ 0.00 |

---

> > > ### Comment · Reviewer_x9Pt · 2021-11-24
> > > **Response**
> > >
> > > Thank for providing an ablation study on $\lambda$ and improving the layout of the paper. I'm inclined to increase my score, however I disagree with the argument on not providing additional results. My point was not that there aren't enough experiments (i.e. insufficient quantity), but that I would like to see results in a qualitatively different setting with a higher number of classes, which is potentially challenging for the proposed method. I appreciate that it can be difficult to provide results over a short time period and that asking for more experiments is a rather cheap point of criticism for a reviewer to make. Hence I specifically suggested using CIFAR100, which should require no more than a couple of lines of code to be changed from the CIFAR10 experiments as the hyperparameters should transfer.
> > >
> > > I believe that this paper makes a solid contribution and therefore argue for acceptance, but will leave my score as is and would encourage the authors, in line with R`cKhx`, to add results on CIFAR100 to the camera-ready version if the paper is accepted.

---

### Decision · Program_Chairs · 2022-01-20

**Decision:**

Accept (Spotlight)

**Comment:**

This paper presents a method for producing higher quality uncertainty estimates by mapping the predictions from an arbitrary (e.g. deep learning) model to an exponential family distribution.  This is achieved by using the model to map from the inputs to a low-dimensional latent space and then using a normalizing flow to map to the parameters of the distribution.  The authors show empirically that this improves over a variety of baselines on a number of OOD and uncertainty quantification tasks.  This paper received 5 reviews who all agreed that the paper should be accepted (6, 6, 8, 8, 8).  The reviewers in general found the method novel compared to existing literature, compelling and the results strong.  Multiple reviewers asked for experiments with higher dimensional output distributions (e.g. CIFAR 100) and had concerns regarding the "entropy regularization" term (akin to the beta term in a beta VAE, this is a constant applied to the entropy term).  The reviewers seemed satisfied with the author response, however, and the concensus decision is to accept.